# Tight Bounds for Schrödinger Potential Estimation in Unpaired Data Translation

**Nikita Puchkin**[*] **& Denis Suchkov**[*]
HSE University, Faculty of Computer Science
{npuchkin,d.suchkov}@hse.ru

**Alexey Naumov**
HSE University and Steklov Mathematical Institute of Russian Academy of Sciences
anaumov@hse.ru

**Denis Belomestny**
Duisburg-Essen University and HSE University
denis.belomestny@uni-due.de

## Abstract

Modern methods of generative modelling and unpaired data translation based on Schrödinger bridges and stochastic optimal control theory aim to transform an initial density to a target one in an optimal way. In the present paper, we assume that we only have access to i.i.d. samples from the initial and final distributions. This makes our setup suitable for both generative modelling and unpaired data translation. Relying on the stochastic optimal control approach, we choose an Ornstein-Uhlenbeck process as the reference one and estimate the corresponding Schrödinger potential. Introducing a risk function as the Kullback-Leibler divergence between couplings, we derive tight bounds on the generalization ability of an empirical risk minimizer over a class of Schrödinger potentials, including Gaussian mixtures. Thanks to the mixing properties of the Ornstein-Uhlenbeck process, we almost achieve fast rates of convergence, up to some logarithmic factors, in favourable scenarios. We also illustrate the performance of the suggested approach with numerical experiments.

## 1 Introduction

The Schrödinger bridge problem has emerged as a powerful framework in modern generative modelling and unpaired data translation (in particular, image-to-image translation). It formulates the task of transforming an initial distribution into a target one via a probabilistic coupling that is as close as possible to a prescribed reference dynamics in the sense of relative entropy. In the present paper, we assume that both the initial and target distributions are absolutely continuous and denote their densities with respect to the Lebesgue measure by $\rho_0$ and $\rho_T$, respectively.

Let $\mathsf{q}_t(y \,|\, x)$, $t \in [0, T]$, denote the transition density of a time-homogeneous reference Markov process $X^0$ (typically a diffusion), and let $\pi^0$ be the joint distribution of $(X_0^0, X_T^0)$ on $\mathbb{R}^d \times \mathbb{R}^d$ given by the initial distribution $\rho_0$ and the forward dynamics of the reference process. Then, in the Schrödinger bridge problem one looks for a coupling $\pi(x, y)$ between $\rho_0$ and $\rho_T$ that minimizes the Kullback-Leibler (KL) divergence

$$\inf_{\pi \in \Pi(\rho_0, \rho_T)} \mathsf{KL}(\pi, \pi^0), \tag{1}$$

where $\Pi(\rho_0, \rho_T)$ denotes the set of all couplings with the prescribed marginals. This marginal formulation of the Schrödinger problem avoids dealing with full path measures and is particularly amenable to sample-based estimation.

---

[*]Equal contribution

Under mild regularity conditions, the minimizer $\pi^*$ of the variational problem (1) admits a specific form (see Leonard (2014, Theorem 2.4)):

$$\pi^*(x,y) = \nu_0^*(x)\, \mathsf{q}_T(y \,|\, x)\, \nu_T^*(y), \tag{2}$$

where $\nu_0^*$ and $\nu_T^*$ are positive functions referred to as Schrödinger potentials. These scaling functions modify the reference transition structure so that the resulting coupling satisfies the marginal constraints. From this coupling, one can construct the corresponding Schrödinger bridge process $X^*$, which is a time-inhomogeneous Markov process with dynamics governed by a drift-modified version of the reference process.

Usually, researchers consider a reference process of the form

$$\mathrm{d}X_t^0 = \sigma\, \mathrm{d}W_t, \quad 0 \leqslant t \leqslant T, \tag{3}$$

where $W_t$ is a standard $d$-dimensional Wiener process and $\sigma > 0$. The motivation for such choice is that the problem (1) with the reference process (3) is equivalent to the entropic optimal transport problem

$$\inf_{\pi \in \Pi(\rho_0, \rho_T)} \int\limits_{\mathbb{R}^d} \int\limits_{\mathbb{R}^d} \frac{1}{2}\|x-y\|^2 \pi(x,y)\, \mathrm{d}x\, \mathrm{d}y + \sigma^2 T\, \mathsf{KL}(\pi, \rho_0 \otimes \rho_T),$$

which is studied quite well. However, there are two drawbacks. First, the entropic optimal transport formulation deals with the scalar parameter $\sigma$ and does not take into account possible anisotropy of the data. Second, if the reference dynamics obeys (3), then the correlation between $X_0^0$ and $X_T^0$ decays at slow polynomial rate. In the context of image-to-image translation, this could mean that initial images have an overabundant and possibly negative influence on the final ones. As a consequence, a learner has to take larger values of $\sigma$ or $T$ to mitigate this effect. In this work, we take the Ornstein-Uhlenbeck (OU) process as the reference dynamics. To be more specific, we will consider the base process $X_t^0$ solving the SDE

$$\mathrm{d}X_t^0 = b\left(m - X_t^0\right)\mathrm{d}t + \Sigma^{1/2}\mathrm{d}W_t, \quad 0 \leqslant t \leqslant T, \tag{4}$$

where $b > 0$ controls the drift rate, $m \in \mathbb{R}^d$ represents the mean-reversion level, and $\Sigma \in \mathbb{R}^{d \times d}$ is a positive definite symmetric matrix. Analytic tractability and exponential mixing properties of the OU dynamics make it especially suitable for both theoretical analysis and practical computation. The corresponding Schrödinger bridge process $X^*$ evolves under the modified drift

$$\mathrm{d}X_t^* = \beta^*(t, X_t^*)\mathrm{d}t + \Sigma^{1/2}\mathrm{d}W_t, \quad X_0^* \sim \rho_0, \tag{5}$$

where

$$\beta^*(t,x) = b(m-x) + \Sigma \nabla \log \nu_t^*(x), \tag{6}$$

and $\nu_t^*(x)$ denotes the time-evolved Schrödinger potential given by

$$\nu_t^*(x) = \int \nu_T^*(y)\, \mathsf{q}_{T-t}(y \,|\, x)\, \mathrm{d}y.$$

This yields a controlled diffusion that transitions from $\rho_0$ to $\rho_T$ in a way that remains close to the reference process. We also would like to note that the controlled diffusion (5) solves the stochastic optimal control problem (see Dai Pra (1991)).

The expression (2) for the optimal coupling $\pi^*$ suggests us to look for the solution of the problem (1) of the form

$$\pi^\varphi(x,y) = \nu_0^\varphi(x)\, \mathsf{q}_T(y \,|\, x)\, e^{\varphi(y)}, \quad \pi^\varphi \in \Pi(\rho_0, \rho_T),$$

where $\varphi$ belongs to a closed parametric class $\Phi$. The best log-potential $\varphi^\circ$ minimizes the Kullback-Leibler (KL) divergence between $\pi^*$ and $\pi^\varphi$:

$$\varphi^\circ \in \operatorname*{argmin}_{\varphi \in \Phi} \mathsf{KL}(\pi^*, \pi^\varphi), \quad \text{where} \quad \mathsf{KL}(\pi^*, \pi^\varphi) = \int\limits_{\mathbb{R}^d} \int\limits_{\mathbb{R}^d} \log \frac{\pi^*(x,y)}{\pi^\varphi(x,y)}\, \pi^\varphi(x,y)\, \mathrm{d}x\, \mathrm{d}y.$$

For any function $g : \mathbb{R}^d \to \mathbb{R}$ and any $t > 0$, let

$$\mathcal{T}_t[g](x) = \int\limits_{\mathbb{R}^d} g(y)\, \mathsf{q}_t(y \,|\, x)\, \mathrm{d}y \tag{7}$$

Table 1: Frequently used notations

| Notation | Eq. | Meaning |
|---|---|---|
| $\rho_0$ | | Source (initial) distribution |
| $\rho_T$ | | Target (terminal) distribution |
| $\mathsf{q}_t(y \mid x)$ | | Transition density of the OU process, density of $\mathcal{N}(m_t(x), \Sigma_t)$ |
| $m_t(x)$ | | $(1 - e^{-bt})m + e^{-bt}x$ |
| $\Sigma_t$ | | $(1 - e^{-2bt})\Sigma/(2b)$ |
| $\nu_0^*, \nu_T^*$ | (2) | Left and right Schrödinger potentials |
| $\varphi^*$ | (9) | Target Schrödinger log-potential |
| $\pi^*$ | (2), (9) | Target coupling, solution of the variational problem (1) |
| $\pi^\varphi$ | (8) | Coupling corresponding to the log-potential candidate $\varphi$ |
| $\widehat{\varphi}$ | (12) | Log-potential estimate, empirical risk minimizer |
| $\widehat{\pi}$ | (12) | Coupling estimate, corresponding to $\widehat{\varphi}$ |

stand for the Ornstein-Uhlenbeck operator applied to $g$. Let us note that the condition

$$\rho_0(x) = \int_{\mathbb{R}^d} \pi^\varphi(x, y) \, \mathrm{d}y$$

yields that $\nu_0^\varphi(x) = \rho_0(x)/\mathcal{T}_T[e^\varphi](x)$. For this reason, we can rewrite $\pi^\varphi$ in the following form:

$$\pi^\varphi(x, y) = \frac{\rho_0(x) \, \mathsf{q}_T(y \mid x) \, e^{\varphi(y)}}{\mathcal{T}_T[e^\varphi](x)}. \tag{8}$$

Similarly, the optimal coupling satisfies the equality

$$\pi^*(x, y) = \frac{\rho_0(x) \, \mathsf{q}_T(y \mid x) \, e^{\varphi^*(y)}}{\mathcal{T}_T[e^{\varphi^*}](x)}, \quad \varphi^* = \log(\nu_T^*). \tag{9}$$

Hence, we can represent the KL-divergence between $\pi^*$ and $\pi^\varphi$ as the difference

$$\mathsf{KL}(\pi^*, \pi^\varphi) = \mathcal{L}(\varphi) - \mathcal{L}(\varphi^*), \tag{10}$$

where

$$\mathcal{L}(\varphi) = \mathbb{E}_{Z \sim \rho_0} \log \mathcal{T}_T[e^\varphi](Z) - \mathbb{E}_{Y \sim \rho_T} \varphi(Y). \tag{11}$$

Our focus lies on statistical aspects of Schrödinger potential estimation. We assume that $\rho_0$ and $\rho_T$ are accessed through i.i.d. samples $Z_1, \ldots, Z_N$ and $Y_1, \ldots, Y_n$, respectively. In what follows, we assume that $N = n$ for simplicity. The objective $\mathcal{L}(\varphi)$ has a remarkable property that it includes only marginal distributions $\rho_0$ and $\rho_T$, rather than the coupling $\pi^*$. Hence, we do not have to make any presumptions about joint distribution of $Z_j$'s and $Y_i$'s.

A natural strategy to estimate $\pi^*$ and $\varphi^*$ is to replace the integrals in (11) by empirical means and consider the estimates

$$\widehat{\pi} = \pi^{\widehat{\varphi}} \quad \text{and} \quad \widehat{\varphi} \in \operatorname*{argmin}_{\varphi \in \Phi} \widehat{\mathcal{L}}(\varphi), \tag{12}$$

where

$$\widehat{\mathcal{L}}(\varphi) = \frac{1}{n} \sum_{j=1}^{n} \log \mathcal{T}_T[e^\varphi](Z_j) - \frac{1}{n} \sum_{i=1}^{n} \varphi(Y_i). \tag{13}$$

In the present paper, we are interested in generalization ability of the empirical risk minimizer $\widehat{\pi}$.

**Contribution.**

- The main contribution of this paper is a non-asymptotic high-probability generalization error bound for the empirical risk minimizer (12) (Theorem 1). Its proof relies on concentration arguments and ergodic properties of the OU process.

- When the approximation error is small, we establish nearly fast convergence rates (up to logarithmic factors) for the population KL risk. These results highlight the statistical efficiency of the Schrödinger bridge approach in the finite-sample regime.

- We conduct numerical experiments on synthetic and real data illustrating that a proper choice of the reference process can improve data generation quality.

**Notations.** Throughout the paper, $q_t(y \mid x)$ denotes the transition density of the OU process (4), which is Gaussian with mean $m_t(x) = m + e^{-bt}(x - m)$ and variance $\Sigma_t = (1 - e^{-2bt})\Sigma/(2b)$. Based on this, we define the Ornstein-Uhlenbeck operator

$$\mathcal{T}_t[g](x) = \int_{\mathbb{R}^d} g(y) \, q_t(y \mid x) \, \mathrm{d}y.$$

The notation $f \lesssim g$ or $g \gtrsim f$ means that $f = \mathcal{O}(g)$. Besides, we often replace $\max\{a, b\}$ and $\min\{a, b\}$ by shorter expressions $a \vee b$ and $a \wedge b$, respectively. For any $s \geqslant 1$, the Orlicz $\psi_s$-norm of a random variable $\xi \sim p$ is defined as $\|\xi\|_{\psi_s} = \inf\{u > 0 : \mathbb{E}e^{|\xi|^s/u^s} \leqslant 2\}$. Sometimes, we use the notation $\|\xi\|_{\psi_s(p)}$ to emphasize that $\xi$ is drawn according to $p$. Finally, given $p \geqslant 1$ and a probability density $\rho$, the weighted $L_p$-norm of a function $f$ is defined as $\|f\|_{L_p(\rho)} = (\mathbb{E}_{\xi \sim \rho}|f(\xi)|^p)^{1/p}$. Given two probability densities $p \ll q$, the Kullback-Leibler divergence between them is defined as $\mathsf{KL}(p, q) = \mathbb{E}_{\xi \sim p} \log(p(\xi)/q(\xi))$. Some other frequently used notations are collected in Table 1.

## 2 RELATED WORK

The empirical success of denoising diffusion models (Ho et al., 2020; Song et al., 2021) and flow matching (Lipman et al., 2023) made researchers study more general frameworks for mapping an initial distribution to a target one, such as stochastic interpolants (Albergo & Vanden-Eijnden, 2023; Albergo et al., 2025), optimal transport (Tong et al., 2020; 2024b;a) and Schrödinger bridges (De Bortoli et al., 2021; Shi et al., 2023; Liu et al., 2023). In the context of the Schrödinger bridge problem, a lot of attention was paid to development of solvers based on iterative proportional fitting (De Bortoli et al., 2021) and iterative Markovian fitting (Shi et al., 2023; Liu et al., 2023; Gushchin et al., 2024b). Some recent papers (Rapakoulias et al., 2025; Domingo-Enrich et al., 2025) suggest using stochastic optimal control theory (see Dai Pra (1991) for background) to construct Schrödinger bridges. Besides, a lot of efforts were made to construct efficient algorithms for solving the Schrödinger bridge problem and accelerate statistical inference (see, for instance, Tang et al. (2024); Gushchin et al. (2024a;b)). Despite numerous methodological advances in application of Schrödinger bridge framework to generative modelling, its theoretical justification is often quite challenging and largely missing. For instance, only recently Conforti et al. (2024) (see also Eckstein (2025)) extended results on geometric convergence of IPF from compactly supported $\rho_0$ and $\rho_T$ to strongly log-concave marginals with unbounded supports. We would like to emphasize that Conforti et al. (2024) consider an idealized setting ignoring sampling error. A comprehensive analysis of the statistical guarantees for the Schrödinger potential estimation is still to be done. We provide a detailed discussion for few contributions in this direction below.

In Rigollet & Stromme (2025), the authors considered the following empirical dual objective:

$$\widehat{S}(\nu_0, \nu_T) := \frac{1}{n}\sum_{j=1}^{n} \log \nu_0(Z_j) + \frac{1}{n}\sum_{i=1}^{n} \log \nu_T(Y_i) - \frac{\sigma^2}{n^2}\sum_{i=1}^{n}\sum_{j=1}^{n} \nu_0(Z_j)\, q_T(Y_i \mid Z_j)\, \nu_T(Y_i) + \sigma^2.$$

This objective arises as a variational representation of the entropic optimal transport problem, where the regularization term penalizes deviations from the empirical product measure $\rho_0^{\otimes n} \otimes \rho_T^{\otimes n}$, with regularization parameter $\sigma^2 > 0$. Note that since $\rho_0$ is fixed in our setting, the empirical KL objective (13) reduces to

$$\widehat{\mathcal{L}}(\nu_0, \nu_T) = \frac{1}{n}\sum_{j=1}^{n} \log \nu_0(Z_j) + \frac{1}{n}\sum_{i=1}^{n} \log \nu_T(Y_i),$$

which corresponds to the log-likelihood of the coupling $\pi(x, y) = \nu_0(x)\, q_T(y \mid x)\, \nu_T(y)$ evaluated on the data, up to an additive constant.

The dual objective $\widehat{S}$ thus coincides with $\widehat{\mathcal{L}}$ up to a constant when the marginal penalty term vanishes, i.e., when $\sigma^2 = 0$. More generally, we may write:

$$\widehat{S} = \widehat{\mathcal{L}} - \text{marginal penalty} + \text{constant}.$$

This formulation highlights the interpretation of $\varepsilon$ as a trade-off parameter controlling how strictly the marginal constraints are enforced: in the limit $\varepsilon \to 0$, the dual objective recovers the pure likelihood form. It was shown in (Rigollet & Stromme, 2025) that the empirical objective $\widehat{S}(\nu_0, \nu_T)$ converges to its population counterpart at the rate $\mathcal{O}(1/n)$ in expectation. However, it is important to note that no theoretical guarantees were provided for the convergence of the coupling-level Kullback-Leibler divergence $\mathrm{KL}(\pi^*, \widehat{\pi})$, where the estimated coupling is given by

$$\widehat{\pi}(x, y) = \widehat{\nu}_0(x)\, \mathsf{q}_T(y \mid x)\, \widehat{\nu}_T(y),$$

and $\pi^*$ denotes the true optimal coupling. As a result, while the dual potentials converge in the objective sense, this does not immediately imply convergence in distribution or divergence between the associated couplings.

An empirical objective similar to our formulation (13) was investigated in (Korotin et al., 2024). The authors studied the practical performance of the objective $\widehat{\mathcal{L}}$ and also established theoretical guarantees on its statistical behavior. In particular, they proved that $\widehat{\mathcal{L}}$ converges to its population counterpart $\mathcal{L}$ in expectation at the rate $\mathcal{O}(1/\sqrt{n})$ when the right Schrödinger potential is a Gaussian mixture. This result implies the convergence of the induced coupling $\widehat{\pi}(x, y)$ in expected KL divergence with the same rate. However, a simple numerical experiment indicates that a learner can hope for much more optimistic rates in the case of Gaussian marginals $\rho_0$ and $\rho_T$ (see Figure 1). In contrast, our work provides significantly stronger guarantees: we prove high-probability upper bounds on the KL divergence $\mathrm{KL}(\pi^*, \widehat{\pi})$, where $\pi^*$ denotes the optimal solution and the coupling $\widehat{\pi}$ is defined in (12). Our result holds for a general class $\Phi$ of log-potentials, including the class of logarithms of Gaussian mixtures considered in (Korotin et al., 2024). Under appropriate assumptions on the function class $\Phi$, we obtain a fast convergence rate of order $\mathcal{O}(\log^3(n)/n)$. This sharp control of the statistical error reflects the actual dependence of $\mathbb{E}\mathrm{KL}(\pi^*, \widehat{\pi})$ on the sample size $n$ in the case of Gaussian $\rho_0$ and $\rho_T$ much better (see Figure 1) and makes our approach particularly attractive for generative modelling and style transfer tasks with limited data.

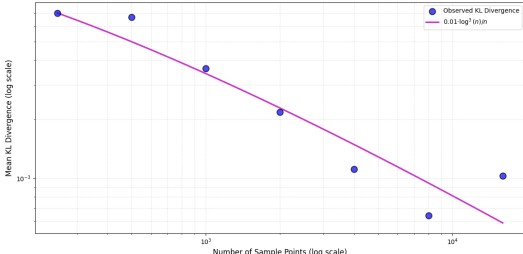

Figure 1: Dependence of $\mathbb{E}\mathrm{KL}(\pi^*, \widehat{\pi})$ on the sample size $n$ in the case of two-dimensional Gaussian marginals $\rho_0$ and $\rho_T$. The expected KL-divergence was estimated from 10 runs. The purple line was fitted with linear regression ($R^2 = 0.94$).

Let us also mention the work (Pooladian & Niles-Weed, 2025), where the authors show that the Schrödinger potentials obtained from solving the static entropic optimal transport problem between source and target samples can be modified to yield a natural plug-in estimator of the time-dependent drift defining the Schrödinger bridge between the two distributions. The authors show that, if both $\rho_0$ and $\rho_T$ are supported on compact $k$-dimensional submanifolds in $\mathbb{R}^d$, then it holds that

$$\mathbb{E}\mathrm{TV}^2(\mathsf{P}^*_{[0,\tau]}, \widehat{\mathsf{P}}_{[0,\tau]}) \lesssim \frac{1}{\sqrt{n}} + \frac{1}{(T-\tau)^{k+2}n}, \quad 0 < \tau < T. \tag{14}$$

Here $\mathsf{P}^*_{[0,\tau]}$ and $\widehat{\mathsf{P}}_{[0,\tau]}$ are the path measures corresponding to the optimally controlled process (5) and the process with estimated drift. Unfortunately, the right-hand side of (14) blows up as $\tau$ approaches $T$[1]. This highlights a fundamental challenge in learning Schrödinger bridges near the boundary of the time interval.

Let us stress that Korotin et al. (2024), Pooladian & Niles-Weed (2025), and Rigollet & Stromme (2025) restrict their attention to the case where the reference process is Brownian motion. While this

---

[1] Probably, this is due to singularity of the target distribution with respect to the Lebesgue measure in $\mathbb{R}^d$.

choice offers analytical convenience and aligns with the classical formulation of the Schrödinger problem, it limits modelling flexibility, especially in contexts where prior structure or time-inhomogeneity is important. In contrast, our framework accommodates more general reference processes, such as the Ornstein–Uhlenbeck process, enabling better control of prior dynamics and improved statistical regularity through mixing properties.

## 3 MAIN RESULT

In this section, we present a high-probability upper bound on the KL-divergence between the optimal coupling $\pi^*$ and the empirical risk minimizer $\widehat{\pi}$ defined in (12). It holds under mild assumptions listed below. First, as we mentioned in the introduction, we assume that the reference process $X^0$ obeys the Ornstein-Uhlenbeck dynamics.

**Assumption 1.** *The base process $X^0$ solves the following SDE*

$$\mathrm{d}X_t^0 = b\left(m - X_t^0\right)\mathrm{d}t + \Sigma^{1/2}\,\mathrm{d}W_t, \quad 0 \leqslant t \leqslant T.$$

*where $b > 0$, $m \in \mathbb{R}^d$, $\Sigma$ is a positive definite symmetric matrix of size $d \times d$, and $W$ is a d-dimensional Brownian motion.*

Second, we assume that the initial distribution has a bounded support.

**Assumption 2.** *The initial density $\rho_0$ has a bounded support and $\|\Sigma^{-1/2}(x - m)\| \leqslant R$ for some $R > 0$ and all $x \in \mathrm{supp}(\rho_0)$.*

We do not require $\rho_0$ to be bounded away from zero or infinity on $\mathrm{supp}(\rho_0)$. For this reason, Assumption 2 is satisfied for a large class of densities. It plays a crucial role in the proof of Lemma C.1, which helps us to verify a Bernstein-type condition (29). Extension of this result to the case of unbounded $\mathrm{supp}(\rho_0)$ will require significant efforts. A requirement on the target distribution is even milder, we just assume that it is sub-Gaussian.

**Assumption 3.** *The target distribution at time $T$ is absolutely continuous with the density $\rho_T$ and sub-Gaussian with variance proxy $\mathrm{v}^2$, that is,*

$$\mathbb{E}_{Y \sim \rho_T} e^{u^\top Y} \leqslant e^{\mathrm{v}^2 \|u\|^2/2} \quad \text{for any } u \in \mathbb{R}^d. \tag{15}$$

We would like to note that (15) yields that $\mathbb{E}_{Y \sim \rho_T} Y = 0$. This is just a normalization condition, which does not diminish generality of the setup we consider. In Korotin et al. (2024), the authors assumed that both $\rho_0$ and $\rho_T$ have compact supports. In contrast, we allow $\mathrm{supp}(\rho_T)$ to be unbounded. We proceed with two assumptions on properties of the log-potential $\varphi^*$ associated with the optimal coupling $\pi^*$ (see (9)) and the reference class $\Phi$.

**Assumption 4.** *There exist positive constants $L$ and $M$ such that every $\varphi \in \Phi \cup \{\varphi^*\}$ satisfies the inequalities*

$$-L\|\Sigma^{-1/2}(x - m)\|^2 - M \leqslant \varphi(x) \leqslant M \quad \text{for all } x \in \mathbb{R}^d.$$

*Moreover, for any $\varphi \in \Phi \cup \{\varphi^*\}$ it holds that $\mathcal{T}_\infty \varphi = 0$.*

We have to introduce the requirement $\mathcal{T}_\infty \varphi = 0$ for all $\varphi \in \Phi \cup \{\varphi^*\}$ because of the fact that Schrödinger log-potentials are defined up to an additive constant. Thus, the condition $\mathcal{T}_\infty \varphi = 0$ helps us to avoid ambiguity. Assumption 4, together with Assumptions 2 and 3, ensures that the random variables $\varphi(Y_i)$, $1 \leqslant i \leqslant n$, and $\log \mathcal{T}_T[e^\varphi](Z_j)$, $1 \leqslant j \leqslant n$, are sub-exponential. This allows us to quantify deviations of the empirical risk (13) from its expectation using concentration inequalities. In Puchkin et al. (2025), the authors provided several examples of reference classes satisfying Assumption 4, including classes of ReLU neural networks and classes of logarithms of Gaussian mixtures. We will use the last ones in Section 4 with numerical experiments. According to Korotin et al. (2024, Theorem 3.4), Gaussian mixtures possess a uniform approximation property. For this reason, a proper choice of a Gaussian mixtures class may lead to convincing practical results. Further discussion of Assumption 4 is deferred to Appendix A.

Finally, we assume that $\Phi$ is parametrized by a finite-dimensional parameter $\theta \in \mathbb{R}^D$: $\Phi = \{\varphi_\theta : \theta \in \Theta\}$, where $\Theta$ is a subset of a $D$-dimensional cube $[-1, 1]^D$ and each function $\varphi_\theta$ maps $\mathbb{R}^d$ onto $\mathbb{R}$. We suppose that the parametrization is sufficiently smooth in the following sense.

**Assumption 5.** *The class $\Phi$ is parametric: $\Phi = \{\varphi_\theta : \theta \in \Theta\}$, where $\Theta \subseteq [-1, 1]^D$. Moreover, there exists $\Lambda \geqslant 0$ such that*

$$|\varphi_\theta(x) - \varphi_{\theta'}(x)| \leqslant \Lambda \left(1 + \|x\|^2\right) \|\theta - \theta'\|_\infty \quad \textit{for all } \theta, \theta' \in \Theta \textit{ and all } x \in \mathbb{R}^d.$$

A similar condition appeared in Puchkin et al. (2025), where the authors argued that Assumption 5 holds for the aforementioned classes of neural networks with ReLU activations and classes of logarithms of Gaussian mixtures. We refer a reader to Puchkin et al. (2025, Section 4) for the details.

We are ready to formulate the main result of this section.

**Theorem 1.** *Grant Assumptions 1, 2, 3, 4, and 5. For any $n \in \mathbb{N}$ and $\delta \in (0, 1)$, introduce*

$$\Upsilon(n, \delta) = D(M + d) \left(1 \vee \frac{L}{b}\right) \left(M + \log\left((L \vee \Lambda)nd\right) + \log(1/\delta)\right) \frac{\log n}{n}.$$

*Then there exists*

$$T_0 \lesssim \frac{1}{b} \log\log \frac{1}{\Upsilon(n, 1)} \vee \frac{1}{b} \log\left(d + LR^2 + \frac{Ld^2}{b} + M\right) \lesssim \frac{\log\log n}{b}$$

*such that for any $\delta \in (0, 1)$ and any $T \geqslant T_0$, with probability at least $(1 - \delta)$, the coupling $\widehat{\pi}$ defined in (12) satisfies the inequality*

$$\mathsf{KL}(\pi^*, \widehat{\pi}) - \Delta \lesssim \sqrt{\Upsilon(n, \delta)\Delta \left(1 \vee \log \frac{(b \vee L)(M + d)}{b\Delta}\right)}$$

$$+ \Upsilon(n, \delta) \left(1 \vee \log \frac{(b \vee L)(M + d)}{b\Delta}\right), \tag{16}$$

*where $\Delta = \inf\limits_{\varphi \in \Phi} \mathsf{KL}(\pi^*, \pi^\varphi)$.*

**Remark 2.** *A reader can note that the quantity $\Upsilon(n, \delta)$ introduced in Theorem 1 scales as*

$$\Upsilon(n, \delta) \lesssim \left(\log n + \log d + \log(1/\delta)\right) \frac{Dd \log n}{n}.$$

*If the class $\Phi$ is rich enough in a sense that $\Delta \lesssim \Upsilon(n, \delta)$, then the upper bound (16) simplifies to*

$$\mathsf{KL}(\pi^*, \widehat{\pi}) \lesssim \Upsilon(n, \delta) \log n \lesssim \left(\log n + \log d + \log(1/\delta)\right) \frac{Dd \log^2 n}{n}.$$

To our knowledge, Theorem 1 provides the first non-asymptotic high-probability upper bound on the KL-divergence between the optimal coupling $\pi^*$ and the empirical risk minimizer $\widehat{\pi}$. The previous result of Korotin et al. (2024) (see Theorem A.1 in their paper) concerned only the expected excess risk $\mathbb{E}\mathsf{KL}(\pi^*, \widehat{\pi}) - \Delta$. We also would like to note that our rate of convergence is sharper than the $\mathcal{O}(1/\sqrt{n})$ bound of Korotin et al. (2024) when the approximation error $\Delta$ is small. This is the case when there exists $\varphi \in \Phi$ approximating $\varphi^*$ with a good accuracy with respect to $L_1(\rho_T)$- and $L_1(\mathcal{N}(\mu, \Sigma))$-norms. Indeed, according to Lemma E.2 and the representation (10), we have

$$\mathsf{KL}(\pi^*, \pi^\varphi) \lesssim \|\varphi - \varphi^*\|_{L_1(\rho_T)} + (\mathcal{T}_\infty|\varphi - \varphi^*|)^{1 + \mathcal{O}(\sqrt{d}e^{-bT})}.$$

Quantitative bounds on $\|\varphi - \varphi^*\|_{L_1(\rho_T)}$ and $\mathcal{T}_\infty|\varphi - \varphi^*|$ require the log-potential $\varphi^*$ to belong to a class of smooth functions, like Hölder or Sobolev classes. Unfortunately, we are not aware of results of this kind. For this reason, we leave the study of the approximation error out of the scope of the present paper. However, we can observe that in a favourable scenario, when $\Delta \lesssim \Upsilon(n, \delta)$, the rate of convergence in Theorem 1 becomes $\mathcal{O}(\log^3(n)/n)$, which is much better than $\mathcal{O}(1/\sqrt{n})$. Besides, Theorem 1 exhibits a nearly linear dependence $\mathcal{O}(d \log d)$ on the dimension $d$, which makes the result suitable for high-dimensional settings.

Theorem 1 requires $bT$ to satisfy the bound $bT \gtrsim \log\log n$. It will become clear from Section B.1 that it plays an essential role in the proof of Theorem 1 and is necessary for verification of a Bernstein-type condition. The assumption $bT \gtrsim \log\log n$ seems to be necessary for the fast rates,

because of the loss function (see (13)). In the limiting case $T = 0$ the loss becomes linear in $\varphi$ and has no curvature. Nevertheless, we would like to emphasize the following. First, the inequality $bT \gtrsim \log\log n$ should be considered as a condition on $bT$, rather than on the time horizon $T$ alone. This means that we can take $T = 1$ (as it is usually done in Schrödinger bridge problems) and $b \gtrsim \log\log n$. Second, a violation of the condition $bT \gtrsim \log\log n$ does not mean that the ERM $\widehat{\pi}$ becomes inconsistent. Using the standard approach based on Rademacher complexities, one can show that the KL-divergence between $\pi^*$ and $\widehat{\pi}$ will still tend to zero but at a slower rate $\mathcal{O}(1/\sqrt{n})$. Finally, the OU process is exponentially ergodic while the condition on $bT$ exhibits the doubly logarithmic dependence. Hence, there is still a significant correlation between the initial and final points of the SDE-governed stochastic process. In the context of image-to-image translation, this means that the style of initial images will be successfully transferred to the final ones.

In generative diffusion models one is often interested in accuracy of the optimal drift estimation. According to Dai Pra (1991, Theorem 3.2), the optimally controlled process $X_t^*, 0 \leqslant t \leqslant T$, obeys the SDE

$$X_0^* \sim \rho_0, \quad \mathrm{d}X_t^* = -b(X_t^* - m)\mathrm{d}t + u^*(X_t^*, t)\mathrm{d}t + \Sigma^{1/2}\mathrm{d}W_t, \quad 0 < t < T, \qquad (17)$$

where $u^*(x, t) = \Sigma\nabla\log\mathcal{T}_{T-t}[e^{\varphi^*}](x)$. With the estimate $\widehat{\varphi}$ at hand (see (12) for the definition), we can consider the corresponding process

$$\widehat{X}_0 \sim \rho_0, \quad \mathrm{d}\widehat{X}_t = -b(\widehat{X}_t - m)\mathrm{d}t + \widehat{u}(\widehat{X}_t, t)\mathrm{d}t + \Sigma^{1/2}\mathrm{d}W_t, \quad 0 < t < T, \qquad (18)$$

with $\widehat{u}(x, t) = \Sigma\nabla\log\mathcal{T}_{T-t}[e^{\widehat{\varphi}}](x)$. This brings us to a natural question whether $\widehat{u}(x, t)$ is close $u^*(x, t)$. According to Girsanov's theorem (see, for example, Domingo-Enrich et al. (2025, Theorem 2 and eq. (138))) it holds that

$$\mathbb{E}_{X^*}\int_0^T \left\|\Sigma^{-1/2}\bigl(\widehat{u}(X_t^*, t) - u^*(X_t^*, t)\bigr)\right\|^2 \mathrm{d}t = 2\,\mathsf{KL}(\pi^*, \widehat{\pi}).$$

The expectation in the left-hand side is taken with respect to the trajectory of $X_t^*, 0 < t < T$, conditioned on data $Y_1, \ldots, Y_n$ and $Z_1, \ldots, Z_n$. Hence, Theorem 1 also establishes a high-probability upper bound on the control estimation.

## 4  NUMERICAL EXPERIMENTS

This section is devoted to numerical experiments illustrating the advantage of the Ornstein-Uhlenbeck dynamics (4) as the reference process. For this purpose, we adopt ideas behind the light Schrödinger bridge (LightSB) algorithm from Korotin et al. (2024), where the authors suggested a simple yet quite efficient approach for generative modelling. We do not pursue the goal of developing a state-of-the-art method, it goes beyond the scope of the present paper. However, we would like to illustrate how a proper choice of the base process improves data generation. Throughout this section, we use the notations $\sigma_t^2 = (1 - e^{-2bt})/(2b)$ and $m_t(x) = e^{-bt}x + (1 - e^{-bt})m, 0 \leqslant t \leqslant T$. In what follows, we consider the process (4) with $T = 1$ and $\Sigma = \varepsilon I_d$, where $\varepsilon > 0$. The light Schrödinger bridge considers a reference class of log-potentials $\Phi = \{\varphi_\theta : \theta \in \Theta\}$, such that the functions $v_\theta(y) = e^{\varphi_\theta(y) - \|y\|^2/(2\varepsilon\sigma_1^2)}$, referred to as adjusted potentials, are nothing but Gaussian mixtures:

$$v_\theta(y) = \sum_{k=1}^K \alpha_k\, \mathsf{p}(y; r_k, \varepsilon\sigma_1^2 S_k). \qquad (19)$$

Here, for any $k \in \{1, \ldots, K\}$, $\mathsf{p}(\cdot; r_k, \varepsilon\sigma_1^2 S_k)$ is the density of the Gaussian distribution with mean $r_k$ and covariance $\varepsilon\sigma_1^2 S_k$. In this case, the parameter $\theta = \{(\alpha_k, r_k, S_k) : 1 \leqslant k \leqslant K\}$ is just the set of triplets. The main advantage of such parametrization is that the Ornstein-Uhlenbeck transform $\mathcal{T}_1[e^{\varphi_\theta}](x)$ admits a closed-form expression. This significantly saves computational resources. At the same time, according to Korotin et al. (2024, Theorem 3.4), choosing a proper class of Gaussian mixtures, one can approximate quite complex potentials. With the described $\varphi_\theta$, the corresponding coupling $\pi^{\varphi_\theta}$ and the risk $\mathcal{L}(\varphi_\theta)$ the (see (8) and (11), respectively) reduce to

$$\pi^{\varphi_\theta}(x, y) = \rho_0(x)\, e^{y^\top m_1(x)/(\varepsilon\sigma_1^2)} v_\theta(y)/c_\theta(x) \qquad (20)$$

and $\mathcal{L}(\varphi_\theta) = \mathbb{E}_{Z\sim\rho_0} \log c_\theta(Z) - \mathbb{E}_{Y\sim\rho_T} \log v_\theta(Y)$, where

$$c_\theta(x) = \int_{\mathbb{R}^d} e^{y^\top m_1(x)/(\varepsilon\sigma_1^2)} v_\theta(y)\, \mathrm{d}y = \sum_{k=1}^{K} \widetilde{\alpha}_k(x) \tag{21}$$

and

$$\widetilde{\alpha}_k(x) = \alpha_k \exp\left\{ \frac{r_k^\top m_1(x)}{\varepsilon\sigma_1^2} + \frac{\|S_k^{1/2} m_1(x)\|^2}{2\varepsilon\sigma_1^2} \right\}. \tag{22}$$

The log-potential estimate $\widehat{\varphi}$ is constructed through minimization of the empirical risk

$$\widehat{\mathcal{L}}(\varphi_\theta) = \frac{1}{N} \sum_{j=1}^{N} \log c_\theta(Z_j) - \frac{1}{n} \sum_{i=1}^{n} \log v_\theta(Y_i)$$

over all admissible values of $\theta$. We call the described algorithm LightSB-OU and provide its pseudocode in Appendix F.2. The standard LightSB algorithm (Korotin et al., 2024) performs the same steps but replaces $q_1(y\,|\,x)$ with the Gaussian kernel $(2\pi\varepsilon)^{-d/2} e^{-\|y-x\|^2/(2\varepsilon)}$ corresponding to the transition density of the scaled Wiener process (3) with $\sigma = \sqrt{\varepsilon}$.

Below we thoroughly evaluate our solver on setups with both synthetic (Section 4.1) and real data sets (Sections 4.2 and 4.3). The method demonstrates a stable increase in quality compared to the original LightSB and shows itself well in comparison with other Schrödinger bridge solvers. The source code is available at GitHub[2].

## 4.1 EVALUATION ON THE GAUSSIAN MIXTURE

We start with evaluation of the modified LightSB algorithm on a Gaussian mixture problem. For this experiment, we used mixtures of 25 Gaussians under three configurations: uniform grid with standard covariance matrices, random grid locations with standard covariance matrices, uniform grid with anisotropic random covariance matrices, with the sliced approximation of the Wasserstein distance $\mathbb{W}_1$ as the metric, see Appendix F.2 for definition of metrics.

Table 2 summarizes the performance for $K = 30$ potential approximation components. With a sufficient number of potential approximation components, our approach demonstrates consistent improvements in both $\mathbb{W}_1$, MMD and Mode Coverage (see Appendix F.3 for detailed plots and Appendix F.6 for hyperparameters values used).

Table 2: Comparison of the LightSB and LightSB-OU performance for $K = 30$ across experiments with Gaussian mixtures with 25 components

| Experiment | Metric | LightSB | **LightSB-OU** |
|---|---|---|---|
| Standard | Sliced $\mathbb{W}_1$ | $0.260 \pm 0.016$ | $\mathbf{0.156 \pm 0.018}$ |
| | MMD | $0.0004 \pm 0.0001$ | $\mathbf{0.0003 \pm 0.0001}$ |
| | Mode Coverage | $24.2 \pm 0.4$ | $\mathbf{25.0 \pm 0.0}$ |
| Irregular | Sliced $\mathbb{W}_1$ | $0.525 \pm 0.024$ | $\mathbf{0.214 \pm 0.028}$ |
| | MMD | $0.0024 \pm 0.0003$ | $\mathbf{0.0004 \pm 0.0001}$ |
| | Mode Coverage | $21.8 \pm 0.4$ | $\mathbf{25.0 \pm 0.0}$ |
| Anisotropic | Sliced $\mathbb{W}_1$ | $0.255 \pm 0.017$ | $\mathbf{0.206 \pm 0.024}$ |
| | MMD | $0.0007 \pm 0.0001$ | $\mathbf{0.0006 \pm 0.0002}$ |
| | Mode Coverage | $24.4 \pm 0.5$ | $\mathbf{25.0 \pm 0.0}$ |

Table 3: The quality of intermediate distribution restoration for the single-cell data

| Solver | $\mathbb{W}_1$ metric |
|---|---|
| OT-CFM (Tong et al., 2024a) | $0.790 \pm 0.068$ |
| [SF]$^2$M-Exact (Tong et al., 2024b) | $0.793 \pm 0.066$ |
| **LightSB-OU** (ours) | $\mathbf{0.812 \pm 0.020}$ |
| LightSB (Korotin et al., 2024) | $0.823 \pm 0.017$ |
| Reg. CNF (Finlay et al., 2020) | $0.825 \pm$ N/A[a] |
| T. Net (Tong et al., 2020) | $0.848 \pm$ N/A[a] |
| DSB (De Bortoli et al., 2021) | $0.862 \pm 0.023$ |
| I-CFM (Tong et al., 2024a) | $0.872 \pm 0.087$ |
| [SF]$^2$M-Geo (Tong et al., 2024b) | $0.879 \pm 0.148$ |
| NLSB (Koshizuka & Sato, 2023) | $0.970 \pm$ N/A[a] |
| [SF]$^2$M-Sink (Tong et al., 2024b) | $1.198 \pm 0.342$ |
| SB-CFM (Tong et al., 2024a) | $1.221 \pm 0.380$ |
| DSBM (Shi et al., 2023) | $1.775 \pm 0.429$ |

[a]The authors did not report the standard deviation.

[2]https://github.com/denvar15/Tight-Bounds-for-Schrodinger-Potential-Estimation-in-Unpaired-Data-Translation

### 4.2 Evaluation on the Single Cell Data

To comprehensively evaluate LightSB-OU, we conducted experiments on biological data. Following Tong et al. (2020), we considered the problem of transporting a cell distribution from time $t_{i-1}$ to time $t_{i+1}$ for $i \in \{1, 2, 3\}$. The results of Tong et al. (2024b) allow us to compare the performance of LightSB-OU not only with LightSB but also with many other methods (see Table 3). We predicted the cell distribution at the intermediate time $t_i$ and computed the Wasserstein distance $\mathbb{W}_1$ between the predicted distribution and the ground truth. The results were averaged across three setups ($i = 1, 2, 3$). To ensure statistical robustness, we repeated the experiment five times. Using the default LightSB parameters from the original paper (Korotin et al., 2024), the baseline achieved $\mathbb{W}_1 = 0.823 \pm 0.017$ and was outperformed by our OU-modified version ($\mathbb{W}_1 = 0.810 \pm 0.020$). The hyperparameter values of LightSB-OU are reported in Appendix F.6.

### 4.3 Unpaired Image-to-Image Translation

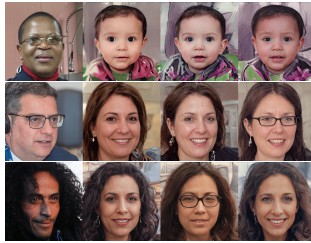 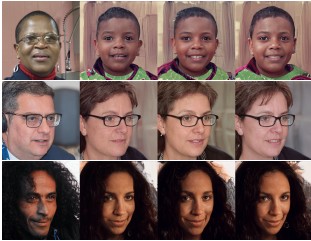

(a) LightSB: FID = 24.1        (b) LightSB-OU: FID = 24.0

Figure 2: Translation results from the latent space of ALAE using LightSB (left) and LightSB-OU(right). Top: Adult to Child – skin color and face shape preserved. Middle: Male to Female – glasses and face shape are preserved. Bottom: Male to Female – skin color and face shape preserved.

To further illustrate the importance of the reference process for data generation, we consider a task of unpaired image-to-image translation. Following the experimental setup of Korotin et al. (2024), we employ the ALAE autoencoder (Pidhorskyi et al., 2020) and focus on the Adult-to-Child translation task using the full $1024 \times 1024$ FFHQ dataset (Karras et al., 2019).

Below, we present comparative translation results for selected samples in Figure 2. While the baseline LightSB successfully performs the translation, it sometimes fails to preserve key attributes of the source distribution such as skin tone, eye color, and facial structure leading to noticeable deviations. In contrast, LightSB-OU achieves more accurate translations while better retaining the distinctive features of the original distribution. We emphasize that these examples are not intended to suggest universal superiority but rather to showcase specific strengths of our method in certain cases (the hyperparameter values are specified in Appendix F.6).

## 5 Conclusion

In the present paper, we established that the empirical risk minimizer $\hat{\pi}$ defined in (12) satisfies a Bernstein-type bound (see Theorem 1). This is a consequence of exponential ergodicity of the OU process and of some properties of the empirical risk (13) exhibiting a kind of curvature when $bT \gtrsim \log \log n$. The importance of the reference process was also illustrated with numerical experiments on artificial and real-world data. Further research may include an extension of Theorem 1 to the case when $\rho_0$ has an unbounded support and to heavy-tailed distributions $\rho_T$. It is also of particular interest whether the result of Theorem 1 holds if one considers a larger class of exponentially ergodic reference processes.

REPRODUCIBILITY STATEMENT

The provide the proof of our main result, Theorem 1, in Appendix B. In Appendix B.1, we discuss some insights behind the proof of Theorem 1 and then perform rigorous derivations in Appendix B.2. The proofs of all the auxiliary statements are collected in Appendix.

In Appendix F, we provide additional details for the numerical experiments described in Section 4 including the pseudocode of the LightSB-OU algorithm (see Algorithm 1). The code for the numerical experiments is available in the supplementary material and at GitHub[3].

We have not used large language models (LLMs) for retrieval, discovery or research ideation. All the mathematical derivations and numerical experiments were performed solely by the authors. We used LLMs just to polish writing.

ACKNOWLEDGEMENTS

The work was supported by the grant for research centers in the field of AI provided by the Ministry of Economic Development of the Russian Federation in accordance with the agreement 000000C313925P4E0002 and the agreement with HSE University №139-15-2025-009.

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

## A   ON ASSUMPTION 4

In Section 3, we suppose that the log-potential $\varphi^*$ associated with the optimal coupling $\pi^*$ meets Assumption 4. While the requirement $\varphi^*(x) \leqslant M$ is mild, the lower bound $\varphi^*(x) \geqslant -L\|\Sigma^{-1/2}(x - m)\|^2 - M$ needs a discussion. We are going to provide a sufficient condition for Assumption 4 to hold. Let us define $\varphi^*_{\max} = \sup_{x \in \mathbb{R}^d} \varphi^*(x) < +\infty$. In view of (9), we have

$$\rho_T(y) = \int_{\mathbb{R}^d} \pi^*(x, y)\, \mathrm{d}x = \int_{\mathbb{R}^d} \frac{\mathsf{q}_T(y\,|\,x)\, e^{\varphi^*(y)}}{\mathcal{T}_T[e^{\varphi^*}](x)}\, \rho_0(x)\, \mathrm{d}x.$$

This yields that

$$\varphi^*(y) = \log \rho_T(y) - \log \int_{\mathbb{R}^d} \frac{\mathsf{q}_T(y\,|\,x)}{\mathcal{T}_T[e^{\varphi^*}](x)}\, \rho_0(x)\, \mathrm{d}x.$$

Using Lemma B.3 from (Puchkin et al., 2025) and taking into account $\varphi^*(x) \leqslant \varphi^*_{\max}$, we obtain that

$$\log \frac{1}{\mathcal{T}_T[e^{\varphi^*}](x)} \lesssim \left( \frac{\|\Sigma^{-1/2}(x - m)\|^2}{\sqrt{d}} + (\varphi^*_{\max} \vee 1)\sqrt{d} \right) e^{-bT},$$

for all $x \in \mathrm{supp}(\rho_0)$. Since, $\log \mathsf{q}_T(y\,|\,x) \lesssim d$ for all $y, x \in \mathbb{R}^d$, it holds that

$$-\varphi^*(y) \lesssim -\log \rho_T(y) + d + \left( \frac{R^2}{\sqrt{d}} + (\varphi^*_{\max} \vee 1)\sqrt{d} \right) e^{-bT}.$$

In the last line, we used Assumption 2 about the support of $\rho_0$. Hence, we observe that, if

$$-\log \rho_T(y) \lesssim \|\Sigma^{-1/2}(y - m)\|^2 + d, \tag{23}$$

then $\varphi^*$ satisfies the inequality

$$-\varphi^*(y) \lesssim \|\Sigma^{-1/2}(y - m)\|^2 + d + \left( \frac{R^2}{\sqrt{d}} + (\varphi^*_{\max} \vee 1)\sqrt{d} \right) e^{-bT}$$

and, as a consequence, fulfils Assumption 4. The inequality (23) is satisfied, for instance, for a nonparametric Gaussian mixture model

$$\rho_T(y) = (\sqrt{2\pi}\sigma)^{-d} \int_{[0,1]^p} e^{-\|y - g(u)\|^2/(2\sigma^2)}\, \mathrm{d}u$$

recently considered by Yakovlev & Puchkin (2025) in the context of generative diffusion models. Here $g : [0,1]^p \to \mathbb{R}^d$ is a smooth map defined on a low-dimensional latent space. In (Yakovlev & Puchkin, 2025), the authors discussed that such a model is quite general and suitable for description of high-dimensional data occupying a vicinity of a smooth low-dimensional manifold.

## B    PROOF OF THEOREM 1

### B.1    PROOF INSIGHTS

Finally, we would like to discuss some intuition behind the proof of Theorem 1. Rigorous derivations are deferred to Appendix B. As we discussed in Section 3, Assumptions 2, 3, and 4 allow us to use concentration inequalities for sub-exponential random variables and quantify the deviation of the empirical risks $\widehat{\mathcal{L}}(\varphi)$, $\varphi \in \Phi$, from its expectations. However, to get rates of convergence possibly sharper than $\mathcal{O}(1/\sqrt{n})$, we must verify a Bernstein-type condition. That is, we have to show that both

$$\mathrm{Var}\big(\varphi^*(Y_1) - \varphi(Y_1)\big) \quad \text{and} \quad \mathrm{Var}\left(\log \frac{\mathcal{T}_T[e^\varphi](Z_1)}{\mathcal{T}_T[e^{\varphi^*}](Z_1)}\right)$$

can be controlled by a subquadratic function of $\mathsf{KL}(\pi^*, \pi^\varphi)$. This is the most challenging part of the proof. We would like to note that in Puchkin et al. (2025), the authors used properties of log-densities with subquadratic growth for a similar purpose (see their Lemmata C.2 and C.3). Unfortunately, we cannot apply them in our setup, because the joint distribution of $Y_i$'s and $Z_j$'s may not follow $\pi^*$. In other words, we have access to samples from the marginals of $\pi^*$ but not from $\pi^*$ itself. This makes us to use another strategy. Let us observe that for any $\varphi \in \Phi$

$$\mathsf{KL}(\pi^*, \pi^\varphi) = \mathbb{E}\big(\varphi^*(X_T^*) - \varphi(X_T^*)\big) + \mathbb{E}\log \mathbb{E}\left[e^{\varphi(X_T^*) - \varphi^*(X_T^*)} \,\big|\, X_0\right],$$

where $X_T^*$ is the endpoint of the Schrödinger bridge process defined in (5). Due to the tower rule, $\mathsf{KL}(\pi^*, \pi^\varphi)$ can be considered as expectation of the conditional cumulant generating function

$$G(\lambda, X_0) = \mathbb{E}\log \mathbb{E}\left[e^{\lambda(\varphi(X_T^*) - \varphi^*(X_T^*))} \,\big|\, X_0\right] - \lambda \mathbb{E}\left[\varphi(X_T^*) - \varphi^*(X_T^*) \,\big|\, X_0\right]$$

at $\lambda = 1$, that is, $\mathsf{KL}(\pi^*, \pi^\varphi) = \mathbb{E}_{X_0 \sim \rho_0} G(1, X_0)$. In Appendix B.3, we prove the following general result relating variance and the cumulant generating function of a sub-exponential random variable.

**Lemma B.1.** *Assume that a random variable $\xi$ has a finite $\psi_1$-norm. Then, for any $\lambda$ satisfying the inequality*

$$4e|\lambda| \left(5 + \log \frac{52\|\xi - \mathbb{E}\xi\|_{\psi_1}^2}{\mathrm{Var}(\xi)}\right) \|\xi - \mathbb{E}\xi\|_{\psi_1} \leqslant 1, \tag{24}$$

*it holds that $\mathrm{Var}(\xi) \leqslant 4\lambda^{-2} \log \mathbb{E}e^{\lambda(\xi - \mathbb{E}\xi)}$.*

Using this lemma, we show that (see Lemma C.1)

$$\mathbb{E}_{X_0 \sim \rho_0} \mathrm{Var}\big(\varphi(X_T^*) - \varphi^*(X_T^*) \,|\, X_0\big) \lesssim \varkappa \,\mathsf{KL}\,(\pi^*, \pi^\varphi) \left(1 \vee \log \frac{\varkappa}{\mathsf{KL}\,(\pi^*, \pi^\varphi)}\right),$$

where $\varkappa \lesssim (1 \vee L/b)(M + d)$. However, this bound is not enough, because $\mathrm{Var}\big(\varphi^*(Y_1) - \varphi(Y_1)\big)$ is always not smaller than the conditional variance. Fortunately, we can use properties of the Ornstein-Uhlenbeck process to derive the following result.

**Lemma B.2.** *Under the assumptions of Theorem 1, for any $\omega > 0$ there exists*

$$T_0 \lesssim \frac{1}{b}\left(\log\log\frac{1}{\omega} \vee \log\left(d + LR^2 + \frac{Ld^2}{b} + M\right)\right)$$

*such that for any $T \geqslant T_0$ and any $\varphi \in \Phi$ satisfying the inequality $\mathrm{Var}\big(\varphi(X_T^*) - \varphi^*(X_T^*)\big) \geqslant \omega$ it holds that*

$$\mathrm{Var}\left(\mathbb{E}\left[\varphi(X_T^*) - \varphi^*(X_T^*) \,\big|\, X_0\right]\right) \leqslant \frac{1}{2}\mathrm{Var}\big(\varphi(X_T^*) - \varphi^*(X_T^*)\big)$$
$$\leqslant \mathbb{E}\mathrm{Var}\big(\varphi(X_T^*) - \varphi^*(X_T^*) \,|\, X_0\big).$$

The proof of Lemma B.2 is postponed to Appendix B.4. It immediately yields that if $T$ satisfies the conditions of Theorem 1, then either the variance is sufficiently small, so that $\mathrm{Var}\big(\varphi(X_T^*) - \varphi^*(X_T^*)\big) \leqslant \Upsilon(n, \delta)$, or

$$\mathrm{Var}\big(\varphi(X_T^*) - \varphi^*(X_T^*)\big) \lesssim \varkappa \,\mathsf{KL}\,(\pi^*, \pi^\varphi) \left(1 \vee \log \frac{\varkappa}{\mathsf{KL}\,(\pi^*, \pi^\varphi)}\right)$$

as desired. On the other hand, according to Lemma C.4, it holds that

$$\mathrm{Var}\left(\log \frac{\mathcal{T}_T[e^\varphi](Z_1)}{\mathcal{T}_T[e^{\varphi^*}](Z_1)}\right) \lesssim \mathrm{Var}\big(\varphi(X_T^*) - \varphi^*(X_T^*)\big) + \varkappa\,\mathsf{KL}\,(\pi^*, \pi^\varphi)\left(1 \vee \log \frac{\varkappa}{\mathsf{KL}\,(\pi^*, \pi^\varphi)}\right).$$

This reveals a dichotomy. Under the assumptions of Theorem 1, if for some $\varphi \in \Phi$ the variance $\mathrm{Var}\big(\varphi^*(Y_1) - \varphi(Y_1)\big)$ is of order $\mathcal{O}(\Upsilon(n, \delta))$, then the corresponding difference $\widehat{\mathcal{L}}(\varphi) - \mathcal{L}(\varphi)$, is of order $\mathcal{O}(\Upsilon(n, \delta))$ as well. Otherwise, we have a Bernstein-type condition allowing us to derive faster rates of convergence using standard techniques from learning theory.

### B.2 RIGOROUS PROOF

The proof of Theorem 1 consists of three major steps. On the first one, we establish uniform concentration bounds for the sample means

$$\frac{1}{n}\sum_{i=1}^n (\varphi(Y_i) - \varphi^*(Y_i)) \quad \text{and} \quad \frac{1}{n}\sum_{j=1}^n \log \frac{\mathcal{T}_T[e^\varphi](Z_j)}{\mathcal{T}_T[e^{\varphi^*}](Z_j)}$$

around their expectations. Then we show that the loss functions of interest satisfy a Bernstein-type condition. Finally, we convert the results of the first two steps into the final large deviation bound.

**Step 1: uniform concentration.** To start with, we would like to note that Assumptions 3 and 4 imply that (see Lemma D.1)

$$\|\varphi^*(X_T^*) - \varphi(X_T^*)\|_{\psi_1} \lesssim \left(1 \vee \frac{L}{b}\right)(M + d).$$

On the other hand, according to Lemma D.2, under Assumptions 2 and 4, we have

$$\left\|\log \frac{\mathcal{T}_T[e^\varphi](X_0)}{\mathcal{T}_T[e^{\varphi^*}](X_0)}\right\|_{\psi_1} \lesssim M$$

for all $\varphi \in \Phi$. This means that we can exploit properties of sub-exponential random variables (including concentration bounds) during analysis of the empirical risk minimizer $\widehat{\varphi}$. In particular, Lemma E.1 states that, under the assumptions of Theorem 1, there exists

$$T_0 \lesssim \frac{1}{b}\log\left(d + LR^2 + \frac{Ld^2}{b} + M\right)$$

such that for any $\delta \in (0, 1)$ and any $T \geqslant T_0$ we have

$$\left|\frac{1}{n}\sum_{i=1}^n (\varphi(Y_i) - \varphi^*(Y_i)) - \mathbb{E}\big(\varphi^*(Y_1) - \varphi(Y_1)\big)\right|$$

$$\lesssim \sqrt{\frac{\mathrm{Var}\big(\varphi^*(Y_1) - \varphi(Y_1)\big)\big(D\log(\Lambda nd) + \log(1/\delta)\big)}{n}}$$

$$+ \left(1 \vee \frac{L}{b}\right)\frac{(M + d)\big(D\log(\Lambda nd) + \log(1/\delta)\big)\log n}{n}$$

and

$$\left|\frac{1}{n}\sum_{j=1}^n \left(\log \frac{\mathcal{T}_T[e^\varphi](Z_j)}{\mathcal{T}_T[e^{\varphi^*}](Z_j)} - \mathbb{E}\log \frac{\mathcal{T}_T[e^\varphi](Z_1)}{\mathcal{T}_T[e^{\varphi^*}](Z_1)}\right)\right|$$

$$\lesssim \sqrt{\mathrm{Var}\left(\log \frac{\mathcal{T}_T[e^\varphi](Z_1)}{\mathcal{T}_T[e^{\varphi^*}](Z_1)}\right)\frac{\big(D\log(\Lambda nd) + \log(1/\delta)\big)}{n}}$$

$$+ \frac{(M + 1)\big(D\log(\Lambda nd) + \log(1/\delta)\big)\log n}{n}$$

with probability at least $(1 - \delta)$ simultaneously for all $\varphi \in \Phi$. We provide the proof of Lemma E.1 in Appendix E.1. It relies on Bernstein's inequality for random variables with finite $\psi_1$-norms (see, for example, (Lecué & Mitchell, 2012, Proposition 5.2)) and the $\varepsilon$-net argument. This approach is quite standard. A similar strategy was used by Puchkin et al. (2025).

$$\Upsilon(n, \delta) = D(M + d) \left(1 \vee \frac{L}{b}\right) \left(M + \log\big((L \vee \Lambda)nd\big) + \log(1/\delta)\right) \frac{\log n}{n}$$

for brevity, we can rewrite the uniform concentration bounds in the following form: for any $\delta \in (0, 1)$, with probability at least $(1 - \delta)$ it holds that

$$\left| \frac{1}{n} \sum_{i=1}^{n} \big(\varphi(Y_i) - \varphi(Y_i)\big) - \mathbb{E}\big(\varphi(Y_1) - \varphi(Y_1)\big) \right|$$
$$\lesssim \sqrt{\frac{\mathrm{Var}\big(\varphi(Y_1) - \varphi(Y_1)\big)\Upsilon(n, \delta)}{(1 \vee L/b)(M + d)}} + \Upsilon(n, \delta) \tag{25}$$

and

$$\left| \frac{1}{n} \sum_{j=1}^{n} \log \frac{\mathcal{T}_T[e^\varphi](Z_j)}{\mathcal{T}_T[e^{\varphi^*}](Z_j)} - \mathbb{E} \log \frac{\mathcal{T}_T[e^\varphi](Z_1)}{\mathcal{T}_T[e^{\varphi^*}](Z_1)} \right|$$
$$\lesssim \sqrt{\mathrm{Var}\left(\log \frac{\mathcal{T}_T[e^\varphi](Z_1)}{\mathcal{T}_T[e^{\varphi^*}](Z_1)}\right) \frac{\Upsilon(n, \delta)}{(1 \vee L/b)(M + d)}} + \Upsilon(n, \delta) \tag{26}$$

simultaneously for all $\varphi \in \Phi$.

**Step 2: variance bounds.** The most challenging and technically involved part of the proof is to show that both

$$\mathrm{Var}\big(\varphi^*(Y_1) - \varphi(Y_1)\big) \quad \text{and} \quad \mathrm{Var}\left(\log \frac{\mathcal{T}_T[e^\varphi](Z_1)}{\mathcal{T}_T[e^{\varphi^*}](Z_1)}\right)$$

can be controlled by a sub-quadratic function of $\mathsf{KL}(\pi^*, \pi^\varphi)$. This is the key ingredient in derivation of rates of convergence possibly faster than $\mathcal{O}(n^{-1/2})$. We would like to note that in Puchkin et al. (2025), the authors used properties of log-densities with subquadratic growth for a similar purpose (see their Lemmata C.2 and C.3). Unfortunately, we cannot apply them in our setup, because we do not have access to samples from $\pi^*$, we are given only samples drawn from the marginals of $\pi^*$. This makes us to use another strategy. Let us observe that for any $\varphi \in \Phi$

$$\mathsf{KL}(\pi^*, \pi^\varphi) = \mathbb{E}\big(\varphi^*(X_T^*) - \varphi(X_T^*)\big) + \mathbb{E} \log \mathbb{E}\left[e^{\varphi(X_T^*) - \varphi^*(X_T^*)} \,\big|\, X_0\right]. \tag{27}$$

Due to the tower rule, $\mathsf{KL}(\pi^*, \pi^\varphi)$ can be considered as expectation of the conditional cumulant generating function

$$G(\lambda, X_0) = \mathbb{E} \log \mathbb{E}\left[e^{\lambda(\varphi(X_T^*) - \varphi^*(X_T^*))} \,\big|\, X_0\right] - \lambda \mathbb{E}\left[\varphi(X_T^*) - \varphi^*(X_T^*) \,\big|\, X_0\right]$$

at $\lambda = 1$, that is, $\mathsf{KL}(\pi^*, \pi^\varphi) = \mathbb{E}_{X_0 \sim \rho_0} G(1, X_0)$. In Appendix B.3, we prove the following general result relating variance and the cumulant generating function of a sub-exponential random variable.

**Lemma B.3** (restatement of Lemma B.1). *Assume that a random variable $\xi$ has a finite $\psi_1$-norm. Then, for any $\lambda$ satisfying the inequality*

$$4e|\lambda| \left(5 + \log \frac{52\|\xi - \mathbb{E}\xi\|_{\psi_1}^2}{\mathrm{Var}(\xi)}\right) \|\xi - \mathbb{E}\xi\|_{\psi_1} \leqslant 1,$$

*it holds that*

$$\mathrm{Var}(\xi) \leqslant \frac{4}{\lambda^2} \log \mathbb{E} e^{\lambda(\xi - \mathbb{E}\xi)}.$$

Based on this result, in Appendix C.1 we prove Lemma C.1 claiming that

$$
\mathbb{E}_{X_0 \sim \rho_0} \mathrm{Var}\big(\varphi(X_T^*) - \varphi^*(X_T^*) \,|\, X_0\big)
$$
$$
\lesssim \left(1 \vee \frac{L}{b}\right)(M + d)\, \mathsf{KL}\,(\pi^*, \pi^\varphi)\left(1 \vee \log \frac{(1 \vee L/b)(M + d)}{\mathsf{KL}\,(\pi^*, \pi^\varphi)}\right).
$$

However, this bound is not enough, because $\mathrm{Var}\big(\varphi^*(Y_1) - \varphi(Y_1)\big)$ is always not smaller than the conditional variance in general. Fortunately, we can use properties of the Ornstein-Uhlenbeck process (in particular, Lemma B.3 from Puchkin et al. (2025)) to derive the following result.

**Lemma B.4** (restatement of Lemma B.2). *Under the assumptions of Theorem 1, for any $\omega > 0$ there exists*

$$
T_0 \lesssim \frac{1}{b}\left(\log\log\frac{1}{\omega} \vee \log\left(d + LR^2 + \frac{Ld^2}{b} + M\right)\right)
$$

*such that for any $T \geq T_0$ and any $\varphi \in \Phi$ satisfying the inequality $\mathrm{Var}\big(\varphi(X_T^*) - \varphi^*(X_T^*)\big) \geq \omega$ it holds that*

$$
\mathrm{Var}\left(\mathbb{E}\left[\varphi(X_T^*) - \varphi^*(X_T^*) \,\big|\, X_0\right]\right) \leq \frac{1}{2}\mathrm{Var}\big(\varphi(X_T^*) - \varphi^*(X_T^*)\big) \leq \mathbb{E}\mathrm{Var}\big(\varphi(X_T^*) - \varphi^*(X_T^*) \,|\, X_0\big).
$$

The proof of Lemma B.2 is postponed to Appendix B.4. Taking $\omega = \Upsilon(n, 1)$, we immediately obtain that there is

$$
T_0 \lesssim \frac{1}{b}\left(\log\log\frac{1}{\omega} \vee \log\left(d + LR^2 + \frac{Ld^2}{b} + M\right)\right)
$$

such that for any $T \geq T_0$ and any $\varphi \in \Phi$ satisfying the inequality $\mathrm{Var}\big(\varphi(X_T^*) - \varphi^*(X_T^*)\big) \geq \Upsilon(n, 1)$ it holds that

$$
\mathrm{Var}\left(\mathbb{E}\left[\varphi(X_T^*) - \varphi^*(X_T^*) \,\big|\, X_0\right]\right) \leq \frac{1}{2}\mathrm{Var}\big(\varphi(X_T^*) - \varphi^*(X_T^*)\big) \leq \mathbb{E}\mathrm{Var}\big(\varphi(X_T^*) - \varphi^*(X_T^*) \,|\, X_0\big).
$$

In other words, if $T$ satisfies the conditions of Theorem 1, then either the variance of $\varphi(X_T^*) - \varphi^*(X_T^*)$ is sufficiently small, so that

$$
\mathrm{Var}\big(\varphi(X_T^*) - \varphi^*(X_T^*)\big) \leq \Upsilon(n, 1) \leq \Upsilon(n, \delta),
$$

or it holds that

$$
\mathrm{Var}\big(\varphi(X_T^*) - \varphi^*(X_T^*)\big) \leq 2\mathbb{E}\mathrm{Var}\big(\varphi(X_T^*) - \varphi^*(X_T^*) \,|\, X_0\big)
$$
$$
\lesssim \left(1 \vee \frac{L}{b}\right)(M + d)\, \mathsf{KL}\,(\pi^*, \pi^\varphi)\left(1 \vee \log \frac{(1 \vee L/b)(M + d)}{\mathsf{KL}\,(\pi^*, \pi^\varphi)}\right).
$$

Summing up these two bounds, we obtain that

$$
\mathrm{Var}\big(\varphi(X_T^*) - \varphi^*(X_T^*)\big) \lesssim \Upsilon(n, \delta) + \left(1 \vee \frac{L}{b}\right)(M + d)
$$
$$
\cdot\, \mathsf{KL}\,(\pi^*, \pi^\varphi)\left(1 \vee \log \frac{(1 \vee L/b)(M + d)}{\mathsf{KL}\,(\pi^*, \pi^\varphi)}\right). \tag{28}
$$

On the other hand, according to Lemma C.4, it holds that

$$
\mathrm{Var}\left(\log \frac{\mathcal{T}_T[e^\varphi](Z_1)}{\mathcal{T}_T[e^{\varphi^*}](Z_1)}\right)
$$
$$
\lesssim \mathrm{Var}\big(\varphi(X_T^*) - \varphi^*(X_T^*)\big)
$$
$$
+ \left(1 \vee \frac{L}{b}\right)(M + d)\, \mathsf{KL}\,(\pi^*, \pi^\varphi)\left(1 \vee \log \frac{(1 \vee L/b)(M + d)}{\mathsf{KL}\,(\pi^*, \pi^\varphi)}\right) \tag{29}
$$
$$
\lesssim \Upsilon(n, \delta) + \left(1 \vee \frac{L}{b}\right)(M + d)\, \mathsf{KL}\,(\pi^*, \pi^\varphi)\left(1 \vee \log \frac{(1 \vee L/b)(M + d)}{\mathsf{KL}\,(\pi^*, \pi^\varphi)}\right).
$$

This reveals a dichotomy. If $T$ is sufficiently large, then either the right-hand sides of (25) and (26) are of order $\Upsilon(n, \delta)$ or we have that the loss functions of interest meet Bernstein-type conditions (28) and (29) allowing us to derive faster rates of convergence.

**Step 3: generalization error bound.** Bringing together (25), (26), (28), and (29), we obtain that, for any $\delta \in (0, 1)$, with probability at least $(1 - \delta)$ it holds that

$$\left| \frac{1}{n} \sum_{i=1}^{n} \big( \varphi(Y_i) - \varphi(Y_i) \big) - \mathbb{E}\big( \varphi(Y_1) - \varphi(Y_1) \big) \right|$$

$$\lesssim \sqrt{\Upsilon(n, \delta) \, \mathsf{KL}\left(\pi^*, \pi^\varphi\right) \left( 1 \vee \log \frac{(1 \vee L/b)(M + d)}{\mathsf{KL}\left(\pi^*, \pi^\varphi\right)} \right)} + \Upsilon(n, \delta)$$

and

$$\left| \frac{1}{n} \sum_{j=1}^{n} \log \frac{\mathcal{T}_T[e^\varphi](Z_j)}{\mathcal{T}_T[e^{\varphi^*}](Z_j)} - \mathbb{E} \log \frac{\mathcal{T}_T[e^\varphi](Z_1)}{\mathcal{T}_T[e^{\varphi^*}](Z_1)} \right|$$

$$\lesssim \sqrt{\Upsilon(n, \delta) \, \mathsf{KL}\left(\pi^*, \pi^\varphi\right) \left( 1 \vee \log \frac{(1 \vee L/b)(M + d)}{\mathsf{KL}\left(\pi^*, \pi^\varphi\right)} \right)} + \Upsilon(n, \delta)$$

simultaneously for all $\varphi \in \Phi$. With these inequalities at hand, a generalization error bound is almost straightforward. For any $\varphi \in \Phi$, let

$$\widehat{\mathsf{KL}}(\pi^*, \pi^\varphi) = \frac{1}{n} \sum_{i=1}^{n} \big( \varphi(Y_i) - \varphi(Y_i) \big) + \frac{1}{n} \sum_{j=1}^{n} \log \frac{\mathcal{T}_T[e^\varphi](Z_j)}{\mathcal{T}_T[e^{\varphi^*}](Z_j)}.$$

Then, due to (10), (11), and the triangle inequality, it holds that

$$\left| \widehat{\mathsf{KL}}(\pi^*, \pi^\varphi) - \mathsf{KL}(\pi^*, \pi^\varphi) \right| \lesssim \sqrt{\Upsilon(n, \delta) \, \mathsf{KL}\left(\pi^*, \pi^\varphi\right) \left( 1 \vee \log \frac{(1 \vee L/b)(M + d)}{\mathsf{KL}\left(\pi^*, \pi^\varphi\right)} \right)} + \Upsilon(n, \delta)$$

with probability at least $(1 - \delta)$ simultaneously for all $\varphi \in \Phi$. Let us define[4]

$$\varphi^\circ \in \operatorname*{argmin}_{\varphi \in \Phi} \mathsf{KL}(\pi^*, \pi^\varphi) \quad \text{and} \quad \Delta = \inf_{\varphi \in \Phi} \mathsf{KL}(\pi^*, \pi^\varphi).$$

Note that, due to the definition of the empirical risk minimizer, we have

$$\widehat{\mathsf{KL}}(\pi^*, \widehat{\pi}) = \widehat{\mathsf{KL}}\left(\pi^*, \pi^{\widehat{\varphi}}\right) \leqslant \widehat{\mathsf{KL}}\left(\pi^*, \pi^\varphi\right) \quad \text{for all } \varphi \in \Phi.$$

Then, with probability at least $(1 - \delta)$, it holds that

$$\mathsf{KL}(\pi^*, \widehat{\pi}) - \Delta \leqslant \mathsf{KL}(\pi^*, \widehat{\pi}) - \widehat{\mathsf{KL}}\left(\pi^*, \pi^{\widehat{\varphi}}\right) + \widehat{\mathsf{KL}}\left(\pi^*, \pi^{\varphi^\circ}\right) - \mathsf{KL}\left(\pi^*, \pi^{\varphi^\circ}\right)$$

$$\lesssim \Upsilon(n, \delta) + \sqrt{\Upsilon(n, \delta) \mathsf{KL}\left(\pi^*, \widehat{\pi}\right) \left( 1 \vee \log \frac{(1 \vee L/b)(M + d)}{\mathsf{KL}\left(\pi^*, \widehat{\pi}\right)} \right)}$$

$$+ \sqrt{\Upsilon(n, \delta) \Delta \left( 1 \vee \log \frac{(1 \vee L/b)(M + d)}{\Delta} \right)}$$

$$\leqslant \Upsilon(n, \delta) + \sqrt{\Upsilon(n, \delta) \mathsf{KL}\left(\pi^*, \widehat{\pi}\right) \left( 1 \vee \log \frac{(1 \vee L/b)(M + d)}{\Delta} \right)}$$

$$+ \sqrt{\Upsilon(n, \delta) \Delta \left( 1 \vee \log \frac{(1 \vee L/b)(M + d)}{\Delta} \right)}.$$

In view of the inequality $\sqrt{a + b} \leqslant \sqrt{a} + \sqrt{b}$, which holds for all non-negative $a$ and $b$, we obtain that

$$\mathsf{KL}(\pi^*, \widehat{\pi}) - \Delta \lesssim \Upsilon(n, \delta) + \sqrt{\Upsilon(n, \delta) \left( \mathsf{KL}\left(\pi^*, \widehat{\pi}\right) - \Delta \right) \left( 1 \vee \log \frac{(1 \vee L/b)(M + d)}{\Delta} \right)}$$

$$+ \sqrt{\Upsilon(n, \delta) \Delta \left( 1 \vee \log \frac{(1 \vee L/b)(M + d)}{\Delta} \right)}.$$

---

[4]If the minimum is not attained, one should consider a minimizing sequence.

on the same event. Solving the quadratic inequality with respect to $\mathsf{KL}(\pi^*, \widehat{\pi}) - \Delta$, we deduce the claim of the theorem: it holds that

$$
\mathsf{KL}(\pi^*, \widehat{\pi}) - \Delta \lesssim \sqrt{\Upsilon(n, \delta)\Delta \left(1 \vee \log \frac{(1 \vee L/b)(M + d)}{\Delta}\right)}
$$
$$
+ \Upsilon(n, \delta) \left(1 \vee \log \frac{(1 \vee L/b)(M + d)}{\Delta}\right)
$$

on an event of probability at least $(1 - \delta)$.

$\square$

## B.3 PROOF OF LEMMA B.1

Let us note that it is enough to check the theorem statement of $\lambda > 0$. In the case $\lambda < 0$ one should introduce $\xi' = -\xi$ and consider its cumulant generating function. Let $g(\lambda)$ stand for the cumulant generating function of $\xi$, that is,

$$
g(\lambda) = \log \mathbb{E} e^{\lambda(\xi - \mathbb{E}\xi)}.
$$

Fix an arbitrary $\lambda > 0$ satisfying (24). For a Borel function $f : \mathbb{R} \to \mathbb{R}$, let us define

$$
\mathsf{P}_\lambda f(\xi) = \frac{\mathbb{E}\left[f(\xi)e^{\lambda(\xi - \mathbb{E}\xi)}\right]}{\mathbb{E} e^{\lambda(\xi - \mathbb{E}\xi)}}.
$$

We would like to note that $\mathsf{P}_\lambda f(\xi)$ can be considered as an expectation with respect to a probability measure $\mathsf{P}_\lambda$. Let

$$
\|\xi - \mathbb{E}\xi\|_{\psi_1(\mathsf{P}_\lambda)} = \inf\left\{t > 0 : \mathsf{P}_\lambda e^{|\xi - \mathbb{E}\xi|/t} \leqslant 2\right\}
$$

stand for the corresponding Orlicz norm. Then, according to Lemma E.5, the derivatives of the cumulant generating function $g(\lambda)$ can be expressed as follows:

$$
g'(\lambda) = \mathsf{P}_\lambda(\xi - \mathbb{E}\xi), \quad g''(\lambda) = \mathsf{P}_\lambda \left(\xi - \mathsf{P}_\lambda \xi\right)^2, \quad \text{and} \quad g'''(\lambda) = \mathsf{P}_\lambda \left(\xi - \mathsf{P}_\lambda \xi\right)^3.
$$

It is straightforward to observe that $g'(0) = 0$ and that $g''(0) = \mathrm{Var}(\xi) \leqslant 4\|\xi - \mathbb{E}\xi\|_{\psi_1}^2$ due to Lemma E.8. Let us elaborate on $g'''(\lambda)$. For this purpose, let us introduce

$$
\varepsilon = \frac{1}{5 + \log\left(\frac{52\|\xi - \mathbb{E}\xi\|_{\psi_1}^2}{g''(0)}\right)} \leqslant \frac{1}{5 + \log(13)}. \tag{30}
$$

Applying Hölder's inequality, we obtain that

$$
|g'''(\lambda)| \leqslant \mathsf{P}_\lambda |\xi - \mathsf{P}_\lambda \xi|^3 \leqslant \left(\mathsf{P}_\lambda \left(\xi - \mathsf{P}_\lambda \xi\right)^2\right)^{1-\varepsilon} \left(\mathsf{P}_\lambda |\xi - \mathsf{P}_\lambda \xi|^{2+1/\varepsilon}\right)^\varepsilon
$$
$$
= (g''(\lambda))^{1-\varepsilon} \left(\mathsf{P}_\lambda |\xi - \mathsf{P}_\lambda \xi|^{2+1/\varepsilon}\right)^\varepsilon. \tag{31}
$$

Consider the second factor in the right-hand side. Due to the triangle inequality, it holds that

$$
\left(\mathsf{P}_\lambda |\xi - \mathsf{P}_\lambda \xi|^{2+1/\varepsilon}\right)^{\varepsilon/(1+2\varepsilon)} \leqslant \left(\mathsf{P}_\lambda |\xi - \mathbb{E}\xi|^{2+1/\varepsilon}\right)^{\varepsilon/(1+2\varepsilon)} + \mathsf{P}_\lambda |\xi - \mathbb{E}\xi|.
$$

In view of Lemma E.8 and Lemma E.6, we have

$$
\mathsf{P}_\lambda |\xi - \mathbb{E}\xi| \leqslant 2\|\xi - \mathbb{E}\xi\|_{\psi_1(\mathsf{P}_\lambda)} \leqslant \frac{2\|\xi - \mathbb{E}\xi\|_{\psi_1}}{1 - \lambda\|\xi - \mathbb{E}\xi\|_{\psi_1}}
$$

and

$$
\left(\mathsf{P}_\lambda |\xi - \mathbb{E}\xi|^{2+1/\varepsilon}\right)^{\varepsilon/(1+2\varepsilon)} \leqslant 2^{\varepsilon/(1+2\varepsilon)} \left(3 + \frac{1}{\varepsilon}\right) \|\xi - \mathbb{E}\xi\|_{\psi_1(\mathsf{P}_\lambda)}
$$
$$
\leqslant 2^{\varepsilon/(1+2\varepsilon)} \left(3 + \frac{1}{\varepsilon}\right) \frac{\|\xi - \mathbb{E}\xi\|_{\psi_1}}{1 - \lambda\|\xi - \mathbb{E}\xi\|_{\psi_1}}.
$$

This yields that

$$\left(\mathsf{P}_\lambda \,|\xi - \mathsf{P}_\lambda \xi|^{2+1/\varepsilon}\right)^{\varepsilon/(1+2\varepsilon)} \leqslant 2^{\varepsilon/(1+2\varepsilon)} \left(3 + \frac{1}{\varepsilon}\right) \frac{\|\xi - \mathbb{E}\xi\|_{\psi_1}}{1 - \lambda\|\xi - \mathbb{E}\xi\|_{\psi_1}} + \frac{2\|\xi - \mathbb{E}\xi\|_{\psi_1}}{1 - \lambda\|\xi - \mathbb{E}\xi\|_{\psi_1}}$$

$$= 2^{\varepsilon/(1+2\varepsilon)} \left(2^{(1+\varepsilon)/(1+2\varepsilon)} + 3 + \frac{1}{\varepsilon}\right) \frac{\|\xi - \mathbb{E}\xi\|_{\psi_1}}{1 - \lambda\|\xi - \mathbb{E}\xi\|_{\psi_1}}$$

$$\leqslant 2^{\varepsilon/(1+2\varepsilon)} \left(5 + \frac{1}{\varepsilon}\right) \frac{\|\xi - \mathbb{E}\xi\|_{\psi_1}}{1 - \lambda\|\xi - \mathbb{E}\xi\|_{\psi_1}}.$$

Since, according to the condition of the lemma, $\lambda$ satisfies the inequality

$$\|\xi - \mathbb{E}\xi\|_{\psi_1} |\lambda| \leqslant \frac{1}{4e} \left(5 + \log \frac{52\|\xi - \mathbb{E}\xi\|_{\psi_1}^2}{g''(0)}\right)^{-1} = \frac{\varepsilon}{4e},$$

we have

$$\left(\mathsf{P}_\lambda \,|\xi - \mathsf{P}_\lambda \xi|^{2+1/\varepsilon}\right)^{\varepsilon/(1+2\varepsilon)} \leqslant 2^{\varepsilon/(1+2\varepsilon)} \left(5 + \frac{1}{\varepsilon}\right) \left(1 - \frac{\varepsilon}{4e}\right)^{-1} \|\xi - \mathbb{E}\xi\|_{\psi_1}.$$

Combining this bound with (31), we obtain that

$$|g'''(\lambda)| \leqslant 2^\varepsilon \left(5 + \frac{1}{\varepsilon}\right)^{1+2\varepsilon} \left(1 - \frac{\varepsilon}{4e}\right)^{-1-2\varepsilon} \|\xi - \mathbb{E}\xi\|_{\psi_1}^{1+2\varepsilon} (g''(\lambda))^{1-\varepsilon}$$

$$= \left(\frac{50}{(1 - \varepsilon/(6e))^2}\right)^\varepsilon \left(1 + \frac{1}{5\varepsilon}\right)^{2\varepsilon} \left(1 - \frac{\varepsilon}{4e}\right)^{-1} \cdot \left(5 + \frac{1}{\varepsilon}\right) \|\xi - \mathbb{E}\xi\|_{\psi_1}^{1+2\varepsilon} (g''(\lambda))^{1-\varepsilon}.$$

The inequality $\varepsilon \leqslant \left(5 + \log(13)\right)^{-1}$ implies that

$$\frac{50}{(1 - \varepsilon/(4e))^2} < 52.$$

Moreover, taking into account the inequality $1 + u \leqslant e^u$, $u \in \mathbb{R}$, we obtain that

$$\left(\frac{50}{(1 - \varepsilon/(4e))^2}\right)^\varepsilon \left(1 + \frac{1}{5\varepsilon}\right)^{2\varepsilon} \left(1 - \frac{\varepsilon}{4e}\right)^{-1} \leqslant 52^\varepsilon \cdot e^{2/5} \cdot \left(1 - \frac{1}{4e}(5 + \log 13)\right)^{-1} \leqslant 2 \cdot 52^\varepsilon,$$

and then

$$|g'''(\lambda)| \leqslant 2 \cdot 52^\varepsilon \left(5 + \frac{1}{\varepsilon}\right) \|\xi - \mathbb{E}\xi\|_{\psi_1}^{1+2\varepsilon} (g''(\lambda))^{1-\varepsilon}.$$

This inequality yields that

$$\left|\frac{\mathrm{d}}{\mathrm{d}\lambda}(g''(\lambda))^\varepsilon\right| = \varepsilon(g''(\lambda))^{\varepsilon-1}|g'''(\lambda)| \leqslant 2 \cdot 52^\varepsilon(1 + 5\varepsilon)\|\xi - \mathbb{E}\xi\|_{\psi_1}^{1+2\varepsilon},$$

and, therefore, for any $\lambda$ satisfying (24) we have

$$(g''(\lambda))^\varepsilon \geqslant (g''(0))^\varepsilon - 2|\lambda| \cdot 52^\varepsilon(1 + 5\varepsilon)\|\xi - \mathbb{E}\xi\|_{\psi_1}^{1+2\varepsilon}$$

$$= (g''(0))^\varepsilon \left(1 - 2|\lambda|(1 + 5\varepsilon)\|\xi - \mathbb{E}\xi\|_{\psi_1} \left(\frac{52\|\xi - \mathbb{E}\xi\|_{\psi_1}^2}{g''(0)}\right)^\varepsilon\right)$$

$$\geqslant (g''(0))^\varepsilon \left(1 - 2|\lambda|\|\xi - \mathbb{E}\xi\|_{\psi_1} \left(\frac{52 \cdot e^5 \|\xi - \mathbb{E}\xi\|_{\psi_1}^2}{g''(0)}\right)^\varepsilon\right).$$

Taking into account (30), we obtain that

$$g''(\lambda) \geqslant g''(0) \left(1 - 2e\|\xi - \mathbb{E}\xi\|_{\psi_1}|\lambda|\right)^{5+\log(52\|\xi - \mathbb{E}\xi\|_{\psi_1}^2/g''(0))} \quad \text{for all } \lambda \text{ fulfilling (24)}.$$

Hence, for any $\lambda$ such that

$$4e|\lambda| \left(5 + \log \frac{52\|\xi - \mathbb{E}\xi\|_{\psi_1}^2}{g''(0)}\right) \|\xi - \mathbb{E}\xi\|_{\psi_1} \leqslant 1,$$

it holds that

$$g''(\lambda) \geqslant g''(0) \left(1 - \frac{1}{2\big(5 + \log(52\|\xi - \mathbb{E}\xi\|_{\psi_1}^2/g''(0))\big)}\right)^{5 + \log(52\|\xi - \mathbb{E}\xi\|_{\psi_1}^2/g''(0))} \geqslant \frac{g''(0)}{2}.$$

In the last line we used the fact that $(1 - x/2) \geqslant 2^{-x}$ for all $x \in (0, 1/2)$ and that

$$5 + \log\left(\frac{52\|\xi - \mathbb{E}\xi\|_{\psi_1}^2}{g''(0)}\right) \geqslant 5 + \log 13 > 5.$$

Then, using Taylor's expansion with the Lagrange remainder term, we deduce that

$$g(\lambda) \geqslant \frac{g''(0)\lambda^2}{4} \quad \text{for any } \lambda \text{ such that} \quad 4e|\lambda|\left(5 + \log\frac{52\|\xi - \mathbb{E}\xi\|_{\psi_1}^2}{g''(0)}\right)\|\xi - \mathbb{E}\xi\|_{\psi_1} \leqslant 1.$$

Hence, we proved that

$$\mathrm{Var}(\xi) = g''(0) \leqslant \frac{4g(\lambda)}{\lambda^2} = \frac{4}{\lambda^2}\log\mathbb{E}e^{\lambda(\xi - \mathbb{E}\xi)}$$

for any $\lambda$ such that

$$4e|\lambda|\left(5 + \log\frac{52\|\xi - \mathbb{E}\xi\|_{\psi_1}^2}{\mathrm{Var}(\xi)}\right)\|\xi - \mathbb{E}\xi\|_{\psi_1} \leqslant 1.$$

$\square$

### B.4 Proof of Lemma B.2

Let us fix an arbitrary $\omega > 0$ and consider a log-potential $\varphi \in \Phi$ satisfying the inequality $\mathrm{Var}\big(\varphi(X_T^*) - \varphi^*(X_T^*)\big) \geqslant \omega$. Also, let $T \gtrsim \log(d)/b$ be such that $\mathcal{K}(T) \leqslant \sqrt{2}$, where the function $\mathcal{K}(t)$ is defined in (39). The proof of Lemma B.2 relies on Lemma C.2, which provides an upper bound on the variance of $\mathbb{E}\left[\varphi(X_T^*) - \varphi^*(X_T^*)\,\big|\,X_0\right]$. Due to the mixing of the diffusion process $X^*$, it is not surprising that $\mathrm{Var}\left(\mathbb{E}\left[\varphi(X_T^*) - \varphi^*(X_T^*)\,\big|\,X_0\right]\right)$ tends to zero as $T$ grows. To be more precise, Lemma C.2 claims that

$$\mathrm{Var}\left(\mathbb{E}\left[\varphi(X_T^*) - \varphi^*(X_T^*)\,\big|\,X_0\right]\right)$$
$$\lesssim de^{-2bT}\left((M^2 \vee 1)\mathrm{Var}\big(\varphi(X_T^*) - \varphi^*(X_T^*)\big)^{1/\mathcal{K}^2(T)} + \mathrm{Var}\big(\varphi(X_T^*) - \varphi^*(X_T^*)\big)^{2/\mathcal{K}^2(T)-1}\right)$$
$$\cdot \exp\left\{\mathcal{O}\left(\sqrt{d}\left(M \vee 1 \vee e^{-bT}\log\left(\widetilde{M} \vee LR^2 \vee (Ld^2/b)\right)\right)\right)\right\}$$
$$\lesssim (M^2 \vee 1)de^{-2bT}\left(\omega^{-1+1/\mathcal{K}^2(T)} + \omega^{-2+2/\mathcal{K}^2(T)}\right)\mathrm{Var}\big(\varphi(X_T^*) - \varphi^*(X_T^*)\big)$$
$$\cdot \exp\left\{\mathcal{O}\left(\sqrt{d}\left(M \vee 1 \vee e^{-bT}\log\left(\widetilde{M} \vee LR^2 \vee (Ld^2/b)\right)\right)\right)\right\},$$

where $\widetilde{M}$ is given by (32). Let us note that the definition of $\widetilde{M}$ implies that

$$\widetilde{M} = L\mathrm{Tr}\big(\Sigma^{-1}\mathrm{Var}(X_T^*)\big) + L\left\|\Sigma^{-1/2}(\mathbb{E}X_T^* - m)\right\|^2 + 4M \lesssim Ld + LR^2 + M.$$

The last inequality yields that

$$\mathrm{Var}\left(\mathbb{E}\left[\varphi(X_T^*) - \varphi^*(X_T^*)\,\big|\,X_0\right]\right)$$
$$\lesssim (M^2 \vee 1)de^{-2bT}\left(\frac{1}{\omega} \vee \frac{1}{\omega^2}\right)^{1-1/\mathcal{K}^2(T)}\mathrm{Var}\big(\varphi(X_T^*) - \varphi^*(X_T^*)\big)$$
$$\cdot \exp\left\{\mathcal{O}\left(\sqrt{d}\left(M \vee 1 \vee e^{-bT}\log\left(M \vee LR^2 \vee (Ld^2/b)\right)\right)\right)\right\}.$$

Since, due to (39)

$$\log\mathcal{K}(T) = 2e^2\sqrt{d}\arcsin(e^{-bT}) - 5e^2\sqrt{d}\log\left(1 - e^{-2bT}\right) \lesssim \sqrt{d}e^{-bT} \quad \text{for all } T \gtrsim \frac{\log d}{b},$$

we obtain that

$$\mathrm{Var}\left(\mathbb{E}\left[\varphi(X_T^*) - \varphi^*(X_T^*) \,\middle|\, X_0\right]\right)$$

$$\lesssim (M^2 \vee 1)\, de^{-2bT} \mathrm{Var}\big(\varphi(X_T^*) - \varphi^*(X_T^*)\big) \exp\left\{\mathcal{O}\left(\sqrt{d}\, e^{-bT}\log\left(1 \vee \frac{1}{\omega}\right)\right)\right\}$$

$$\cdot \exp\left\{\mathcal{O}\left(\sqrt{d}\left(M \vee 1 \vee e^{-bT}\log\left(\widetilde{M} \vee LR^2 \vee (Ld^2/b)\right)\right)\right)\right\}.$$

Let $T_0 > 0$ be such that

$$(M^2 \vee 1)\, de^{-2bT}\exp\left\{\mathcal{O}\left(\sqrt{d}\, e^{-bT}\log\left(1 \vee \frac{1}{\omega}\right)\right)\right\}$$

$$\cdot \exp\left\{\mathcal{O}\left(\sqrt{d}\left(M \vee 1 \vee e^{-bT}\log\left(\widetilde{M} \vee LR^2 \vee (Ld^2/b)\right)\right)\right)\right\} \leqslant 1/2.$$

Note that one can take $T_0$ satisfying the inequality

$$T_0 \lesssim \frac{1}{b}\left(\log\log\frac{1}{\omega} \vee \log\left(d + LR^2 + \frac{Ld^2}{b} + M\right)\right).$$

Then, for any $T \geqslant T_0$, it holds that

$$\mathrm{Var}\left(\mathbb{E}\left[\varphi(X_T^*) - \varphi^*(X_T^*) \,\middle|\, X_0\right]\right) \leqslant \frac{1}{2}\mathrm{Var}\big(\varphi(X_T^*) - \varphi^*(X_T^*)\big).$$

Since

$$\mathrm{Var}\big(\varphi(X_T^*) - \varphi^*(X_T^*)\big) = \mathbb{E}\mathrm{Var}\big(\varphi(X_T^*) - \varphi^*(X_T^*)\,|\,X_0\big) + \mathrm{Var}\left(\mathbb{E}\left[\varphi(X_T^*) - \varphi^*(X_T^*)\,\middle|\,X_0\right]\right),$$

this yields that

$$\mathrm{Var}\left(\mathbb{E}\left[\varphi(X_T^*) - \varphi^*(X_T^*)\,\middle|\,X_0\right]\right) \leqslant \frac{1}{2}\mathrm{Var}\big(\varphi(X_T^*) - \varphi^*(X_T^*)\big) \leqslant \mathbb{E}\mathrm{Var}\big(\varphi(X_T^*) - \varphi^*(X_T^*)\,|\,X_0\big).$$

$\square$

## C    LOG-POTENTIAL VARIANCE AND CONDITIONAL VARIANCE BOUNDS

In this section, we elaborate on variance bounds. The presented results is a key argument in derivation of the rates of convergence, which are possibly faster than $\mathcal{O}(n^{-1/2})$. We use Lemma C.2 in the proof of Lemma B.2 and then combine the results of Lemma B.2 and Lemma C.1 to derive an important intermediate result in the proof of Theorem 1, the inequality (28).

**Lemma C.1.** *Grant Assumptions 1, 2, and 4. Then it holds that*

$$\mathbb{E}_{X_0 \sim \rho_0} \mathrm{Var}\big(\varphi(X_T^*) - \varphi^*(X_T^*)\,\big|\,X_0\big) \lesssim \mathsf{KL}\left(\pi^*, \pi^\varphi\right)$$

$$+ \left(1 \vee \frac{L}{b}\right)\left(d + M + R^2 e^{-2bT}\right) \mathsf{KL}\left(\pi^*, \pi^\varphi\right)$$

$$\cdot \log\left(1 + \frac{(1 \vee L/b)(d + M + R^2 e^{-2bT})}{\mathsf{KL}\left(\pi^*, \pi^\varphi\right)}\right).$$

*The hidden constants behind $\lesssim$ are absolute.*

**Lemma C.2.** *Grant Assumptions 1, 2, and 4. Let $T$ be large enough in a sense that $\mathcal{K}^2(T) \leqslant 2$ with $\mathcal{K}(t)$ defined in (39). Then it holds that*

$$\mathrm{Var}\left(\mathbb{E}\left[\varphi(X_T^*) - \varphi^*(X_T^*)\,\middle|\,X_0\right]\right)$$

$$\lesssim de^{-2bT}\left((M^2 \vee 1)\mathrm{Var}\big(\varphi(X_T^*) - \varphi^*(X_T^*)\big)^{1/\mathcal{K}^2(T)} + \mathrm{Var}\big(\varphi(X_T^*) - \varphi^*(X_T^*)\big)^{2/\mathcal{K}^2(T)-1}\right)$$

$$\cdot \exp\left\{\mathcal{O}\left(\sqrt{d}\left(M \vee 1 \vee \log\left(\widetilde{M} \vee LR^2 \vee (Ld^2/b)\right)\right)e^{-bT}\right)\right\}.$$

*where $\mathcal{K}(t)$ is defined in (39),*

$$\widetilde{M} = L\mathrm{Tr}\big(\Sigma^{-1}\mathrm{Var}(X_T^*)\big) + L\left\|\Sigma^{-1/2}(\mathbb{E}X_T^* - m)\right\|^2 + 4M, \tag{32}$$

*and the hidden constants behind $\lesssim$ and $\mathcal{O}(\cdot)$ are absolute.*

The proofs of Lemma C.1 and Lemma C.2 are moved to Appendices C.1 and C.2, respectively. While Lemma C.1 is an almost immediate consequence of Lemma B.1 and the representation (27), the proof of Lemma C.2 relies on the following property of the Ornstein-Uhlenbeck operator.

**Lemma C.3.** *Let $\varphi^* : \mathbb{R}^d \to \mathbb{R}$ be such that $\varphi(x) \leqslant M$ for all $x \in \mathbb{R}^d$ and $\mathcal{T}_\infty e^{\varphi^*} \geqslant 1$. Let $f : \mathbb{R}^d \to \mathbb{R}$ be an arbitrary function such that*

$$0 \leqslant f(x) \leqslant A \left\| \Sigma^{-1/2}(x - m) \right\|^2 + B \quad \text{for all } x \in \mathbb{R}^d$$

*for some non-negative constants $A$ and $B$. Then, under Assumption 2, for any $T > 0$, it holds that*

$$\mathrm{Var}\left( \frac{\mathcal{T}_T[fe^{\varphi^*}](X_0)}{\mathcal{T}_T[e^{\varphi^*}](X_0)} \right) \leqslant F_{\max}^2 \left( \frac{\mathcal{T}_\infty[fe^{\varphi^*}]}{F_{\max}\mathcal{T}_\infty[e^{\varphi^*}]} \right)^{2/\mathcal{K}(T)}$$

$$\cdot \exp\left\{ 2M\big(\mathcal{K}(T) - 1\big) + 2\overline{\mathcal{A}}(T)\left( \mathcal{K}(T) + \frac{1}{\mathcal{K}(T)} \right) \right\}$$

$$\cdot \left\{ M^2 \left( \mathcal{K}(T) - \frac{1}{\mathcal{K}(T)} \right)^2 + 2\big(\overline{\mathcal{A}}(T)\big)^2 \left( \mathcal{K}(T) + \frac{1}{\mathcal{K}(T)} \right)^2 \right\}$$

$$+ \frac{F_{\max}^2}{e^2} \left( \mathcal{K}(T) - \frac{1}{\mathcal{K}(T)} \right)^2 \left( \frac{\mathcal{T}_\infty[fe^{\varphi^*}]}{F_{\max}\mathcal{T}_\infty[e^{\varphi^*}]} \right)^{4/\mathcal{K}(T) - 2\mathcal{K}(T)}$$

$$\cdot \exp\left\{ 2M\big(\mathcal{K}(T) - 1\big) + 2\overline{\mathcal{A}}(T)\left( \mathcal{K}(T) + \frac{1}{\mathcal{K}(T)} \right) \right\}.$$

*where*

$$F_{\max} = Be^M + 2Ae^M R^2 + \frac{2Ae^M}{b}(400d + d^2),$$

*the function $\mathcal{K}(t)$ is defined in (39), and for any $t > 0$*

$$\overline{\mathcal{A}}(t) = \left( \frac{be^2 R^2}{\sqrt{d}} + 4e^2\sqrt{d} \right) \arcsin(e^{-bt}) - 10e^2\sqrt{d}\log\big(1 - e^{-2bt}\big). \tag{33}$$

We provide the proof of Lemma C.3 in Appendix C.3. Finally, in Appendix C.4 we prove the following counterpart of Lemma C.1 for $\log\big(\mathcal{T}_T[e^{\varphi}](X_0)/\mathcal{T}_T[e^{\varphi^*}](X_0)\big)$.

**Lemma C.4.** *Let us fix an arbitrary $\varphi \in \Phi$ and denote*

$$\zeta = \log \frac{\mathcal{T}_T[e^{\varphi}](X_0)}{\mathcal{T}_T[e^{\varphi^*}](X_0)} - \mathbb{E}\left[ \varphi(X_T^*) - \varphi^*(X_T^*) \,\big|\, X_0 \right].$$

*Then, for any $T > 0$, it holds that*

$$\mathrm{Var}\left( \log \frac{\mathcal{T}_T[e^{\varphi}](X_0)}{\mathcal{T}_T[e^{\varphi^*}](X_0)} \right) \leqslant 2\mathrm{Var}\left( \mathbb{E}\left[ \varphi(X_T^*) - \varphi^*(X_T^*) \,\big|\, X_0 \right] \right)$$

$$+ 4e^{3/2}\,\mathsf{KL}(\pi^*, \pi^\varphi)\|\zeta\|_{\psi_1}\left( 1 + \frac{1}{2}\log\frac{2e\|\zeta\|_{\psi_1}}{\mathsf{KL}(\pi^*, \pi^\varphi)} \right).$$

Similarly to Lemmata C.1 and C.2, we apply Lemma C.4 on the second step of the proof of Theorem 1 to derive the inequality (29).

## C.1 Proof of Lemma C.1

Due to the definition of the KL-divergence, we have

$$\mathsf{KL}\left(\pi^*, \pi^\varphi\right) = \int_{\mathbb{R}^d} \int_{\mathbb{R}^d} \log\frac{\pi^*(x_0, x_T)}{\pi^\varphi(x_0, x_T)} \pi^*(x_0, x_T)\,\mathrm{d}x_0\mathrm{d}x_T$$

$$= \int_{\mathbb{R}^d} \big(\varphi^*(x_T) - \varphi(x_T)\big)\rho_T(x_T)\,\mathrm{d}x_T + \int_{\mathbb{R}^d} \log\frac{\mathcal{T}_T[e^{\varphi}](x_0)}{\mathcal{T}_T[e^{\varphi^*}](x_0)}\rho_0(x_0)\,\mathrm{d}x_0$$

$$= \mathbb{E}_{X_T^* \sim \rho_T}\big(\varphi^*(X_T^*) - \varphi(X_T^*)\big) + \mathbb{E}_{X_0 \sim \rho_0}\log\mathbb{E}\left[ e^{\varphi(X_T^*) - \varphi^*(X_T^*)} \,\big|\, X_0 \right].$$

Introducing the conditional cumulant generating function

$$G(\lambda, x_0) = \log \mathbb{E}\left[\exp\left\{\lambda\big(\varphi(X_T^*) - \varphi^*(X_T^*)\big)\right\} \,\Big|\, X_0 = x_0\right]$$
$$- \lambda\mathbb{E}\left[\varphi(X_T^*) - \varphi^*(X_T^*) \,|\, X_0 = x_0\right],$$

we observe that

$$\mathsf{KL}\left(\pi^*, \pi^\varphi\right) = \mathbb{E}_{X_0 \sim \rho_0} G(1, X_0).$$

For any $x_0$, let us denote

$$\Psi(x_0) = \left\|\varphi(X_T^*) - \varphi^*(X_T^*) - \mathbb{E}\big[\varphi(X_T^*) - \varphi^*(X_T^*) \,|\, X_0 = x_0\big]\right\|_{\psi_1(\pi^*(\cdot \,|\, x_0))},$$

where $\pi^*(\cdot \,|\, x_0)$ is the conditional distribution of $X_T^*$ given $X_0 = x_0$, and

$$\Lambda(x_0) = \frac{1}{4e\Psi(x_0)}\left(5 + \log\frac{52\Psi^2(x_0)}{\mathrm{Var}\big(\varphi(X_T^*) - \varphi^*(X_T^*) \,|\, X_0 = x_0\big)}\right)^{-1}.$$

Then, according to Lemma B.1,

$$\mathrm{Var}\big(\varphi(X_T^*) - \varphi^*(X_T^*) \,|\, X_0 = x_0\big) \leqslant \frac{4G(\lambda, x_0)}{\lambda^2} \quad \text{for any } x_0 \text{ and any } |\lambda| \leqslant \Lambda(x_0).$$

If $\Lambda(x_0) \leqslant 1$, then

$$\mathrm{Var}\big(\varphi(X_T^*) - \varphi^*(X_T^*) \,|\, X_0 = x_0\big) \leqslant 4G(1, x_0).$$

Otherwise, we have

$$\mathrm{Var}\big(\varphi(X_T^*) - \varphi^*(X_T^*) \,|\, X_0 = x_0\big) \leqslant \frac{4G\big(\Lambda(x_0), x_0\big)}{\Lambda^2(x_0)} \leqslant \frac{4G\big(1, x_0\big)}{\Lambda(x_0)}.$$

Here we used the fact that, for any $x_0$, the map $\lambda \mapsto G(\lambda, x_0)/\lambda$ is non-decreasing on $\mathbb{R}_+$. Thus, we obtain that

$$\mathrm{Var}\big(\varphi(X_T^*) - \varphi^*(X_T^*) \,|\, X_0 = x_0\big) \leqslant 4\left(1 \vee \frac{1}{\Lambda(x_0)}\right)G\big(1, x_0\big),$$

or, equivalently,

$$\frac{13e^5\Psi^2(x_0)}{G(1, x_0)} \leqslant \frac{52e^5\Psi^2(x_0)}{\mathrm{Var}\big(\varphi(X_T^*) - \varphi^*(X_T^*) \,|\, X_0 = x_0\big)}\left(1 \vee \frac{1}{\Lambda(x_0)}\right)$$
$$= \frac{52e^5\Psi^2(x_0)}{\mathrm{Var}\big(\varphi(X_T) - \varphi^*(X_T) \,|\, X_0 = x_0\big)}$$
$$\cdot\left(1 \vee 4e\Psi(x_0)\log\frac{52e^5\Psi^2(x_0)}{\mathrm{Var}\big(\varphi(X_T^*) - \varphi^*(X_T^*) \,|\, X_0 = x_0\big)}\right).$$

Then Lemma E.9 implies that

$$\frac{52e^5\Psi^2(x_0)}{\mathrm{Var}\big(\varphi(X_T^*) - \varphi^*(X_T^*) \,|\, X_0 = x_0\big)} \geqslant \frac{13e^5\Psi^2(x_0)}{G(1, x_0)}\left(1 \vee 4e\Psi(x_0)\log\frac{13e^5\Psi^2(x_0)}{G(1, x_0)}\right)^{-1}.$$

Hence, we have

$$\mathrm{Var}\big(\varphi(X_T^*) - \varphi^*(X_T^*) \,|\, X_0 = x_0\big) \leqslant 4G(1, x_0)\left(1 \vee 4e\Psi(x_0)\log\frac{13e^5\Psi^2(x_0)}{G(1, x_0)}\right)$$
$$\leqslant 4G(1, x_0)\left(1 + 4e\Psi(x_0)\log\left(1 + \frac{13e^5\Psi^2(x_0)}{G(1, x_0)}\right)\right).$$

Applying Lemma D.3, we obtain that

$$\Psi(x_0) \lesssim \left(1 \vee \frac{L}{b}\right)\left(d + M + e^{-2bT}\left\|\Sigma^{-1/2}(x_0 - m)\right\|^2\right) \quad \text{for any } x_0 \in \mathrm{supp}(\rho_0).$$

In view of Assumption 2, we have

$$\sup_{x_0 \in \text{supp}(\rho_0)} \Psi(x_0) \lesssim \left(1 \vee \frac{L}{b}\right)\left(d + M + R^2 e^{-2bT}\right).$$

This yields that

$$\mathbb{E}_{X_0 \sim \rho_0} \text{Var}\big(\varphi(X_T^*) - \varphi^*(X_T^*) \,|\, X_0\big)$$
$$\lesssim \mathbb{E}_{X_0 \sim \rho_0} G(1, X_0) + \left(1 \vee \frac{L}{b}\right)\left(d + M + R^2 e^{-2bT}\right)$$
$$\cdot \mathbb{E}_{X_0 \sim \rho_0}\left[G(1, X_0)\log\left(1 + \frac{(1 \vee L/b)(d + M + R^2 e^{-2bT})}{G(1, X_0)}\right)\right].$$

Let us note that the map

$$u \mapsto u\log\left(1 + \frac{(1 \vee L/b)(d + M + R^2 e^{-2bT})}{u}\right)$$

is concave on $\mathbb{R}_+$. Then, due to Jensen's inequality, we obtain that

$$\mathbb{E}_{X_0 \sim \rho_0} \text{Var}\big(\varphi(X_T^*) - \varphi^*(X_T^*) \,|\, X_0\big)$$
$$\lesssim \mathbb{E}_{X_0 \sim \rho_0} G(1, X_0) + \left(1 \vee \frac{L}{b}\right)\left(d + M + R^2 e^{-2bT}\right)\mathbb{E}_{X_0 \sim \rho_0} G(1, X_0)$$
$$\cdot \log\left(1 + \frac{(1 \vee L/b)(d + M + R^2 e^{-2bT})}{\mathbb{E}_{X_0 \sim \rho_0} G(1, X_0)}\right)$$
$$= \text{KL}\,(\pi^*, \pi^\varphi) + \left(1 \vee \frac{L}{b}\right)\left(d + M + R^2 e^{-2bT}\right)\text{KL}\,(\pi^*, \pi^\varphi)$$
$$\cdot \log\left(1 + \frac{(1 \vee L/b)(d + M + R^2 e^{-2bT})}{\text{KL}\,(\pi^*, \pi^\varphi)}\right).$$

$\square$

## C.2 PROOF OF LEMMA C.2

Let us introduce

$$\widetilde{\varphi}(x) = \varphi(x) - \mathbb{E}\varphi(X_T^*) \quad \text{and} \quad \widetilde{\varphi}^*(x) = \varphi^*(x) - \mathbb{E}\varphi^*(X_T^*).$$

Both $\widetilde{\varphi}(x)$ and $\widetilde{\varphi}^*(x)$ exhibit sub-quadratic growth. Indeed, due to Assumption 4, it holds that

$$-M \leqslant -\mathbb{E}\varphi(X_T^*) \leqslant L\mathbb{E}\left\|\Sigma^{-1/2}(X_T^* - m)\right\|^2 + M$$
$$= L\text{Tr}\big(\Sigma^{-1}\text{Var}(X_T^*)\big) + L\left\|\Sigma^{-1/2}(\mathbb{E}X_T^* - m)\right\|^2 + M$$

and, similarly,

$$-M \leqslant -\mathbb{E}\varphi^*(X_T^*) \leqslant L\mathbb{E}\left\|\Sigma^{-1/2}(X_T^* - m)\right\|^2 + M$$
$$= L\text{Tr}\big(\Sigma^{-1}\text{Var}(X_T^*)\big) + L\left\|\Sigma^{-1/2}(\mathbb{E}X_T^* - m)\right\|^2 + M.$$

Recalling that (see (32))

$$\widetilde{M} = L\text{Tr}\big(\Sigma^{-1}\text{Var}(X_T^*)\big) + L\left\|\Sigma^{-1/2}(\mathbb{E}X_T^* - m)\right\|^2 + 4M$$

and taking into account Assumption 4, we obtain that

$$|\widetilde{\varphi}(x) - \widetilde{\varphi}^*(x)| \leqslant L\left\|\Sigma^{-1/2}(\mathbb{E}X_T^* - m)\right\|^2 + \widetilde{M}.$$

At the same time, it is straightforward to observe that
$$\mathrm{Var}\left(\mathbb{E}\left[\varphi(X_T^*) - \varphi^*(X_T^*)\,\middle|\, X_0\right]\right) = \mathrm{Var}\left(\mathbb{E}\left[\widetilde{\varphi}(X_T^*) - \widetilde{\varphi}^*(X_T^*)\,\middle|\, X_0\right]\right)$$
and
$$\mathrm{Var}\left(\varphi(X_T^*) - \varphi^*(X_T^*)\right) = \mathrm{Var}\left(\widetilde{\varphi}(X_T^*) - \widetilde{\varphi}^*(X_T^*)\right) = \mathbb{E}\left(\widetilde{\varphi}(X_T^*) - \widetilde{\varphi}^*(X_T^*)\right)^2.$$

For this reason, it is enough to check that
$$\mathrm{Var}\left(\mathbb{E}\left[\widetilde{\varphi}(X_T^*) - \widetilde{\varphi}^*(X_T^*)\,\middle|\, X_0\right]\right)$$
$$\lesssim de^{-2bT}\left((M^2 \vee 1)\mathrm{Var}\left(\varphi(X_T^*) - \varphi^*(X_T^*)\right)^{1/\mathcal{K}^2(T)} + \mathrm{Var}\left(\varphi(X_T^*) - \varphi^*(X_T^*)\right)^{2/\mathcal{K}^2(T)-1}\right)$$
$$\cdot \exp\left\{\mathcal{O}\left(\sqrt{d}\left(M \vee 1 \vee \log\left(\widetilde{M} \vee LR^2 \vee (Ld^2/b)\right)\right)e^{-bT}\right)\right\}.$$

Let us note that the conditional density of $X_T^*$ given $X_0 = x_0$ is equal to
$$\pi^*(x_T \mid x_0) = \frac{e^{\varphi^*(x_T)}\pi^0(x_T \mid x_0)}{\mathcal{T}_T[e^{\varphi^*}](x_0)}$$

This yields that
$$\mathbb{E}\left[\widetilde{\varphi}(X_T^*) - \widetilde{\varphi}^*(X_T^*)\,\middle|\, X_0\right] = \frac{1}{\mathcal{T}_T[e^{\varphi^*}](x_0)}\int_{\mathbb{R}^d}\left(\widetilde{\varphi}(x_T) - \widetilde{\varphi}^*(x_T)\right)e^{\varphi^*(x_T)}\pi^0(x_T \mid x_0)\,\mathrm{d}x_T$$
$$= \frac{\mathcal{T}_T\left[(\widetilde{\varphi} - \widetilde{\varphi}^*)e^{\varphi^*}\right](X_0)}{\mathcal{T}_T\left[e^{\varphi^*}\right](X_0)}.$$

Let us introduce
$$(\widetilde{\varphi} - \widetilde{\varphi}^*)_+ = 0 \vee (\widetilde{\varphi} - \widetilde{\varphi}^*), \quad \text{and} \quad (\widetilde{\varphi} - \widetilde{\varphi}^*)_- = 0 \vee (\widetilde{\varphi}^* - \widetilde{\varphi}).$$

Then we can represent $(\widetilde{\varphi} - \widetilde{\varphi}^*)$ as a difference of two non-negative functions:
$$\widetilde{\varphi} - \widetilde{\varphi}^* = (\widetilde{\varphi} - \widetilde{\varphi}^*)_+ - (\widetilde{\varphi} - \widetilde{\varphi}^*)_-.$$

As a consequence, we have
$$\mathbb{E}\left[\widetilde{\varphi}(X_T^*) - \widetilde{\varphi}^*(X_T^*)\,\middle|\, X_0\right] = \frac{\mathcal{T}_T\left[(\widetilde{\varphi} - \widetilde{\varphi}^*)_+ e^{\varphi^*}\right](X_0)}{\mathcal{T}_T\left[e^{\varphi^*}\right](X_0)} - \frac{\mathcal{T}_T\left[(\widetilde{\varphi} - \widetilde{\varphi}^*)_- e^{\varphi^*}\right](X_0)}{\mathcal{T}_T\left[e^{\varphi^*}\right](X_0)}.$$

Applying Young's inequality, we obtain that
$$\mathrm{Var}\left(\mathbb{E}\left[\widetilde{\varphi}(X_T^*) - \widetilde{\varphi}^*(X_T^*)\,\middle|\, X_0\right]\right) \leqslant 2\mathrm{Var}\left(\frac{\mathcal{T}_T\left[(\widetilde{\varphi} - \widetilde{\varphi}^*)_+ e^{\varphi^*}\right](X_0)}{\mathcal{T}_T\left[e^{\varphi^*}\right](X_0)}\right)$$
$$+ 2\mathrm{Var}\left(\frac{\mathcal{T}_T\left[(\widetilde{\varphi} - \widetilde{\varphi}^*)_- e^{\varphi^*}\right](X_0)}{\mathcal{T}_T\left[e^{\varphi^*}\right](X_0)}\right).$$

An upper bound on the terms in the right-hand side easily follows from Lemma C.3. Indeed, introducing
$$\varphi_{\max} = \widetilde{M}e^M + 2Le^M R^2 + \frac{2Le^M}{b}(400d + d^2)$$
and applying Lemma C.3, we obtain that
$$\mathrm{Var}\left(\frac{\mathcal{T}_T[(\widetilde{\varphi} - \widetilde{\varphi}^*)_+ e^{\varphi^*}](X_0)}{\mathcal{T}_T[e^{\varphi^*}](X_0)}\right) \leqslant \varphi_{\max}^2\left(\frac{\mathcal{T}_\infty[(\widetilde{\varphi} - \widetilde{\varphi}^*)_+ e^{\varphi^*}]}{\varphi_{\max}\mathcal{T}_\infty[e^{\varphi^*}]}\right)^{2/\mathcal{K}(T)}$$
$$\cdot \exp\left\{2M(\mathcal{K}(T) - 1) + 2\overline{\mathcal{A}}(T)\left(\mathcal{K}(T) + \frac{1}{\mathcal{K}(T)}\right)\right\}$$
$$\cdot \left\{M^2\left(\mathcal{K}(T) - \frac{1}{\mathcal{K}(T)}\right)^2 + 2(\overline{\mathcal{A}}(T))^2\left(\mathcal{K}(T) + \frac{1}{\mathcal{K}(T)}\right)^2\right\}$$
$$+ \left(\mathcal{K}(T) - \frac{1}{\mathcal{K}(T)}\right)^2\left(\frac{\mathcal{T}_\infty[(\widetilde{\varphi} - \widetilde{\varphi}^*)_+ e^{\varphi^*}]}{\varphi_{\max}\mathcal{T}_\infty[e^{\varphi^*}]}\right)^{4/\mathcal{K}(T)-2\mathcal{K}(T)}$$
$$\cdot \frac{\varphi_{\max}^2}{e^2}\cdot\exp\left\{2M(\mathcal{K}(T) - 1) + 2\overline{\mathcal{A}}(T)\left(\mathcal{K}(T) + \frac{1}{\mathcal{K}(T)}\right)\right\},$$

where $\overline{\mathcal{A}}(t)$ and $\mathcal{K}(t)$ are defined in (33) and (39), respectively. Similarly, we have

$$
\mathrm{Var}\left(\frac{\mathcal{T}_T[(\widetilde{\varphi} - \widetilde{\varphi}^*)_- e^{\varphi^*}](X_0)}{\mathcal{T}_T[e^{\varphi^*}](X_0)}\right)
$$

$$
\leqslant \varphi_{\max}^2 \left(\frac{\mathcal{T}_\infty[(\widetilde{\varphi} - \widetilde{\varphi}^*)_- e^{\varphi^*}]}{\varphi_{\max} \mathcal{T}_\infty[e^{\varphi^*}]}\right)^{2/\mathcal{K}(T)} \exp\left\{2M(\mathcal{K}(T) - 1) + 2\overline{\mathcal{A}}(T)\left(\mathcal{K}(T) + \frac{1}{\mathcal{K}(T)}\right)\right\}
$$

$$
\cdot \left\{M^2\left(\mathcal{K}(T) - \frac{1}{\mathcal{K}(T)}\right)^2 + 2(\overline{\mathcal{A}}(T))^2\left(\mathcal{K}(T) + \frac{1}{\mathcal{K}(T)}\right)^2\right\}
$$

$$
+ \frac{\varphi_{\max}^2}{e^2}\left(\mathcal{K}(T) - \frac{1}{\mathcal{K}(T)}\right)^2 \left(\frac{\mathcal{T}_\infty[(\widetilde{\varphi} - \widetilde{\varphi}^*)_- e^{\varphi^*}]}{\varphi_{\max} \mathcal{T}_\infty[e^{\varphi^*}]}\right)^{4/\mathcal{K}(T) - 2\mathcal{K}(T)}
$$

$$
\cdot \exp\left\{2M(\mathcal{K}(T) - 1) + 2\overline{\mathcal{A}}(T)\left(\mathcal{K}(T) + \frac{1}{\mathcal{K}(T)}\right)\right\},
$$

This yields that

$$
\mathrm{Var}\left(\mathbb{E}\left[\widetilde{\varphi}(X_T^*) - \widetilde{\varphi}^*(X_T^*) \,|\, X_0\right]\right)
$$

$$
\leqslant 2\varphi_{\max}^2 \left(\frac{\mathcal{T}_\infty[|\widetilde{\varphi} - \widetilde{\varphi}^*| e^{\varphi^*}]}{\varphi_{\max} \mathcal{T}_\infty[e^{\varphi^*}]}\right)^{2/\mathcal{K}(T)} \exp\left\{2M(\mathcal{K}(T) - 1) + 2\overline{\mathcal{A}}(T)\left(\mathcal{K}(T) + \frac{1}{\mathcal{K}(T)}\right)\right\}
$$

$$
\cdot \left\{M^2\left(\mathcal{K}(T) - \frac{1}{\mathcal{K}(T)}\right)^2 + 2(\overline{\mathcal{A}}(T))^2\left(\mathcal{K}(T) + \frac{1}{\mathcal{K}(T)}\right)^2\right\}
$$

$$
+ \frac{2\varphi_{\max}^2}{e^2}\left(\mathcal{K}(T) - \frac{1}{\mathcal{K}(T)}\right)^2 \left(\frac{\mathcal{T}_\infty[|\widetilde{\varphi} - \widetilde{\varphi}^*| e^{\varphi^*}]}{\varphi_{\max} \mathcal{T}_\infty[e^{\varphi^*}]}\right)^{4/\mathcal{K}(T) - 2\mathcal{K}(T)}
$$

$$
\cdot \exp\left\{2M(\mathcal{K}(T) - 1) + 2\overline{\mathcal{A}}(T)\left(\mathcal{K}(T) + \frac{1}{\mathcal{K}(T)}\right)\right\}.
$$

Here we used the fact that both $\mathcal{T}_\infty[(\widetilde{\varphi} - \widetilde{\varphi}^*)_+ e^{\varphi^*}]$ and $\mathcal{T}_\infty[(\widetilde{\varphi} - \widetilde{\varphi}^*)_- e^{\varphi^*}]$ do not exceed $\mathcal{T}_\infty[|\widetilde{\varphi} - \widetilde{\varphi}^*| e^{\varphi^*}]$ and that $4/\mathcal{K}(T) - 2\mathcal{K}(T) \geqslant 0$ due to the conditions of the lemma. On the other hand, it holds that

$$
\mathrm{Var}\left(\varphi(X_T^*) - \varphi^*(X_T^*)\right) = \mathrm{Var}\left(\widetilde{\varphi}(X_T^*) - \widetilde{\varphi}^*(X_T^*)\right) = \mathbb{E}\left(\widetilde{\varphi}(X_T^*) - \widetilde{\varphi}^*(X_T^*)\right)^2.
$$

Due to the tower rule, we have

$$
\mathbb{E}\left(\widetilde{\varphi}(X_T^*) - \widetilde{\varphi}^*(X_T^*)\right)^2 = \mathbb{E}\,\mathbb{E}\left[\left(\widetilde{\varphi}(X_T^*) - \widetilde{\varphi}^*(X_T^*)\right)^2 \,\big|\, X_0\right] = \mathbb{E}\,\frac{\mathcal{T}_T[(\widetilde{\varphi} - \widetilde{\varphi}^*)^2 e^{\varphi^*}](X_0)}{\mathcal{T}_T[e^{\varphi^*}](X_0)}.
$$

Applying Lemma E.10 and using the bound

$$
(\widetilde{\varphi}(x) - \widetilde{\varphi}^*(x))^2 \leqslant \left(L\left\|\Sigma^{-1/2}(\mathbb{E}X_T^* - m)\right\|^2 + \widetilde{M}\right)^2 \leqslant 2L^2\left\|\Sigma^{-1/2}(\mathbb{E}X_T^* - m)\right\|^4 + 2\widetilde{M}^2,
$$

we obtain that

$$
\mathrm{Var}\left(\varphi(X_T^*) - \varphi^*(X_T^*)\right)
$$

$$
= \mathbb{E}\,\frac{\mathcal{T}_T[(\widetilde{\varphi} - \widetilde{\varphi}^*)^2 e^{\varphi^*}](X_0)}{\mathcal{T}_T[e^{\varphi^*}](X_0)}
$$

$$
\geqslant \left(\frac{\mathcal{T}_\infty[(\widetilde{\varphi} - \widetilde{\varphi}^*)^2 e^{\varphi^*}]}{\mathcal{T}_\infty[e^{\varphi^*}]}\right)^{\mathcal{K}(T)}
$$

$$
\cdot \mathbb{E}\left[(H(X_0))^{1 - \mathcal{K}(T)} \exp\left\{-M\left(1 - \frac{1}{\mathcal{K}(T)}\right) - \mathcal{A}(X_0, T)\left(\mathcal{K}(T) + \frac{1}{\mathcal{K}(T)}\right)\right\}\right],
$$

where we introduced

$$
H(x) = 2\widetilde{M}^2 e^M + 16L^2 e^M \left\|\Sigma^{-1/2}(x - m)\right\|^4 + \frac{32L^2 e^M}{b^2}\left(2560000d^2 + d^4\right).
$$

Let

$$H_{\max} = 2\widetilde{M}^2 e^M + 16L^2 e^M R^4 + \frac{32L^2 e^M}{b^2}\left(2560000 d^2 + d^4\right)$$

and note that

$$H(x) \leqslant H_{\max} \quad \text{and} \quad \mathcal{A}(x,t) \leqslant \overline{\mathcal{A}}(t) \quad \text{for all } x \in \mathrm{supp}(\rho_0).$$

This yields that

$$\mathrm{Var}\big(\varphi(X_T^*) - \varphi^*(X_T^*)\big) \geqslant H_{\max}^{1-\mathcal{K}(T)}\left(\frac{\mathcal{T}_\infty[(\widetilde{\varphi} - \widetilde{\varphi}^*)^2 e^{\varphi^*}]}{\mathcal{T}_\infty[e^{\varphi^*}]}\right)^{\mathcal{K}(T)}$$
$$\cdot \exp\left\{-M\left(1 - \frac{1}{\mathcal{K}(T)}\right) - \overline{\mathcal{A}}(T)\left(\mathcal{K}(T) + \frac{1}{\mathcal{K}(T)}\right)\right\},$$

Applying the Cauchy-Schwarz inequality, we obtain that

$$\left(\frac{\mathcal{T}_\infty[|\widetilde{\varphi} - \widetilde{\varphi}^*| e^{\varphi^*}]}{\mathcal{T}_\infty[e^{\varphi^*}]}\right)^{2\mathcal{K}(T)} \leqslant \left(\frac{\mathcal{T}_\infty[(\widetilde{\varphi} - \widetilde{\varphi}^*)^2 e^{\varphi^*}]}{\mathcal{T}_\infty[e^{\varphi^*}]}\right)^{\mathcal{K}(T)}$$
$$\leqslant H_{\max}^{\mathcal{K}(T)-1}\,\mathrm{Var}\big(\varphi(X_T^*) - \varphi^*(X_T^*)\big)$$
$$\cdot \exp\left\{M\left(1 - \frac{1}{\mathcal{K}(T)}\right) + \overline{\mathcal{A}}(T)\left(\mathcal{K}(T) + \frac{1}{\mathcal{K}(T)}\right)\right\}.$$

This allows us to conclude that

$$\mathrm{Var}\left(\mathbb{E}\left[\widetilde{\varphi}(X_T^*) - \widetilde{\varphi}^*(X_T^*)\,\big|\,X_0\right]\right)$$
$$\leqslant 2\varphi_{\max}^{2-2/\mathcal{K}(T)} H_{\max}^{1/\mathcal{K}(T)-1/\mathcal{K}^2(T)}\,\mathrm{Var}\big(\varphi(X_T^*) - \varphi^*(X_T^*)\big)^{1/\mathcal{K}^2(T)}$$
$$\cdot \exp\left\{M\left(2 + \frac{1}{\mathcal{K}^3(T)}\right)(\mathcal{K}(T) - 1) + \overline{\mathcal{A}}(T)\left(2 + \frac{1}{\mathcal{K}^3(T)}\right)\left(\mathcal{K}(T) + \frac{1}{\mathcal{K}(T)}\right)\right\}$$
$$\cdot \left\{M^2\left(\mathcal{K}(T) - \frac{1}{\mathcal{K}(T)}\right)^2 + 2(\overline{\mathcal{A}}(T))^2\left(\mathcal{K}(T) + \frac{1}{\mathcal{K}(T)}\right)^2\right\}$$
$$+ \frac{2\varphi_{\max}^{2+2\mathcal{K}(T)-4/\mathcal{K}(T)} H_{\max}^{(\mathcal{K}(T)-1)(2/\mathcal{K}^2(T)-1)}}{e^2}$$
$$\cdot \left(\mathcal{K}(T) - \frac{1}{\mathcal{K}(T)}\right)^2 \mathrm{Var}\big(\varphi(X_T^*) - \varphi^*(X_T^*)\big)^{2/\mathcal{K}^2(T)-1}$$
$$\cdot \exp\left\{M\left(2 + \frac{2}{\mathcal{K}^3(T)} - \frac{1}{\mathcal{K}(T)}\right)(\mathcal{K}(T) - 1) + \overline{\mathcal{A}}(T)\left(1 + \frac{2}{\mathcal{K}^2(T)}\right)\left(\mathcal{K}(T) + \frac{1}{\mathcal{K}(T)}\right)\right\}.$$

It remains to simplify the expression in the right-hand side relying on the definitions of $\varphi_{\max}$, $H_{\max}$, $\overline{\mathcal{A}}(t)$, and $\mathcal{K}(t)$. First, note that

$$\max\left\{\log\varphi_{\max}, \log H_{\max}\right\} \lesssim M + \log\left(\widetilde{M} \vee LR^2 \vee (Ld^2/b)\right).$$

Second, the relations

$$0 \leqslant \mathcal{K}(T) - 1 \lesssim \sqrt{d}e^{-bT} \quad \text{and} \quad 0 \leqslant \overline{\mathcal{A}}(T) \lesssim \left(\frac{R^2}{\sqrt{d}} + \sqrt{d}\right)e^{-bT}$$

yield that

$$\mathrm{Var}\left(\mathbb{E}\left[\widetilde{\varphi}(X_T^*) - \widetilde{\varphi}^*(X_T^*)\,\big|\,X_0\right]\right) \lesssim \varphi_{\max}^{\mathcal{O}(\sqrt{d}e^{-bT})} H_{\max}^{\mathcal{O}(\sqrt{d}e^{-bT})}\exp\left\{\mathcal{O}\left(\sqrt{d}(M\vee 1)e^{-bT}\right)\right\}$$
$$\cdot \left(d(M^2 \vee 1)e^{-2bT}\right)\mathrm{Var}\big(\varphi(X_T^*) - \varphi^*(X_T^*)\big)^{1/\mathcal{K}^2(T)}$$
$$+ \varphi_{\max}^{\mathcal{O}(\sqrt{d}e^{-bT})} H_{\max}^{\mathcal{O}(\sqrt{d}e^{-bT})}\exp\left\{\mathcal{O}\left(\sqrt{d}(M\vee 1)e^{-bT}\right)\right\}$$
$$\cdot de^{-2bT}\mathrm{Var}\big(\varphi(X_T^*) - \varphi^*(X_T^*)\big)^{2/\mathcal{K}^2(T)-1}.$$

Applying the inequality

$$\varphi_{\max}^{\mathcal{O}(\sqrt{d}e^{-bT})} H_{\max}^{\mathcal{O}(\sqrt{d}e^{-bT})} \exp\left\{\mathcal{O}\left(\sqrt{d}(M \vee 1)e^{-bT}\right)\right\}$$
$$\leqslant \exp\left\{\mathcal{O}\left(\sqrt{d}\left(M \vee 1 \vee \log\left(\widetilde{M} \vee LR^2 \vee (Ld^2/b)\right)\right)e^{-bT}\right)\right\},$$

we finally obtain that

$$\mathrm{Var}\left(\mathbb{E}\left[\widetilde{\varphi}(X_T^*) - \widetilde{\varphi}^*(X_T^*) \mid X_0\right]\right)$$
$$\lesssim de^{-2bT}\left((M^2 \vee 1)\mathrm{Var}\left(\varphi(X_T^*) - \varphi^*(X_T^*)\right)^{1/\mathcal{K}^2(T)} + \mathrm{Var}\left(\varphi(X_T^*) - \varphi^*(X_T^*)\right)^{2/\mathcal{K}^2(T)-1}\right)$$
$$\cdot \exp\left\{\mathcal{O}\left(\sqrt{d}\left(M \vee 1 \vee \log\left(\widetilde{M} \vee LR^2 \vee (Ld^2/b)\right)\right)e^{-bT}\right)\right\}.$$

The proof is finished.

$\square$

## C.3 Proof of Lemma C.3

Let us define

$$F(x) = Be^M + 2Ae^M \left\|\Sigma^{-1/2}(x-m)\right\|^2 + \frac{2Ae^M}{b}\left(400d + d^2\right).$$

Then, due to Lemma E.10, for any $x \in \mathbb{R}^d$, it holds that

$$\frac{\mathcal{T}_T[fe^{\varphi^*}](x)}{\mathcal{T}_T[e^{\varphi^*}](x)} \leqslant \left(F(x)\right)^{1-1/\mathcal{K}(T)}\left(\frac{\mathcal{T}_\infty[fe^{\varphi^*}]}{\mathcal{T}_\infty[e^{\varphi^*}]}\right)^{1/\mathcal{K}(T)}$$
$$\cdot \exp\left\{M\left(\mathcal{K}(T)-1\right) + \mathcal{A}(x,T)\left(\mathcal{K}(T) + \frac{1}{\mathcal{K}(T)}\right)\right\}$$

and

$$\frac{\mathcal{T}_T[fe^{\varphi^*}](x)}{\mathcal{T}_T[e^{\varphi^*}](x)} \geqslant \left(F(x)\right)^{1-\mathcal{K}(T)}\left(\frac{\mathcal{T}_\infty[fe^{\varphi^*}]}{\mathcal{T}_\infty[e^{\varphi^*}]}\right)^{\mathcal{K}(T)}$$
$$\cdot \exp\left\{-M\left(1 - \frac{1}{\mathcal{K}(T)}\right) - \mathcal{A}(x,T)\left(\mathcal{K}(T) + \frac{1}{\mathcal{K}(T)}\right)\right\},$$

where the functions $\mathcal{A}(x,t)$ and $\mathcal{K}(t)$ are defined in (38) and (39), respectively. In view of Assumption 2, for any $x \in \mathrm{supp}(\rho_0)$, it holds that

$$\mathcal{A}(x,t) \leqslant \overline{\mathcal{A}}(t) \quad \text{and} \quad F(x) \leqslant F_{\max}.$$

This yields that

$$\frac{\mathcal{T}_T[fe^{\varphi^*}](X_0)}{\mathcal{T}_T[e^{\varphi^*}](X_0)} \leqslant \left(F_{\max}\right)^{1-1/\mathcal{K}(T)}\left(\frac{\mathcal{T}_\infty[fe^{\varphi^*}]}{\mathcal{T}_\infty[e^{\varphi^*}]}\right)^{1/\mathcal{K}(T)}$$
$$\cdot \exp\left\{M\left(\mathcal{K}(T)-1\right) + \overline{\mathcal{A}}(T)\left(\mathcal{K}(T) + \frac{1}{\mathcal{K}(T)}\right)\right\}$$

and

$$\frac{\mathcal{T}_T[fe^{\varphi^*}](X_0)}{\mathcal{T}_T[e^{\varphi^*}](X_0)} \geqslant \left(F_{\max}\right)^{1-\mathcal{K}(T)}\left(\frac{\mathcal{T}_\infty[fe^{\varphi^*}]}{\mathcal{T}_\infty[e^{\varphi^*}]}\right)^{\mathcal{K}(T)}$$
$$\cdot \exp\left\{-M\left(1 - \frac{1}{\mathcal{K}(T)}\right) - \overline{\mathcal{A}}(T)\left(\mathcal{K}(T) + \frac{1}{\mathcal{K}(T)}\right)\right\}$$

almost surely. Then the variance of $\mathcal{T}_T[fe^{\varphi^*}](X_0)/\mathcal{T}_T[e^{\varphi^*}](X_0)$ does not exceed

$$\operatorname{Var}\left(\frac{\mathcal{T}_T[fe^{\varphi^*}](X_0)}{\mathcal{T}_T[e^{\varphi^*}](X_0)}\right) \leqslant \frac{1}{4}\left((F_{\max})^{1-1/\mathcal{K}(T)}\left(\frac{\mathcal{T}_\infty[fe^{\varphi^*}]}{\mathcal{T}_\infty[e^{\varphi^*}]}\right)^{1/\mathcal{K}(T)}\right.$$

$$\cdot \exp\left\{M(\mathcal{K}(T)-1) + \overline{\mathcal{A}}(T)\left(\mathcal{K}(T) + \frac{1}{\mathcal{K}(T)}\right)\right\}$$

$$- (F_{\max})^{1-\mathcal{K}(T)}\left(\frac{\mathcal{T}_\infty[fe^{\varphi^*}]}{\mathcal{T}_\infty[e^{\varphi^*}]}\right)^{\mathcal{K}(T)}$$

$$\left.\cdot \exp\left\{-M\left(1 - \frac{1}{\mathcal{K}(T)}\right) - \overline{\mathcal{A}}(T)\left(\mathcal{K}(T) + \frac{1}{\mathcal{K}(T)}\right)\right\}\right)^2.$$

Here we used the fact that if a random variable $\xi$ takes values in $[a, b]$ almost surely, then its variance is not greater than $(b-a)^2/4$. Simplifying the expression in the right-hand side, we obtain that

$$\operatorname{Var}\left(\frac{\mathcal{T}_T[fe^{\varphi^*}](X_0)}{\mathcal{T}_T[e^{\varphi^*}](X_0)}\right)$$

$$\leqslant \frac{F_{\max}^2}{4}\left(\frac{\mathcal{T}_\infty[fe^{\varphi^*}]}{F_{\max}\mathcal{T}_\infty[e^{\varphi^*}]}\right)^{2/\mathcal{K}(T)}\exp\left\{2M(\mathcal{K}(T)-1) + 2\overline{\mathcal{A}}(T)\left(\mathcal{K}(T) + \frac{1}{\mathcal{K}(T)}\right)\right\}$$

$$\cdot\left(1 - \exp\left\{-\left(\mathcal{K}(T) - \frac{1}{\mathcal{K}(T)}\right)\left(M + \log\frac{F_{\max}\mathcal{T}_\infty[e^{\varphi^*}]}{\mathcal{T}_\infty[fe^{\varphi^*}]}\right) - 2\overline{\mathcal{A}}(T)\left(\mathcal{K}(T) + \frac{1}{\mathcal{K}(T)}\right)\right\}\right)^2.$$

Taking into account that $0 \leqslant 1 - e^{-v} \leqslant v$ for all $v \geqslant 0$ and the inequalities

$$\mathcal{T}_\infty[fe^{\varphi^*}] \leqslant F_{\max}, \quad \mathcal{T}_\infty[e^{\varphi^*}] \geqslant 1, \quad \text{and} \quad \overline{\mathcal{A}}(T) \geqslant 0,$$

we deduce that

$$\operatorname{Var}\left(\frac{\mathcal{T}_T[fe^{\varphi^*}](X_0)}{\mathcal{T}_T[e^{\varphi^*}](X_0)}\right)$$

$$\leqslant \frac{F_{\max}^2}{4}\left(\frac{\mathcal{T}_\infty[fe^{\varphi^*}]}{F_{\max}\mathcal{T}_\infty[e^{\varphi^*}]}\right)^{2/\mathcal{K}(T)}\exp\left\{2M(\mathcal{K}(T)-1) + 2\overline{\mathcal{A}}(T)\left(\mathcal{K}(T) + \frac{1}{\mathcal{K}(T)}\right)\right\}$$

$$\cdot\left[\left(\mathcal{K}(T) - \frac{1}{\mathcal{K}(T)}\right)\left(M + \log\frac{F_{\max}\mathcal{T}_\infty[e^{\varphi^*}]}{\mathcal{T}_\infty[fe^{\varphi^*}]}\right) + 2\overline{\mathcal{A}}(T)\left(\mathcal{K}(T) + \frac{1}{\mathcal{K}(T)}\right)\right]^2.$$

The Cauchy-Schwarz inequality $(a + b + c)^2 \leqslant 4a^2 + 4b^2 + 2c^2$ implies that

$$\operatorname{Var}\left(\frac{\mathcal{T}_T[fe^{\varphi^*}](X_0)}{\mathcal{T}_T[e^{\varphi^*}](X_0)}\right)$$

$$\leqslant F_{\max}^2\left(\frac{\mathcal{T}_\infty[fe^{\varphi^*}]}{F_{\max}\mathcal{T}_\infty[e^{\varphi^*}]}\right)^{2/\mathcal{K}(T)}\exp\left\{2M(\mathcal{K}(T)-1) + 2\overline{\mathcal{A}}(T)\left(\mathcal{K}(T) + \frac{1}{\mathcal{K}(T)}\right)\right\}$$

$$\cdot\left\{\left(\mathcal{K}(T) - \frac{1}{\mathcal{K}(T)}\right)^2\left[M^2 + \log^2\left(\frac{F_{\max}\mathcal{T}_\infty[e^{\varphi^*}]}{\mathcal{T}_\infty[fe^{\varphi^*}]}\right)\right] + 2(\overline{\mathcal{A}}(T))^2\left(\mathcal{K}(T) + \frac{1}{\mathcal{K}(T)}\right)^2\right\}.$$

Finally, taking into account that

$$\frac{F_{\max}\mathcal{T}_\infty[e^{\varphi^*}]}{\mathcal{T}_\infty[fe^{\varphi^*}]} \geqslant 1$$

and using the inequality

$$\frac{\log u}{u} \leqslant \frac{1}{e} \quad \text{for all } u \geqslant 1,$$

we obtain that

$$0 \leqslant \left(\mathcal{K}(T) - \frac{1}{\mathcal{K}(T)}\right)\log\left(\frac{F_{\max}\mathcal{T}_\infty[e^{\varphi^*}]}{\mathcal{T}_\infty[fe^{\varphi^*}]}\right) \leqslant \frac{1}{e}\left(\frac{F_{\max}\mathcal{T}_\infty[e^{\varphi^*}]}{\mathcal{T}_\infty[fe^{\varphi^*}]}\right)^{\mathcal{K}(T)-1/\mathcal{K}(T)}.$$

Hence, it holds that

$$
\begin{aligned}
\mathrm{Var}\left(\frac{\mathcal{T}_T[fe^{\varphi^*}](X_0)}{\mathcal{T}_T[e^{\varphi^*}](X_0)}\right) &\leqslant F_{\max}^2 \left(\frac{\mathcal{T}_\infty[fe^{\varphi^*}]}{F_{\max}\mathcal{T}_\infty[e^{\varphi^*}]}\right)^{2/\mathcal{K}(T)} \\
&\quad \cdot \exp\left\{2M\big(\mathcal{K}(T)-1\big) + 2\overline{\mathcal{A}}(T)\left(\mathcal{K}(T)+\frac{1}{\mathcal{K}(T)}\right)\right\} \\
&\quad \cdot \left\{M^2\left(\mathcal{K}(T)-\frac{1}{\mathcal{K}(T)}\right)^2 + 2\big(\overline{\mathcal{A}}(T)\big)^2\left(\mathcal{K}(T)+\frac{1}{\mathcal{K}(T)}\right)^2\right\} \\
&\quad + \frac{F_{\max}^2}{e^2}\left(\mathcal{K}(T)-\frac{1}{\mathcal{K}(T)}\right)^2\left(\frac{\mathcal{T}_\infty[fe^{\varphi^*}]}{F_{\max}\mathcal{T}_\infty[e^{\varphi^*}]}\right)^{4/\mathcal{K}(T)-2\mathcal{K}(T)} \\
&\quad \cdot \exp\left\{2M\big(\mathcal{K}(T)-1\big) + 2\overline{\mathcal{A}}(T)\left(\mathcal{K}(T)+\frac{1}{\mathcal{K}(T)}\right)\right\}.
\end{aligned}
$$

$\square$

## C.4 PROOF OF LEMMA C.4

Due to Young's inequality, we have

$$
\begin{aligned}
\mathrm{Var}\left(\log\frac{\mathcal{T}_T[e^\varphi](X_0)}{\mathcal{T}_T[e^{\varphi^*}](X_0)}\right) &= \mathrm{Var}\left(\zeta + \mathbb{E}\left[\varphi(X_T^*)-\varphi^*(X_T^*)\,\big|\,X_0\right]\right) \\
&\leqslant 2\mathrm{Var}(\zeta) + 2\mathrm{Var}\left(\mathbb{E}\left[\varphi(X_T^*)-\varphi^*(X_T^*)\,\big|\,X_0\right]\right).
\end{aligned}
$$

Thus, it remains to prove that

$$
\mathrm{Var}(\zeta) \leqslant 2e^{3/2}\,\mathsf{KL}(\pi^*,\pi^\varphi)\|\zeta\|_{\psi_1}\left(1+\frac{1}{2}\log\frac{2e\|\zeta\|_{\psi_1}}{\mathsf{KL}(\pi^*,\pi^\varphi)}\right).
$$

For this purpose, let us recall that

$$
\pi^*(x_T\,|\,x_0) = \frac{e^{\varphi^*(x_T)}}{\mathcal{T}_T[e^{\varphi^*}](x_0)}\,\pi^0(x_T\,|\,x_0)
$$

is the density of $X_T^*$ given $X_0 = x_0$. This allows us to rewrite $\log\big(\mathcal{T}_T[e^\varphi](X_0)/\mathcal{T}_T[e^{\varphi^*}](X_0)\big)$ in the following form:

$$
\log\frac{\mathcal{T}_T[e^\varphi](X_0)}{\mathcal{T}_T[e^{\varphi^*}](X_0)} = \log\mathbb{E}\left[e^{\varphi(X_T^*)-\varphi^*(X_T^*)}\,\big|\,X_0\right].
$$

This yields that $\zeta \geqslant 0$ almost surely. Indeed, according to Jensen's inequality, it holds that

$$
\begin{aligned}
\zeta &= \log\mathbb{E}\left[e^{\varphi(X_T^*)-\varphi^*(X_T^*)}\,\big|\,X_0\right] - \mathbb{E}\left[\varphi(X_T^*)-\varphi^*(X_T^*)\,\big|\,X_0\right] \\
&\geqslant \log e^{\mathbb{E}[\varphi(X_T^*)-\varphi^*(X_T^*)\,|\,X_0]} - \mathbb{E}\left[\varphi(X_T^*)-\varphi^*(X_T^*)\,\big|\,X_0\right] = 0 \quad\text{almost surely.}
\end{aligned}
$$

Furthermore, is it straightforward to check that

$$
\mathbb{E}\zeta = \mathbb{E}\log\frac{\mathcal{T}_T[e^\varphi](X_0)}{\mathcal{T}_T[e^{\varphi^*}](X_0)} - \mathbb{E}\big(\varphi(X_T^*)-\varphi^*(X_T^*)\big) = \mathsf{KL}(\pi^*,\pi^\varphi).
$$

Let us take

$$
\varepsilon = \left(\log\frac{2e\|\zeta\|_{\psi_1}}{\mathsf{KL}(\pi^*,\pi^\varphi)}\right)^{-1}.
$$

We would like to note that, according to Lemma E.8, $\mathsf{KL}(\pi^*,\pi^\varphi) = \mathbb{E}\zeta \leqslant 2\|\zeta\|_{\psi_1}$. This implies that $\varepsilon \in (0,1)$. Then, applying Hölder's inequality, we obtain that

$$
\mathrm{Var}(\zeta) \leqslant \mathbb{E}\zeta^2 = \mathbb{E}\big(\zeta^{1-\varepsilon}\cdot\zeta^{1+\varepsilon}\big) \leqslant (\mathbb{E}\zeta)^{1-\varepsilon}\left(\mathbb{E}\zeta^{(1+\varepsilon)/\varepsilon}\right)^\varepsilon = \mathsf{KL}(\pi^*,\pi^\varphi)^{1-\varepsilon}\left(\mathbb{E}\zeta^{(1+\varepsilon)/\varepsilon}\right)^\varepsilon.
$$

Let us elaborate on $\mathbb{E}\zeta^{(1+\varepsilon)/\varepsilon}$. Using Lemma E.8 again, we observe that the expectation of $\zeta^{(1+\varepsilon)/\varepsilon}$ does not exceed

$$\mathbb{E}\zeta^{(1+\varepsilon)/\varepsilon} \leqslant 2\left(2 + \frac{1}{\varepsilon}\right)^{(1+\varepsilon)/\varepsilon}\|\zeta\|_{\psi_1}^{(1+\varepsilon)/\varepsilon} = 2^{2+1/\varepsilon}\left(1 + \frac{1}{2\varepsilon}\right)^{1+1/\varepsilon}\|\zeta\|_{\psi_1}^{1+1/\varepsilon}.$$

This yields that

$$\left(\mathbb{E}\zeta^{(1+\varepsilon)/\varepsilon}\right)^{\varepsilon} \leqslant 2^{1+2\varepsilon}\left(1 + \frac{1}{2\varepsilon}\right)^{1+\varepsilon}\|\zeta\|_{\psi_1}^{1+\varepsilon} = \frac{1}{2}\left(1 + \frac{1}{2\varepsilon}\right)^{\varepsilon}\cdot\left(1 + \frac{1}{2\varepsilon}\right)(4\|\zeta\|_{\psi_1})^{1+\varepsilon}.$$

Since $1 + u \leqslant e^u$ for all $u \in \mathbb{R}$, we obtain that

$$\left(\mathbb{E}\zeta^{(1+\varepsilon)/\varepsilon}\right)^{\varepsilon} \leqslant \frac{\sqrt{e}}{2}\left(1 + \frac{1}{2\varepsilon}\right)(4\|\zeta\|_{\psi_1})^{1+\varepsilon}.$$

Hence, it holds that

$$\mathrm{Var}(\zeta) \leqslant \frac{\sqrt{e}}{2}\mathsf{KL}(\pi^*, \pi^{\varphi})^{1-\varepsilon}\left(\mathbb{E}\zeta^{(1+\varepsilon)/\varepsilon}\right)^{\varepsilon}\left(1 + \frac{1}{2\varepsilon}\right)(4\|\zeta\|_{\psi_1})^{1+\varepsilon}$$

$$= 2\sqrt{e}\,\mathsf{KL}(\pi^*, \pi^{\varphi})\|\zeta\|_{\psi_1}\left(1 + \frac{1}{2\varepsilon}\right)\left(\frac{4\|\zeta\|_{\psi_1}}{\mathsf{KL}(\pi^*, \pi^{\varphi})}\right)^{\varepsilon}.$$

Since

$$\frac{1}{\varepsilon} = \log\frac{2e\|\zeta\|_{\psi_1}}{\mathsf{KL}(\pi^*, \pi^{\varphi})} \geqslant \log\frac{4\|\zeta\|_{\psi_1}}{\mathsf{KL}(\pi^*, \pi^{\varphi})},$$

we deduce that

$$\left(\frac{4\|\zeta\|_{\psi_1}}{\mathsf{KL}(\pi^*, \pi^{\varphi})}\right)^{\varepsilon} \leqslant e.$$

This yields that

$$\mathrm{Var}(\zeta) \leqslant 2e^{3/2}\mathsf{KL}(\pi^*, \pi^{\varphi})\|\zeta\|_{\psi_1}\left(1 + \frac{1}{2}\log\frac{2e\|\zeta\|_{\psi_1}}{\mathsf{KL}(\pi^*, \pi^{\varphi})}\right),$$

and we finally obtain that

$$\mathrm{Var}\left(\log\frac{\mathcal{T}_T[e^{\varphi}](X_0)}{\mathcal{T}_T[e^{\varphi^*}](X_0)}\right) \leqslant 2\mathrm{Var}\left(\mathbb{E}\left[\varphi(X_T^*) - \varphi^*(X_T^*)\,\middle|\,X_0\right]\right)$$

$$+ 4e^{3/2}\mathsf{KL}(\pi^*, \pi^{\varphi})\|\zeta\|_{\psi_1}\left(1 + \frac{1}{2}\log\frac{2e\|\zeta\|_{\psi_1}}{\mathsf{KL}(\pi^*, \pi^{\varphi})}\right).$$

$\square$

# D  ORLICZ NORM BOUNDS

In this section, we present upper bounds on the $\psi_1$-norm of the log-potential difference $\varphi(X_T^*) - \varphi^*(X_T^*)$ as well as of $\log\left(\mathcal{T}_T[e^{\varphi}](X_0)/\mathcal{T}_T[e^{\varphi^*}](X_0)\right)$. This is a starting point in our analysis of the empirical risk minimizer $\widehat{\varphi}$ defined in (12). These results play a crucial role in the proof of Lemma E.1 about uniform concentration of the empirical means

$$\frac{1}{n}\sum_{i=1}^{n}\left(\varphi^*(Y_i) - \varphi(Y_i)\right) \quad\text{and}\quad \frac{1}{n}\sum_{j=1}^{n}\log\frac{\mathcal{T}_T[e^{\varphi}](Z_j)}{\mathcal{T}_T[e^{\varphi^*}](Z_j)}$$

around their expectation as they allow us to use Bernstein's inequality for sub-exponential random variables (Lecué & Mitchell, 2012, Proposition 5.2).

**Lemma D.1.** *Under Assumptions 1, 2, and 4, for any $T \gtrsim 1/b$ it holds that*

$$\|\varphi^*(X_T^*) - \varphi(X_T^*)\|_{\psi_1} \lesssim \left(1 \vee \frac{L}{b}\right)\left(M + d + e^{-bT}\left(\frac{R^2}{\sqrt{d}} + \sqrt{d} + bR^2\right)\right).$$

**Lemma D.2.** *Assume that log-potentials $\varphi \in \Phi$ and $\varphi^*$ satisfy Assumption 4. Suppose that $X_0 \sim \rho_0$, and $\rho_0$ fulfils Assumption 2. Then, for any $T > 0$, it holds that*

$$\left\| \log \frac{\mathcal{T}_T[e^\varphi](X_0)}{\mathcal{T}_T[e^{\varphi^*}](X_0)} \right\|_{\psi_1} \leqslant \left( \frac{be^2 R^2 \arcsin(e^{-bT})}{\sqrt{d} \log 2} + \frac{2 \log \mathcal{K}(T)}{\log 2} \right) \left( \frac{1}{\mathcal{K}(T)} + \mathcal{K}(T) \right) + \frac{M\mathcal{K}(T)}{\log 2}.$$

*where $\mathcal{K}(T) \geqslant 1$ is defined in (39).*

The proofs of Lemmata D.1 and D.2 are deferred to Appendices D.1 and D.2, respectively. We also provide an upper bound on the Orlicz norm of $\log \left( \pi^*(X_T^* \mid x_0) / \pi^\varphi(X_T^* \mid x_0) \right)$ with respect to the conditional density $\pi^*(\cdot \mid x_0)$). Unlike Lemmata D.1 and D.2, we use Lemma D.3 in verification of a Bernstein-type condition for the empirical KL-loss. To be more precise, we combine this result with Lemma B.1 to obtain an upper bound on the expected conditional variance $\mathbb{E}\mathrm{Var}\left( \varphi^*(X_T^*) - \varphi(X_T^*) \,\middle|\, X_0 \right)$ (see Lemma C.1 for the details).

**Lemma D.3.** *Grant Assumption 1 and suppose that the log-potential $\varphi^*$ corresponding to the optimal coupling $\pi^*$ satisfies*

$$\mathcal{T}_\infty \varphi^* = 0 \quad and \quad \varphi^*(x) \leqslant M \quad for\ all\ x \in \mathbb{R}^d.$$

*Let us take an arbitrary $\varphi$ satisfying Assumption 4. Then, for any $x_0 \in \mathrm{supp}(\rho_0)$, it holds that*

$$\left\| \log \frac{\pi^*(X_T^* \mid x_0)}{\pi^\varphi(X_T^* \mid x_0)} \right\|_{\psi_1(\pi^*(\cdot \mid x_0))} \lesssim \left( 1 \vee \frac{L}{b} \right) \left( d + M + e^{-2bT} \left\| \Sigma^{-1/2}(x_0 - m) \right\|^2 \right),$$

*where $\lesssim$ stands for inequality up to an absolute multiplicative constant.*

The proof of Lemma D.3 is postponed to Appendix D.3.

### D.1 PROOF OF LEMMA D.1

$$\|\varphi^*(X_T^*) - \varphi(X_T^*)\|_{\psi_1} \lesssim \left( 1 \vee \frac{L}{b} \right) \left( M + d + e^{-bT} \left( \frac{R^2}{\sqrt{d}} + \sqrt{d} + bR^2 \right) \right)$$

Let us note that Assumption 4 ensures that

$$|\varphi(X_T^*) - \varphi^*(X_T^*)| \leqslant L \left\| \Sigma^{-1/2}(X_T^* - m) \right\|^2 + M.$$

For this reason, the Orlicz norm of $\varphi(X_T^*) - \varphi^*(X_T^*)$ does not exceed

$$\|\varphi(X_T^*) - \varphi^*(X_T^*)\|_{\psi_1} \leqslant \left\| L \left\| \Sigma^{-1/2}(X_T^* - m) \right\|^2 + M \right\|_{\psi_1}$$

$$\leqslant L \left\| \left\| \Sigma^{-1/2}(X_T^* - m) \right\|^2 \right\|_{\psi_1} + \frac{M}{\log 2}.$$

In the last inequality, we used the fact that $\|c\|_{\psi_1} \leqslant |c|/\log 2$ for any constant $c \in \mathbb{R}$. The rest of the proof is devoted to the study of the Orlicz norm of $\|\Sigma^{-1/2}(X_T^* - m)\|^2$. According to Lemma E.3, for any $x_0 \in \mathrm{supp}(\rho_0)$, it holds that

$$\mathbb{E}\left[ e^{b\|\Sigma^{-1/2}(X_T^* - m)\|^2} \,\middle|\, X_0 = x_0 \right] \leqslant 2^{d/2} \exp \left\{ M\mathcal{K}(T) + \mathcal{A}(x_0, T)\mathcal{K}(T) \right\}$$

$$\cdot \exp \left\{ 2be^{-2bT} \left\| \Sigma^{-1/2}(x_0 - m) \right\|^2 \right\}.$$

In view of Assumption 2 and the definition of $\mathcal{A}(x, t)$ (see (38)), we have

$$\mathcal{A}(x_0, T) \leqslant \left( \frac{be^2 R^2}{\sqrt{d}} + 4e^2 \sqrt{d} \right) \arcsin(e^{-bT}) - 10e^2 \sqrt{d} \log \left( 1 - e^{-2bT} \right)$$

and

$$\left\| \Sigma^{-1/2}(x_0 - m) \right\|^2 \leqslant R^2$$

for any $x_0 \in \text{supp}(\rho_0)$. Thus, due to the tower rule, it holds that

$$
\mathbb{E}e^{b\|\Sigma^{-1/2}(X_T^*-m)\|^2}
$$
$$
= \mathbb{E}\mathbb{E}\left[ e^{b\|\Sigma^{-1/2}(X_T^*-m)\|^2} \,\Big|\, X_0 \right]
$$
$$
\leqslant \frac{2^{d/2}}{\left(1 - e^{-2bT}\right)^{10e^2\sqrt{d}}}
$$
$$
\cdot \exp\left\{ M\mathcal{K}(T) + \mathcal{K}(T)\left( \frac{be^2R^2}{\sqrt{d}} + 4e^2\sqrt{d} \right)\arcsin(e^{-bT}) + 2be^{-2bT}R^2 \right\}.
$$

Let us denote the expression in the right-hand side by $B$ and take

$$
v = \frac{1 \vee \log_2 B}{b} \lesssim \frac{1}{b} \vee \frac{1}{b}\left( M + d + e^{-bT}\left( \frac{R^2}{\sqrt{d}} + \sqrt{d} + bR^2 \right) \right).
$$

Then, applying the Lyapunov inequality, we obtain that

$$
\mathbb{E}e^{\|\Sigma^{-1/2}(X_T^*-m)\|^2/v} \leqslant \left( \mathbb{E}e^{b\|\Sigma^{-1/2}(X_T^*-m)\|^2} \right)^{1/(bv)} \leqslant B^{1/(1\vee\log_2 B)} \leqslant 2.
$$

Hence, the $\psi_1$-norm of $\|\Sigma^{-1/2}(X_T^*-m)\|^2$ does not exceed $v$, and we conclude that

$$
\|\varphi(X_T^*) - \varphi^*(X_T^*)\|_{\psi_1} \leqslant L\left\| \left\|\Sigma^{-1/2}(X_T^*-m)\right\|^2 \right\|_{\psi_1} + \frac{M}{\log 2}
$$
$$
\leqslant Lv + \frac{M}{\log 2}
$$
$$
\lesssim \left(1 \vee \frac{L}{b}\right)\left( M + d + e^{-bT}\left( \frac{R^2}{\sqrt{d}} + \sqrt{d} + bR^2 \right) \right).
$$

$\qquad\square$

## D.2 Proof of Lemma D.2

According to Lemma E.10, for any $x \in \mathbb{R}^d$, it holds that

$$
-\mathcal{A}(x,T)\mathcal{K}(T) + \mathcal{K}(T)\log\frac{\mathcal{T}_\infty[e^\varphi]}{e^M} \leqslant \log\frac{\mathcal{T}_T[e^\varphi](x)}{e^M} \leqslant \frac{\mathcal{A}(x,T)}{\mathcal{K}(T)} + \frac{1}{\mathcal{K}(T)}\log\frac{\mathcal{T}_\infty[e^\varphi]}{e^M}
$$

and, similarly,

$$
-\mathcal{A}(x,T)\mathcal{K}(T) + \mathcal{K}(T)\log\frac{\mathcal{T}_\infty[e^{\varphi^*}]}{e^M} \leqslant \log\frac{\mathcal{T}_T[e^{\varphi^*}](x)}{e^M} \leqslant \frac{\mathcal{A}(x,T)}{\mathcal{K}(T)} + \frac{1}{\mathcal{K}(T)}\log\frac{\mathcal{T}_\infty[e^{\varphi^*}]}{e^M},
$$

where the functions $\mathcal{A}(x,t)$ and $\mathcal{K}(t)$ are defined in (38) and (39), respectively. This, together with the inequalities

$$
e^M \geqslant \mathcal{T}_\infty e^\varphi \geqslant e^{\mathcal{T}_\infty\varphi} = 1 \quad\text{and}\quad e^M \geqslant \mathcal{T}_\infty e^{\varphi^*} \geqslant e^{\mathcal{T}_\infty\varphi^*} = 1,
$$

implies that

$$
\left| \log\frac{\mathcal{T}_T[e^\varphi](X_0)}{\mathcal{T}_T[e^{\varphi^*}](X_0)} \right| \leqslant \mathcal{A}(X_0,T)\left( \frac{1}{\mathcal{K}(T)} + \mathcal{K}(T) \right) + M\mathcal{K}(T).
$$

Thus, the Orlicz norm of $\mathcal{T}_T[e^\varphi](X_0)/\mathcal{T}_T[e^{\varphi^*}](X_0)$ satisfies the inequality

$$
\left\| \log\frac{\mathcal{T}_T[e^\varphi](X_0)}{\mathcal{T}_T[e^{\varphi^*}](X_0)} \right\|_{\psi_1} \leqslant \left\| \mathcal{A}(X_0,T)\left( \frac{1}{\mathcal{K}(T)} + \mathcal{K}(T) \right) + M\mathcal{K}(T) \right\|_{\psi_1}
$$
$$
\leqslant \left( \frac{1}{\mathcal{K}(T)} + \mathcal{K}(T) \right)\|\mathcal{A}(X_0,T)\|_{\psi_1} + \mathcal{K}(T)\|M\|_{\psi_1}.
$$

Let us note that any constant $c \in \mathbb{R}$ has a finite $\psi_1$-norm $|c|/\log 2$. This helps us to deduce that

$$\left\| \log \frac{\mathcal{T}_T[e^\varphi](X_0)}{\mathcal{T}_T[e^{\varphi^*}](X_0)} \right\|_{\psi_1} \leqslant \left( \frac{1}{\mathcal{K}(T)} + \mathcal{K}(T) \right) \|\mathcal{A}(X_0, T)\|_{\psi_1} + \frac{M\mathcal{K}(T)}{\log 2}.$$

Let us elaborate on the first term in the right-hand side. Taking the definitions of $\mathcal{A}(x, t)$ and $\mathcal{K}(t)$ into account (see (38) and (39)), one can observe that

$$\mathcal{A}(X_0, T) = \frac{be^2}{\sqrt{d}} \left\| \Sigma^{-1/2}(X_0 - m) \right\|^2 \arcsin(e^{-bT}) + 2\log \mathcal{K}(T).$$

In view of Assumption 2, the first term in the right-hand side does not exceed

$$\frac{be^2}{\sqrt{d}} \left\| \Sigma^{-1/2}(X_0 - m) \right\|^2 \arcsin(e^{-bT}) \leqslant \frac{be^2 R^2 \arcsin(e^{-bT})}{\sqrt{d}} \quad \text{almost surely.}$$

This yields that

$$\|\mathcal{A}(X_0, T)\|_{\psi_1} \leqslant \left\| \frac{be^2 R^2 \arcsin(e^{-bT})}{\sqrt{d}} + 2\log \mathcal{K}(T) \right\|_{\psi_1} \leqslant \frac{be^2 R^2 \arcsin(e^{-bT})}{\sqrt{d}\log 2} + \frac{2\log \mathcal{K}(T)}{\log 2},$$

and, finally,

$$\left\| \log \frac{\mathcal{T}_T[e^\varphi](X_0)}{\mathcal{T}_T[e^{\varphi^*}](X_0)} \right\|_{\psi_1} \leqslant \left( \frac{be^2 R^2 \arcsin(e^{-bT})}{\sqrt{d}\log 2} + \frac{2\log \mathcal{K}(T)}{\log 2} \right) \left( \frac{1}{\mathcal{K}(T)} + \mathcal{K}(T) \right) + \frac{M\mathcal{K}(T)}{\log 2}.$$

$\square$

## D.3 PROOF OF LEMMA D.3

Let us take $\lambda = \min\{1/2, b/(2L)\}$ and consider the conditional exponential moment

$$\mathbb{E}\left[ \exp\left\{ \lambda \left| \log \frac{\pi^*(X_T^* \,|\, x_0)}{\pi^\varphi(X_T^* \,|\, x_0)} \right| \right\} \,\Big|\, X_0 = x_0 \right].$$

According the Cauchy-Schwarz inequality, it holds that

$$\mathbb{E}\left[ \exp\left\{ \lambda \left| \log \frac{\pi^*(X_T^* \,|\, x_0)}{\pi^\varphi(X_T^* \,|\, x_0)} \right| \right\} \,\Big|\, X_0 = x_0 \right] \leqslant \sqrt{\mathbb{E}\left[ \exp\left\{ 2\lambda \left| \log \frac{\pi^*(X_T^* \,|\, x_0)}{\pi^\varphi(X_T^* \,|\, x_0)} \right| \right\} \,\Big|\, X_0 = x_0 \right]}.$$

Let us note that the expression in the right-hand side does not exceed

$$\sqrt{\mathbb{E}\left[ \exp\left\{ 2\lambda \log \frac{\pi^*(X_T^* \,|\, x_0)}{\pi^\varphi(X_T^* \,|\, x_0)} \right\} \,\Big|\, X_0 = x_0 \right] + \mathbb{E}\left[ \exp\left\{ 2\lambda \log \frac{\pi^\varphi(X_T^* \,|\, x_0)}{\pi^*(X_T^* \,|\, x_0)} \right\} \,\Big|\, X_0 = x_0 \right]}.$$

Using Hölder's inequality and taking into account that $\lambda \leqslant 1/2$, we obtain that the latter term satisfies the bound

$$\mathbb{E}\left[ \exp\left\{ 2\lambda \log \frac{\pi^\varphi(X_T^* \,|\, x_0)}{\pi^*(X_T^* \,|\, x_0)} \right\} \,\Big|\, X_0 = x_0 \right] \leqslant \left( \mathbb{E}\left[ \frac{\pi^\varphi(X_T^* \,|\, x_0)}{\pi^*(X_T^* \,|\, x_0)} \,\Big|\, X_0 = x_0 \right] \right)^{2\lambda}$$

$$= \left( \int_{\mathbb{R}^d} \frac{\pi^\varphi(x_T \,|\, x_0)}{\pi^*(x_T \,|\, x_0)} \pi^*(x_T \,|\, x_0) \,\mathrm{d}x_T \right)^{2\lambda}$$

$$= \left( \int_{\mathbb{R}^d} \pi^\varphi(x_T \,|\, x_0) \,\mathrm{d}x_T \right)^{2\lambda} = 1.$$

Thus, we proved that

$$\mathbb{E}\left[ \exp\left\{ \lambda \left| \log \frac{\pi^*(X_T^* \,|\, x_0)}{\pi^\varphi(X_T^* \,|\, x_0)} \right| \right\} \,\Big|\, X_0 = x_0 \right]$$

$$\leqslant \sqrt{1 + \mathbb{E}\left[ \exp\left\{ 2\lambda \log \frac{\pi^*(X_T^* \,|\, x_0)}{\pi^\varphi(X_T^* \,|\, x_0)} \right\} \,\Big|\, X_0 = x_0 \right]}.$$

In the rest of the proof, we are going to show that

$$\mathbb{E}\left[\exp\left\{2\lambda\log\frac{\pi^*(X_T^*\,|\,x_0)}{\pi^\varphi(X_T^*\,|\,x_0)}\right\}\,\Big|\,X_0 = x_0\right]$$
$$\leqslant \exp\left\{\frac{d\log 2}{2} + M\big(\mathcal{K}(T) + 3\big) + \mathcal{A}(x_0, T)\big(2\mathcal{K}(T) + 1\big)\right\}$$
$$\cdot \exp\left\{2be^{-2bT}\left\|\Sigma^{-1/2}(x_0 - m)\right\|^2\right\}.$$

For this purpose, let us note that

$$\log\frac{\pi^*(X_T^*\,|\,x_0)}{\pi^\varphi(X_T^*\,|\,x_0)} = \varphi^*(X_T^*) - \varphi(X_T^*) + \log\frac{\mathcal{T}_T[e^\varphi](x_0)}{\mathcal{T}_T[e^{\varphi^*}](x_0)}.$$

Since, due to the conditions of the lemma, $\varphi^*(x) \leqslant M$ and $\varphi(x) \geqslant -L\|\Sigma^{-1/2}(x-m)\|^2 - M$ for all $x \in \mathbb{R}^d$, it holds that

$$\varphi^*(X_T^*) - \varphi(X_T^*) \leqslant L\|\Sigma^{-1/2}(X_T^* - m)\|^2 + 2M. \tag{34}$$

Moreover, according to Lemma E.10, we have

$$\log\frac{\mathcal{T}_T[e^\varphi](x_0)}{\mathcal{T}_T[e^{\varphi^*}](x_0)} \leqslant \frac{\mathcal{A}(x_0, T)}{\mathcal{K}(T)} + \frac{1}{\mathcal{K}(T)}\log\mathcal{T}_\infty[e^\varphi] + \mathcal{A}(x_0, T)\mathcal{K}(T) - \mathcal{K}(T)\log\mathcal{T}_\infty[e^{\varphi^*}]$$
$$\leqslant \mathcal{A}(x_0, T)\left(\mathcal{K}(T) + \frac{1}{\mathcal{K}(T)}\right) + \frac{M}{\mathcal{K}(T)}. \tag{35}$$

In the last line, we used the fact that $\mathcal{T}_\infty[e^\varphi] \leqslant e^M$ and

$$\mathcal{T}_\infty[e^{\varphi^*}] \geqslant \exp\{\mathcal{T}_\infty[\varphi^*]\} = 1.$$

Summing up (34) and (35), we obtain that

$$\log\frac{\pi^*(X_T^*\,|\,x_0)}{\pi^\varphi(X_T^*\,|\,x_0)} \leqslant L\|\Sigma^{-1/2}(X_T - m)\|^2 + \left(2 + \frac{1}{\mathcal{K}(T)}\right)M + \mathcal{A}(x_0, T)\left(\mathcal{K}(T) + \frac{1}{\mathcal{K}(T)}\right)$$
$$\leqslant L\|\Sigma^{-1/2}(X_T^* - m)\|^2 + 3M + \mathcal{A}(x_0, T)\left(\mathcal{K}(T) + 1\right).$$

This immediately implies that

$$\mathbb{E}\left[\exp\left\{2\lambda\log\frac{\pi^*(X_T^*\,|\,x_0)}{\pi^\varphi(X_T^*\,|\,x_0)}\right\}\,\Big|\,X_0 = x_0\right]$$
$$\leqslant e^{6\lambda M + 2\lambda\mathcal{A}(x_0, T)(\mathcal{K}(T) + 1)}\,\mathbb{E}\left[e^{2\lambda L\|\Sigma^{-1/2}(X_T^* - m)\|^2}\,\Big|\,X_0 = x_0\right].$$

Applying Lemma E.3 and taking into account that $\lambda = \min\{1/2, b/(2L)\} \leqslant b/(2L)$, we conclude that

$$\mathbb{E}\left[\exp\left\{2\lambda\log\frac{\pi^*(X_T^*\,|\,x_0)}{\pi^\varphi(X_T^*\,|\,x_0)}\right\}\,\Big|\,X_0 = x_0\right]$$
$$\leqslant \exp\left\{\frac{d\log 2}{2} + M\big(\mathcal{K}(T) + 6\lambda\big) + \mathcal{A}(x_0, T)\big(\mathcal{K}(T) + 2\lambda\mathcal{K}(T) + 2\lambda\big)\right\}$$
$$\cdot \exp\left\{4e^{-2bT}\lambda L\left\|\Sigma^{-1/2}(x_0 - m)\right\|^2\right\}$$
$$\leqslant \exp\left\{\frac{d\log 2}{2} + M\big(\mathcal{K}(T) + 3\big) + \mathcal{A}(x_0, T)\big(2\mathcal{K}(T) + 1\big)\right\}$$
$$\cdot \exp\left\{2be^{-2bT}\left\|\Sigma^{-1/2}(x_0 - m)\right\|^2\right\}.$$

Hence, it holds that

$$\mathbb{E}\left[\exp\left\{\lambda\left|\log\frac{\pi^*(X_T^*\,|\,x_0)}{\pi^\varphi(X_T^*\,|\,x_0)}\right|\right\}\,\Big|\,X_0=x_0\right]$$

$$\leqslant\sqrt{1+\mathbb{E}\left[\exp\left\{2\lambda\log\frac{\pi^*(X_T^*\,|\,x_0)}{\pi^\varphi(X_T^*\,|\,x_0)}\right\}\,\Big|\,X_0=x_0\right]}$$

$$\leqslant 2^{(d+1)/2}\exp\left\{\frac{M}{2}\big(\mathcal{K}(T)+3\big)+\mathcal{A}(x_0,T)\big(\mathcal{K}(T)+1/2\big)\right\}$$

$$\cdot\exp\left\{be^{-2bT}\left\|\Sigma^{-1/2}(x_0-m)\right\|^2\right\}.$$

This inequality yields the desired upper bound on the Orlicz norm of $\log\big(\pi^*(X_T^*\,|\,x_0)/\pi^\varphi(X_T^*\,|\,x_0)\big)$. Indeed, let $v$ be the maximum of $\log(2)/\lambda$ and

$$\frac{\log 2}{\lambda}\left(\frac{d+1}{2}+\frac{M}{2}\big(\mathcal{K}(T)+3\big)+\mathcal{A}(x_0,T)\big(\mathcal{K}(T)+1/2\big)+be^{-2bT}\left\|\Sigma^{-1/2}(x_0-m)\right\|^2\right).$$

Then it holds that

$$\mathbb{E}\left[\exp\left\{\frac{1}{v}\left|\log\frac{\pi^*(X_T^*\,|\,x_0)}{\pi^\varphi(X_T^*\,|\,x_0)}\right|\right\}\,\Big|\,X_0=x_0\right]$$

$$\leqslant\left(\mathbb{E}\left[\exp\left\{\lambda\left|\log\frac{\pi^*(X_T^*\,|\,x_0)}{\pi^\varphi(X_T^*\,|\,x_0)}\right|\right\}\,\Big|\,X_0=x_0\right]\right)^{1/(\lambda v)}$$

$$\leqslant\left(2^{(d+1)/2}\exp\left\{\frac{M}{2}\big(\mathcal{K}(T)+3\big)+\mathcal{A}(x_0,T)\big(\mathcal{K}(T)+1/2\big)\right\}\right)^{1/(\lambda v)}$$

$$\cdot\exp\left\{\frac{be^{-2bT}}{\lambda v}\left\|\Sigma^{-1/2}(x_0-m)\right\|^2\right\}$$

$$\leqslant 2.$$

Thus, we obtain that

$$\left\|\log\frac{\pi^*(X_T^*\,|\,x_0)}{\pi^\varphi(X_T^*\,|\,x_0)}\right\|_{\psi_1(\pi^*(\cdot\,|\,x_0))}\leqslant v\lesssim\left(1\vee\frac{L}{b}\right)\left(d+M+e^{-2bT}\left\|\Sigma^{-1/2}(x_0-m)\right\|^2\right),$$

where $\lesssim$ stands for inequality up to an absolute multiplicative constant.

$\square$

## E  AUXILIARY RESULTS

This section collects auxiliary results. We provide their proofs (except for Lemma E.10) in Appendices E.1 – E.9. For the proof of Lemma E.10, a reader is referred to (Puchkin et al., 2025, Lemma B.3).

**Lemma E.1.** *Grant Assumptions 1, 2, 3, 4, and 5. Then there exists*

$$T_0\lesssim\frac{1}{b}\log\left(d+LR^2+\frac{Ld^2}{b}+M\right)$$

*such that for any $\delta\in(0,1)$ and any $T\geqslant T_0$ we have*

$$\left|\frac{1}{n}\sum_{i=1}^n\big(\varphi(Y_i)-\varphi^*(Y_i)\big)-\mathbb{E}\big(\varphi^*(Y_1)-\varphi(Y_1)\big)\right|$$

$$\lesssim\sqrt{\frac{\mathrm{Var}\big(\varphi^*(Y_1)-\varphi(Y_1)\big)\big(D\log(\Lambda nd)+\log(1/\delta)\big)}{n}}$$

$$+\left(1\vee\frac{L}{b}\right)\frac{(M+d)\big(D\log(\Lambda nd)+\log(1/\delta)\big)\log n}{n}$$

*and*

$$\left| \frac{1}{n} \sum_{j=1}^{n} \left( \log \frac{\mathcal{T}_T[e^\varphi](Z_j)}{\mathcal{T}_T[e^{\varphi^*}](Z_j)} - \mathbb{E} \log \frac{\mathcal{T}_T[e^\varphi](Z_1)}{\mathcal{T}_T[e^{\varphi^*}](Z_1)} \right) \right|$$

$$\lesssim \sqrt{\text{Var} \left( \log \frac{\mathcal{T}_T[e^\varphi](Z_1)}{\mathcal{T}_T[e^{\varphi^*}](Z_1)} \right) \frac{\big(D \log(\Lambda n d) + \log(1/\delta)\big)}{n}}$$

$$+ \frac{(M+1)\big(D \log(\Lambda n d) + \log(1/\delta)\big) \log n}{n}$$

*with probability at least $(1 - \delta)$ simultaneously for all $\varphi \in \Phi$.*

**Lemma E.2.** *Grant Assumption 2. Let $f_0 : \mathbb{R}^d \to \mathbb{R}$ and $f_1 : \mathbb{R}^d \to \mathbb{R}$ be arbitrary functions satisfying Assumption 4. Then, for any $x \in \text{supp}(\rho_0)$, it holds that*

$$\left| \log \mathcal{T}_T\big[e^{f_1}\big](x) - \log \mathcal{T}_T\big[e^{f_0}\big](x) \right|$$
$$\leqslant e^{\mathcal{A}(x,T)(\mathcal{K}(T)+1/\mathcal{K}(T))+M\mathcal{K}(T)}$$

$$\cdot \left( M + \frac{LR^2}{2} + \frac{L(50d + d^2/8)}{b} \right)^{1-1/\mathcal{K}(T)} \left( \mathcal{T}_\infty |f_1 - f_0| \right)^{1/\mathcal{K}(T)}.$$

**Lemma E.3.** *Grant Assumption 1. Then, for any $\lambda \leqslant b/(2L)$, it holds that*

$$\mathbb{E}\left[ e^{2\lambda L \|\Sigma^{-1/2}(X_T^* - m)\|^2} \,\Big|\, X_0 = x_0 \right] \leqslant 2^{d/2} \exp\{ M\mathcal{K}(T) + \mathcal{A}(x_0, T)\mathcal{K}(T) \}$$

$$\exp\left\{ 4e^{-2bT} \lambda L \left\| \Sigma^{-1/2}(x_0 - m) \right\|^2 \right\}.$$

**Lemma E.4.** *Under Assumption 1, it holds that*

$$\log \mathbb{E}\left[ e^{u^\top \Sigma^{-1/2}(X_T^* - m)} \,\big|\, X_0 = x_0 \right]$$

$$\leqslant M\mathcal{K}(T) + \mathcal{A}(x_0, T)\mathcal{K}(T) + e^{-bT} u^\top \Sigma^{-1/2}(x_0 - m) + \frac{(1 - e^{-2bT})\|u\|^2}{4b},$$

*where $\mathcal{A}(x, t)$ and $\mathcal{K}(t)$, are defined in (38) and (39), respectively.*

**Lemma E.5.** *Let $\xi$ be a random variable with a finite Orlicz norm $\|\xi\|_{\psi_1} < +\infty$. For any $\lambda$ such that $0 \leqslant \lambda \leqslant \|\xi - \mathbb{E}\xi\|_{\psi_1}$ let us define*

$$g(\lambda) = \log \mathbb{E} e^{\lambda(\xi - \mathbb{E}\xi)}.$$

*Then for all $\lambda \in (0, 1/\|\xi - \mathbb{E}\xi\|_{\psi_1})$ it holds that*

$$g'(\lambda) = \mathsf{P}_\lambda(\xi - \mathbb{E}\xi), \quad g''(\lambda) = \mathsf{P}_\lambda(\xi - \mathbb{E}\xi), \quad \text{and} \quad g'''(\lambda) = \mathsf{P}_\lambda(\xi - \mathsf{P}_\lambda\xi)^3,$$

*where, for any Borel function $f : \mathbb{R} \to \mathbb{R}$,*

$$\mathsf{P}_\lambda f(\xi) = \frac{\mathbb{E}\left[ f(\xi) e^{\lambda(\xi - \mathbb{E}\xi)} \right]}{\mathbb{E} e^{\lambda(\xi - \mathbb{E}\xi)}}.$$

**Lemma E.6.** *Let $\xi$ be a random variable with a finite Orlicz norm $\|\xi\|_{\psi_1} < +\infty$. For any Borel function $f : \mathbb{R} \to \mathbb{R}$ and any $\lambda \in (0, 1/\|\xi - \mathbb{E}\xi\|_{\psi_1})$, let us define*

$$\mathsf{P}_\lambda f(\xi) = \frac{\mathbb{E}\left[ f(\xi) e^{\lambda(\xi - \mathbb{E}\xi)} \right]}{\mathbb{E} e^{\lambda(\xi - \mathbb{E}\xi)}},$$

*and let*

$$\|\xi - \mathbb{E}\xi\|_{\psi_1(\mathsf{P}_\lambda)} = \inf \left\{ t > 0 : \mathsf{P}_\lambda e^{|\xi - \mathbb{E}\xi|/t} \leqslant 2 \right\}$$

*be the weighted Orlicz norm corresponding to the measure $\mathsf{P}_\lambda$. Then it holds that*

$$\|\xi - \mathbb{E}\xi\|_{\psi_1(\mathsf{P}_\lambda)} \leqslant \frac{\|\xi - \mathbb{E}\xi\|_{\psi_1}}{1 - \lambda \|\xi - \mathbb{E}\xi\|_{\psi_1}} \quad \text{for all } 0 \leqslant \lambda < \frac{1}{\|\xi - \mathbb{E}\xi\|_{\psi_1}}.$$

**Lemma E.7.** *For any $p > 0$ it holds that $\Gamma(p+1) \leqslant (p+1)^p$, where*

$$\Gamma(p+1) = \int\limits_0^{+\infty} u^p e^{-u}\,\mathrm{d}u$$

*stands for the gamma function.*

**Lemma E.8.** *Let $\xi$ be an arbitrary random variable with a finite Orlicz norm $\|\xi\|_{\psi_1} < +\infty$. Then, for any $p > 0$, it holds that*

$$\mathbb{E}|\xi|^p \leqslant 2\Gamma(p+1)\|\xi\|_{\psi_1}^p \leqslant 2(p+1)^p\|\xi\|_{\psi_1}^p.$$

**Lemma E.9.** *Let positive numbers $a$, $u$, and $\varkappa$ be such that*

$$u\big(1 \vee (\varkappa \log u)\big) \geqslant a.$$

*Then it holds that*

$$u \geqslant \frac{a}{1 \vee (\varkappa \log a)}.$$

**Lemma E.10** (Puchkin et al. (2025))**.** *Let $f : \mathbb{R}^d \to \mathbb{R}$ and $g : \mathbb{R}^d \to \mathbb{R}$. Assume that there exists $M \in \mathbb{R}$ and some non-negative constants $A$, $B$, and $\alpha$ such that*

$$f(x) \leqslant M, \quad 0 \leqslant g(x) \leqslant A\left\|\Sigma^{-1/2}(x-m)\right\|^\alpha + B \quad \text{for all } x \in \mathbb{R}^d. \tag{36}$$

*Let us fix arbitrary $x \in \mathbb{R}^d$ and $t > 0$ and introduce*

$$G(x) = Be^M + 2^{\alpha-1}Ae^M\left\|\Sigma^{-1/2}(x-m)\right\|^\alpha + 4^{\alpha-1}Ae^M(2b)^{-\alpha/2}\left((10\alpha\sqrt{d})^\alpha + d^\alpha\right), \tag{37}$$

$$\mathcal{A}(x,t) = \left(\frac{be^2}{\sqrt{d}}\left\|\Sigma^{-1/2}(x-m)\right\|^2 + 4e^2\sqrt{d}\right)\arcsin(e^{-bt}) - 10e^2\sqrt{d}\log\left(1 - e^{-2bt}\right), \tag{38}$$

*and*

$$\mathcal{K}(t) = \left(1 - e^{-2bt}\right)^{-5e^2\sqrt{d}} \cdot \exp\left\{2e^2\sqrt{d}\arcsin(e^{-bt})\right\} \tag{39}$$

*Then $\mathcal{T}_t[ge^f](x) \leqslant G(x)$ for all $x \in \mathbb{R}^d$ for any $t \in [0, +\infty]$, and, moreover, it holds that*

$$e^{-\mathcal{A}(x,t)\mathcal{K}(t)}\left(\frac{\mathcal{T}_\infty[ge^f]}{G(x)}\right)^{\mathcal{K}(t)} \leqslant \frac{\mathcal{T}_t[ge^f](x)}{G(x)} \leqslant e^{\mathcal{A}(x,t)/\mathcal{K}(t)}\left(\frac{\mathcal{T}_\infty[ge^f]}{G(x)}\right)^{1/\mathcal{K}(t)}.$$

## E.1 PROOF OF LEMMA E.1

The proof relies on concentration inequalities for sub-exponential random variables, smooth parametrization of $\varphi_\theta$ guaranteed by Assumption 5, and the standard $\varepsilon$-net argument. Let $\varepsilon \in (0, 1)$ be a parameter to be specified a bit later. Let $\Theta_\varepsilon$ stand for the minimal $\varepsilon$-net of $\Theta$ with respect to the $\ell_\infty$-norm and introduce

$$\Phi_\varepsilon = \{\varphi_\theta : \theta \in \Theta_\varepsilon\}.$$

Since $\Theta \subseteq [-1, 1]^D$, it is known that

$$|\Phi_\varepsilon| \leqslant |\Theta_\varepsilon| \leqslant \left(\frac{2}{\varepsilon}\right)^D.$$

The rest of the proof proceeds in three steps. First, we ensure that for any $\varphi \in \Phi$ the random variables

$$\varphi^*(Y_1) - \varphi(Y_1), \dots, \varphi^*(Y_n) - \varphi(Y_n) \quad \text{and} \quad \log\frac{\mathcal{T}_T[e^\varphi](Z_1)}{\mathcal{T}_T[e^{\varphi^*}](Z_1)}, \dots, \log\frac{\mathcal{T}_T[e^\varphi](Z_n)}{\mathcal{T}_T[e^{\varphi^*}](Z_n)}$$

are sub-exponential. Second, we apply (Lecué & Mitchell, 2012, Proposition 5.2) to quantify large deviation bounds simultaneously for all $\varphi \in \Phi_\varepsilon$. Finally, we extend these bounds to concentration inequalities holding uniformly for all $\varphi \in \Phi$.

**Step 1: Orlicz norm bounds.** The goal of this step is to establish upper bounds on $\psi_1$-norms of

$$\varphi^*(Y_1) - \varphi(Y_1), \ldots, \varphi^*(Y_n) - \varphi(Y_n) \quad \text{and} \quad \log \frac{\mathcal{T}_T[e^\varphi](Z_1)}{\mathcal{T}_T[e^{\varphi^*}](Z_1)}, \ldots, \log \frac{\mathcal{T}_T[e^\varphi](Z_n)}{\mathcal{T}_T[e^{\varphi^*}](Z_n)}.$$

They follow from Assumption 2, 3, and 4. Indeed, according to Lemma D.1, under Assumptions 3 and 4, it holds that

$$\|\varphi^*(Y_i) - \varphi(Y_i)\|_{\psi_1} \lesssim \left(1 \vee \frac{L}{b}\right)\left(M + d + e^{-bT}\left(\frac{R^2}{\sqrt{d}} + \sqrt{d} + bR^2\right)\right)$$

$$\lesssim \left(1 \vee \frac{L}{b}\right)(M + d) \quad \text{for all } 1 \leqslant i \leqslant n. \tag{40}$$

On the other hand, Lemma D.2 implies that

$$\left\|\log \frac{\mathcal{T}_T[e^\varphi](Z_j)}{\mathcal{T}_T[e^{\varphi^*}](Z_j)}\right\|_{\psi_1} \leqslant \left(\frac{be^2 R^2 \arcsin(e^{-bT})}{\sqrt{d}\log 2} + \frac{2\log \mathcal{K}(T)}{\log 2}\right)\left(\frac{1}{\mathcal{K}(T)} + \mathcal{K}(T)\right) + \frac{M\mathcal{K}(T)}{\log 2}$$

for all $1 \leqslant j \leqslant n$, where $\mathcal{K}(T)$ is defined in (39). Let us note that there exists $T_0 \lesssim (\log d \vee \log R)/b$ such that for all $T \geqslant T_0$

$$\left(\frac{R^2}{\sqrt{d}} + \sqrt{d}\right)e^{-bT} \lesssim 1 \quad \text{and} \quad \log \mathcal{K}(T) \lesssim \sqrt{d}e^{-bT} \lesssim 1.$$

Hence, if $T$ is large enough, then

$$\left\|\log \frac{\mathcal{T}_T[e^\varphi](Z_j)}{\mathcal{T}_T[e^{\varphi^*}](Z_j)}\right\|_{\psi_1} \lesssim \left(\frac{R^2}{\sqrt{d}} + \sqrt{d}\right)e^{-bT} + M \lesssim M + 1 \quad \text{for any } 1 \leqslant j \leqslant n. \tag{41}$$

**Step 2: concentration of sub-exponential random variables.** The $\psi_1$-norm bounds obtained on the previous step allow us to use Bernstein's inequality for unbounded random variables. Let us fix an arbitrary $\varphi \in \Phi_\varepsilon$ and apply (Lecué & Mitchell, 2012, Proposition 5.2)). Then, for any $\delta \in (0, 1)$, with probability at least $1 - \delta/(4|\Phi_\varepsilon|)$, it holds that

$$\left|\frac{1}{n}\sum_{i=1}^n \left(\varphi^*(Y_i) - \varphi(Y_i)\right) - \mathbb{E}\left(\varphi^*(Y_1) - \varphi(Y_1)\right)\right|$$

$$\lesssim \sqrt{\frac{\mathrm{Var}\left(\varphi^*(Y_1) - \varphi(Y_1)\right)\log(4|\Phi_\varepsilon|/\delta)}{n}}$$

$$+ \left\|\max_{1 \leqslant i \leqslant n}\left(\varphi^*(Y_i) - \varphi(Y_i)\right)\right\|_{\psi_1} \cdot \frac{\log(4|\Phi_\varepsilon|/\delta)}{n}. \tag{42}$$

Similarly, with probability at least $1 - \delta/(4|\Phi_\varepsilon|)$, we have

$$\left|\frac{1}{n}\sum_{j=1}^n \left(\log \frac{\mathcal{T}_T[e^\varphi](Z_j)}{\mathcal{T}_T[e^{\varphi^*}](Z_j)} - \mathbb{E}\log \frac{\mathcal{T}_T[e^\varphi](Z_1)}{\mathcal{T}_T[e^{\varphi^*}](Z_1)}\right)\right|$$

$$\lesssim \sqrt{\mathrm{Var}\left(\log \frac{\mathcal{T}_T[e^\varphi](Z_1)}{\mathcal{T}_T[e^{\varphi^*}](Z_1)}\right)\frac{\log(4|\Phi_\varepsilon|/\delta)}{n}}$$

$$+ \left\|\max_{1 \leqslant j \leqslant n}\log \frac{\mathcal{T}_T[e^\varphi](Z_j)}{\mathcal{T}_T[e^{\varphi^*}](Z_j)}\right\|_{\psi_1} \cdot \frac{\log(4|\Phi_\varepsilon|/\delta)}{n}. \tag{43}$$

Due to the the union bound, there exists an event $\mathcal{E}_0$ of probability at least $1 - \delta/2$ such that the inequalities (42) and (43) hold simultaneously for all $\varphi \in \Phi_\varepsilon$ on this event. Moreover, using Pisier's inequality (see, for example, (Lecué & Mitchell, 2012, p. 1827)), we conclude that

$$\left|\frac{1}{n}\sum_{i=1}^n \left(\varphi^*(Y_i) - \varphi(Y_i)\right) - \mathbb{E}\left(\varphi^*(Y_1) - \varphi(Y_1)\right)\right| \lesssim \sqrt{\frac{\mathrm{Var}\left(\varphi^*(Y_1) - \varphi(Y_1)\right)\log(4|\Phi_\varepsilon|/\delta)}{n}}$$

$$+ \|\varphi^*(Y_1) - \varphi(Y_1)\|_{\psi_1} \cdot \frac{\log(4|\Phi_\varepsilon|/\delta)\log n}{n}.$$

and

$$\left| \frac{1}{n} \sum_{j=1}^{n} \left( \log \frac{\mathcal{T}_T[e^\varphi](Z_j)}{\mathcal{T}_T[e^{\varphi^*}](Z_j)} - \mathbb{E} \log \frac{\mathcal{T}_T[e^\varphi](Z_1)}{\mathcal{T}_T[e^{\varphi^*}](Z_1)} \right) \right|$$

$$\lesssim \sqrt{\operatorname{Var}\left( \log \frac{\mathcal{T}_T[e^\varphi](Z_1)}{\mathcal{T}_T[e^{\varphi^*}](Z_1)} \right) \frac{\log(4|\Phi_\varepsilon|/\delta)}{n}}$$

$$+ \left\| \log \frac{\mathcal{T}_T[e^\varphi](Z_1)}{\mathcal{T}_T[e^{\varphi^*}](Z_1)} \right\|_{\psi_1} \cdot \frac{\log(4|\Phi_\varepsilon|/\delta) \log n}{n}$$

on the event $\mathcal{E}_0$ simultaneously for all $\varphi \in \Phi_\varepsilon$. In view of (40) and (41), we have

$$\left| \frac{1}{n} \sum_{i=1}^{n} \left( \varphi^*(Y_i) - \varphi(Y_i) \right) - \mathbb{E}\left( \varphi^*(Y_1) - \varphi(Y_1) \right) \right|$$

$$\lesssim \sqrt{\frac{\operatorname{Var}\left( \varphi^*(Y_1) - \varphi(Y_1) \right) \log(4|\Phi_\varepsilon|/\delta)}{n}} \tag{44}$$

$$+ \left( 1 \vee \frac{L}{b} \right) \frac{(M+d)\log(4|\Phi_\varepsilon|/\delta)\log n}{n}.$$

and

$$\left| \frac{1}{n} \sum_{j=1}^{n} \left( \log \frac{\mathcal{T}_T[e^\varphi](Z_j)}{\mathcal{T}_T[e^{\varphi^*}](Z_j)} - \mathbb{E} \log \frac{\mathcal{T}_T[e^\varphi](Z_1)}{\mathcal{T}_T[e^{\varphi^*}](Z_1)} \right) \right|$$

$$\lesssim \sqrt{\operatorname{Var}\left( \log \frac{\mathcal{T}_T[e^\varphi](Z_1)}{\mathcal{T}_T[e^{\varphi^*}](Z_1)} \right) \frac{\log(4|\Phi_\varepsilon|/\delta)}{n}} \tag{45}$$

$$+ \frac{(M+1)\log(4|\Phi_\varepsilon|/\delta)\log n}{n}$$

on the same event simultaneously for all $\varphi \in \Phi_\varepsilon$.

**Step 3: from $\varepsilon$-nets to uniform bounds.** The goal of this step is to transform this upper bound to a one holding uniformly for all $\varphi \in \Phi$. According to (Rigollet & Hütter, 2023, Theorem 1.19), the norm of a sub-Gaussian random vector satisfies the inequality

$$\mathbb{P}\left( \|Y_1\| \geqslant u \right) \leqslant 6^d \exp\left\{ -\frac{u^2}{8\mathrm{v}^2} \right\} \quad \text{for all } u > 0.$$

Then, due to the union bound there is an event $\mathcal{E}_1$ of probability at least $(1 - \delta/2)$ such that

$$\max_{1 \leqslant i \leqslant n} \|Y_i\|^2 \leqslant d\log 6 + 8\mathrm{v}^2 \log(2n/\delta) \quad \text{on } \mathcal{E}_1. \tag{46}$$

We are going to show that the desired uniform bounds hold on an event $\mathcal{E} = \mathcal{E}_0 \cup \mathcal{E}_1$ of probability at least $(1 - \delta)$.

Let us fix an arbitrary $\theta \in \Theta$ and let $\theta_\varepsilon$ be an element of $\Theta_\varepsilon$ such that

$$\|\theta - \theta_\varepsilon\|_\infty \leqslant \varepsilon.$$

The existence of such $\theta_\varepsilon$ follows from the definition of the $\varepsilon$-net. Let us denote the corresponding to $\theta$ and $\theta_\varepsilon$ functions by $\varphi \in \Phi$ and $\varphi_\varepsilon \in \Phi$, respectively. Due to Assumption 5, it holds that

$$|\varphi(Y_i) - \varphi_\varepsilon(Y_i)| \leqslant \Lambda\varepsilon\left( 1 + \|Y_i\|^2 \right) \quad \text{almost surely for all } 1 \leqslant i \leqslant n.$$

This implies that

$$\left| \mathbb{E}\left( \varphi^*(Y_1) - \varphi(Y_1) \right) \right| \leqslant \left| \mathbb{E}\left( \varphi^*(Y_1) - \varphi_\varepsilon(Y_1) \right) \right| + \left| \mathbb{E}\left( \varphi(Y_1) - \varphi_\varepsilon(Y_1) \right) \right|$$

$$\leqslant \left| \mathbb{E}\left( \varphi^*(Y_1) - \varphi_\varepsilon(Y_1) \right) \right| + \Lambda\varepsilon\,\mathbb{E}\left( 1 + \|Y_1\|^2 \right) \tag{47}$$

$$\lesssim \left| \mathbb{E}\left( \varphi^*(Y_1) - \varphi_\varepsilon(Y_1) \right) \right| + \Lambda\varepsilon\left( 1 + \mathrm{v}^2 d \right)$$

and

$$
\begin{aligned}
\mathrm{Var}\big(\varphi_\varepsilon(Y_1) - \varphi^*(Y_1)\big) &\leqslant 2\mathrm{Var}\big(\varphi(Y_1) - \varphi^*(Y_1)\big) + 2\mathrm{Var}\big(\varphi(Y_1) - \varphi_\varepsilon(Y_1)\big) \\
&\leqslant 2\mathrm{Var}\big(\varphi(Y_1) - \varphi^*(Y_1)\big) + 2\Lambda\varepsilon\,\mathbb{E}\big(1 + \|Y_1\|^2\big)^2 \\
&\lesssim \mathrm{Var}\big(\varphi(Y_1) - \varphi^*(Y_1)\big) + \Lambda\varepsilon(1 + \mathrm{v}^4 d^2).
\end{aligned} \tag{48}
$$

Moreover, in view of (46), we have

$$
\begin{aligned}
\left|\frac{1}{n}\sum_{i=1}^{n}\big(\varphi(Y_i) - \varphi^*(Y_i)\big)\right| &\leqslant \left|\frac{1}{n}\sum_{i=1}^{n}\big(\varphi_\varepsilon(Y_i) - \varphi^*(Y_i)\big)\right| + \left|\frac{1}{n}\sum_{i=1}^{n}\big(\varphi(Y_i) - \varphi_\varepsilon(Y_i)\big)\right| \\
&\leqslant \left|\frac{1}{n}\sum_{i=1}^{n}\big(\varphi_\varepsilon(Y_i) - \varphi^*(Y_i)\big)\right| + \frac{\Lambda\varepsilon}{n}\sum_{i=1}^{n}\big(1 + \|Y_i\|^2\big) \\
&\leqslant \left|\frac{1}{n}\sum_{i=1}^{n}\big(\varphi_\varepsilon(Y_i) - \varphi^*(Y_i)\big)\right| + \Lambda\varepsilon\left(1 + d\log 6 + 8\mathrm{v}^2\log\frac{2n}{\delta}\right).
\end{aligned} \tag{49}
$$

Combining the bounds (47), (48), and (49) with the inequality (44), we obtain that

$$
\begin{aligned}
&\left|\frac{1}{n}\sum_{i=1}^{n}\big(\varphi(Y_i) - \varphi^*(Y_i)\big) - \mathbb{E}\big(\varphi^*(Y_1) - \varphi(Y_1)\big)\right| \\
&\lesssim \Lambda\varepsilon d + \sqrt{\frac{\mathrm{Var}\big(\varphi^*(Y_1) - \varphi(Y_1)\big)\log(4|\Phi_\varepsilon|/\delta)}{n}} \\
&\quad + \left(1 \vee \frac{L}{b}\right)\frac{(M + d)\log(4|\Phi_\varepsilon|/\delta)\log n}{n}
\end{aligned} \tag{50}
$$

on the event $\mathcal{E}$ simultaneously for all $\varphi \in \Phi$.

The analysis of

$$
\left|\frac{1}{n}\sum_{j=1}^{n}\left(\log\frac{\mathcal{T}_T[e^\varphi](Z_j)}{\mathcal{T}_T[e^{\varphi^*}](Z_j)} - \mathbb{E}\log\frac{\mathcal{T}_T[e^\varphi](Z_1)}{\mathcal{T}_T[e^{\varphi^*}](Z_1)}\right)\right|
$$

is carried out in a similar way. As before, let us fix an arbitrary $\theta \in \Theta$, let $\theta_\varepsilon$ be the closest to $\theta$ element of $\Theta_\varepsilon$ and denote the log-potentials corresponding to $\theta$ and $\theta_\varepsilon$ by $\varphi$ and $\varphi_\varepsilon$, respectively. According to Lemma E.2, it holds that

$$
\begin{aligned}
&\big|\log\mathcal{T}_T\big[e^\varphi\big](x) - \log\mathcal{T}_T\big[e^{\varphi_\varepsilon}\big](x)\big| \\
&\leqslant e^{\mathcal{A}(x,T)(\mathcal{K}(T)+1/\mathcal{K}(T))+M\mathcal{K}(T)}\left(M + \frac{LR^2}{2} + \frac{L(50d + d^2/8)}{b}\right)^{1-1/\mathcal{K}(T)}\big(\mathcal{T}_\infty|\varphi - \varphi_\varepsilon|\big)^{1/\mathcal{K}(T)}
\end{aligned}
$$

for all $x \in \mathrm{supp}(\rho_0)$, where the functions $\mathcal{A}(x,t)$ and $\mathcal{K}(t)$ are defined in (38) and (39), respectively. Assumption 5 ensures that

$$
\begin{aligned}
\mathcal{T}_\infty|\varphi - \varphi_\varepsilon| &= \mathbb{E}_{Y\sim\mathcal{N}(m,\Sigma)}|\varphi(Y) - \varphi_\varepsilon(Y)| \\
&\leqslant \mathbb{E}_{Y\sim\mathcal{N}(m,\Sigma)}\big[\Lambda\varepsilon\big(1 + \|Y\|^2\big)\big] \\
&= \Lambda\varepsilon(1 + \|m\|^2 + \mathrm{Tr}(\Sigma)) \lesssim \Lambda\varepsilon.
\end{aligned}
$$

Moreover, due to (38), (33), and Assumption 2, we have

$$
\mathcal{A}(x,T) \leqslant \overline{\mathcal{A}}(T) = \left(\frac{be^2R^2}{\sqrt{d}} + 4e^2\sqrt{d}\right)\arcsin(e^{-bT}) - 10e^2\sqrt{d}\log\big(1 - e^{-2bT}\big)
$$

for all $x \in \mathrm{supp}(\rho_0)$. Note that the expression in the right-hand side is of order

$$
\mathcal{O}\left(\left(\frac{R^2}{\sqrt{d}} + \sqrt{d}\right)e^{-bT}\right) \quad \text{whenever} \quad bT \gtrsim \log R \vee \log d.
$$

Introducing

$$\mathcal{H}(T) = e^{\overline{\mathcal{A}}(T)(\mathcal{K}(T)+1/\mathcal{K}(T))+M\mathcal{K}(T)} \left( M + \frac{LR^2}{2} + \frac{L(50d + d^2/8)}{b} \right)^{1-1/\mathcal{K}(T)}$$

for brevity, we obtain that

$$\left| \log \mathcal{T}_T[e^{\varphi}](Z_j) - \log \mathcal{T}_T[e^{\varphi_\varepsilon}](Z_j) \right| \lesssim \mathcal{H}(T) (\Lambda\varepsilon)^{1/\mathcal{K}(T)} \tag{51}$$

almost surely for all $j \in \{1, \ldots, n\}$. This yields that

$$\left| \mathbb{E} \log \mathcal{T}_T[e^{\varphi}](Z_1) - \log \mathcal{T}_T[e^{\varphi^*}](Z_1) \right|$$

$$\leqslant \left| \mathbb{E} \log \mathcal{T}_T[e^{\varphi}](Z_1) - \log \mathcal{T}_T[e^{\varphi_\varepsilon}](Z_1) \right| + \left| \mathbb{E} \log \mathcal{T}_T[e^{\varphi_\varepsilon}](Z_1) - \log \mathcal{T}_T[e^{\varphi^*}](Z_1) \right| \tag{52}$$

$$\lesssim \left| \mathbb{E} \log \mathcal{T}_T[e^{\varphi_\varepsilon}](Z_1) - \log \mathcal{T}_T[e^{\varphi^*}](Z_1) \right| + \mathcal{H}(T) (\Lambda\varepsilon)^{1/\mathcal{K}(T)}$$

and

$$\mathrm{Var}\left( \log \mathcal{T}_T[e^{\varphi_\varepsilon}](Z_1) - \log \mathcal{T}_T[e^{\varphi^*}](Z_1) \right)$$

$$\leqslant 2\mathrm{Var}\left( \log \mathcal{T}_T[e^{\varphi}](Z_1) - \log \mathcal{T}_T[e^{\varphi^*}](Z_1) \right)$$

$$+ 2\mathrm{Var}\left( \log \mathcal{T}_T[e^{\varphi_\varepsilon}](Z_1) - \log \mathcal{T}_T[e^{\varphi}](Z_1) \right) \tag{53}$$

$$\lesssim \mathrm{Var}\left( \log \mathcal{T}_T[e^{\varphi}](Z_1) - \log \mathcal{T}_T[e^{\varphi^*}](Z_1) \right) + \mathcal{H}^2(T) (\Lambda\varepsilon)^{2/\mathcal{K}(T)}.$$

The inequalities (51), (52), and (53) combined with (45) yield that

$$\left| \frac{1}{n} \sum_{j=1}^{n} \left( \log \frac{\mathcal{T}_T[e^{\varphi}](Z_j)}{\mathcal{T}_T[e^{\varphi^*}](Z_j)} - \mathbb{E} \log \frac{\mathcal{T}_T[e^{\varphi}](Z_1)}{\mathcal{T}_T[e^{\varphi^*}](Z_1)} \right) \right|$$

$$\lesssim \sqrt{\mathrm{Var}\left( \log \frac{\mathcal{T}_T[e^{\varphi}](Z_1)}{\mathcal{T}_T[e^{\varphi^*}](Z_1)} \right) \frac{\log(4|\Phi_\varepsilon|/\delta)}{n}} \tag{54}$$

$$+ \mathcal{H}(T) (\Lambda\varepsilon)^{1/\mathcal{K}(T)} + \frac{(M + 1)\log(4|\Phi_\varepsilon|/\delta) \log n}{n}$$

simultaneously for all $\varphi \in \Phi$ on the event $\mathcal{E}$.

**Step 4: choice of $\varepsilon$ and final bounds.** It remains to choose a proper $\varepsilon > 0$ to finish the proof. The inequalities (50), (54), and $|\Phi_\varepsilon| \leqslant (2/\varepsilon)^D$ imply that

$$\left| \frac{1}{n} \sum_{i=1}^{n} (\varphi(Y_i) - \varphi^*(Y_i)) - \mathbb{E}(\varphi^*(Y_1) - \varphi(Y_1)) \right|$$

$$\lesssim \Lambda\varepsilon d + \sqrt{\frac{\mathrm{Var}(\varphi^*(Y_1) - \varphi(Y_1))(D\log(1/\varepsilon) + \log(1/\delta))}{n}} \tag{55}$$

$$+ \left( 1 \vee \frac{L}{b} \right) \frac{(M + d)(D\log(1/\varepsilon) + \log(1/\delta)) \log n}{n}$$

and

$$\left| \frac{1}{n} \sum_{j=1}^{n} \left( \log \frac{\mathcal{T}_T[e^{\varphi}](Z_j)}{\mathcal{T}_T[e^{\varphi^*}](Z_j)} - \mathbb{E} \log \frac{\mathcal{T}_T[e^{\varphi}](Z_1)}{\mathcal{T}_T[e^{\varphi^*}](Z_1)} \right) \right|$$

$$\lesssim \sqrt{\mathrm{Var}\left( \log \frac{\mathcal{T}_T[e^{\varphi}](Z_1)}{\mathcal{T}_T[e^{\varphi^*}](Z_1)} \right) \frac{(D\log(1/\varepsilon) + \log(1/\delta))}{n}} \tag{56}$$

$$+ \mathcal{H}(T) (\Lambda\varepsilon)^{1/\mathcal{K}(T)} + \frac{(M + 1)(D\log(1/\varepsilon) + \log(1/\delta)) \log n}{n}$$

simultaneously for all $\varphi \in \Phi$ on the event $\mathcal{E}$ of probability at least $1 - \delta$. Let us take

$$\varepsilon = \frac{1}{\Lambda} \min \left\{ \left( \frac{1}{n\mathcal{H}(T)} \right)^{\mathcal{K}(T)}, \frac{1}{nd} \right\}.$$

Such a choice ensures that

$$\max \left\{ \Lambda \varepsilon d, \mathcal{H}(T) (\Lambda \varepsilon)^{1/\mathcal{K}(T)} \right\} = \frac{1}{n}$$

and that

$$\log \frac{1}{\varepsilon} \leqslant \log \Lambda + (\log(nd) \vee \mathcal{K}(T) \log(n\mathcal{H}(T)))$$

$$\lesssim \log \Lambda + \log(nd) + \overline{\mathcal{A}}(T) \left( \mathcal{K}(T) + \frac{1}{\mathcal{K}(T)} \right)$$

$$+ M\mathcal{K}(T) + \left( 1 - \frac{1}{\mathcal{K}(T)} \right) \log(M + LR^2 + Ld^2)$$

Then, due to (33) and (39), there exists

$$T_0 \lesssim \frac{1}{b} \log \left( d + LR^2 + \frac{Ld^2}{b} + M \right)$$

such that for any $T \geqslant T_0$ we have

$$\log \frac{1}{\varepsilon} \lesssim \log(\Lambda nd) + \left( \frac{R^2}{\sqrt{d}} + \sqrt{d} \right) e^{-bT} + \sqrt{d} e^{-bT} \left( M + \log(M + LR^2 + Ld^2) \right) \lesssim \log(\Lambda nd).$$

Substituting this bound into (55) and (56), we obtain that

$$\left| \frac{1}{n} \sum_{i=1}^{n} (\varphi(Y_i) - \varphi^*(Y_i)) - \mathbb{E}(\varphi^*(Y_1) - \varphi(Y_1)) \right|$$

$$\lesssim \sqrt{\frac{\operatorname{Var}(\varphi^*(Y_1) - \varphi(Y_1))(D\log(\Lambda nd) + \log(1/\delta))}{n}}$$

$$+ \left( 1 \vee \frac{L}{b} \right) \frac{(M + d)(D\log(\Lambda nd) + \log(1/\delta)) \log n}{n}$$

and

$$\left| \frac{1}{n} \sum_{j=1}^{n} \left( \log \frac{\mathcal{T}_T[e^\varphi](Z_j)}{\mathcal{T}_T[e^{\varphi^*}](Z_j)} - \mathbb{E} \log \frac{\mathcal{T}_T[e^\varphi](Z_1)}{\mathcal{T}_T[e^{\varphi^*}](Z_1)} \right) \right|$$

$$\lesssim \sqrt{\operatorname{Var}\left( \log \frac{\mathcal{T}_T[e^\varphi](Z_1)}{\mathcal{T}_T[e^{\varphi^*}](Z_1)} \right) \frac{(D\log(\Lambda nd) + \log(1/\delta))}{n}}$$

$$+ \frac{(M + 1)(D\log(\Lambda nd) + \log(1/\delta)) \log n}{n}$$

simultaneously for all $\varphi \in \Phi$ on $\mathcal{E}$. The proof is finished.

$\square$

### E.2 PROOF OF LEMMA E.2

Let us fix an arbitrary $x \in \operatorname{supp}(\rho_0)$. For any $s \in [0, 1]$, we introduce $f_s(y) = s f_1(y) + (1 - s) f_0(y)$ and

$$F(s) = \log \mathcal{T}_T [e^{f_s}](x).$$

By the mean value theorem, we have

$$\left| \log \mathcal{T}_T [e^{f_1}](x) - \log \mathcal{T}_T [e^{f_0}](x) \right| = |F(1) - F(0)| \leqslant \sup_{s \in [0,1]} \left| \frac{dF(s)}{ds} \right|.$$

For this reason, the rest of the proof is devoted to the study of $\mathrm{d}F(s)/\mathrm{d}s$. Let us fix an arbitrary $s \in [0, 1]$ and note that the absolute value of the derivative does not exceed

$$\left| \frac{\mathrm{d}F(s)}{\mathrm{d}s} \right| = \left| \frac{\mathcal{T}_T \left[ (f_1 - f_0) e^{f_s} \right](x)}{\mathcal{T}_T[e^{f_s}](x)} \right| \leqslant \frac{\mathcal{T}_T \left[ |f_1 - f_0| e^{f_s} \right](y)}{\mathcal{T}_T[e^{f_s}](x)}.$$

An upper bound on $|\mathrm{d}F(s)/\mathrm{d}s|$ will follow from Lemma E.10. Indeed, due to the conditions of the lemma, for any $y \in \mathbb{R}^d$, it holds that

$$f_s(y) = s f_1(y) + (1 - s) f_0(y) \leqslant sM + (1 - s)M = M$$

and

$$|f_s(y)| \leqslant s |f_1(y)| + (1 - s) |f_0(y)| \leqslant L \|\Sigma^{-1/2}(y - m)\|^2 + M.$$

Applying Lemma E.10, we obtain that

$$\frac{\mathcal{T}_T e^{f_s(x)}}{e^M} \geqslant e^{-\mathcal{A}(x,T)\mathcal{K}(T)} \left( \frac{\mathcal{T}_\infty e^{f_s}}{e^M} \right)^{\mathcal{K}(T)} \tag{57}$$

and

$$\frac{\mathcal{T}_T \left[ |f_1 - f_0| e^{f_s} \right](x)}{\mathcal{G}(x)} \leqslant e^{\mathcal{A}(x,T)/\mathcal{K}(T)} \left( \frac{\mathcal{T}_\infty \left[ |f_1 - f_0| e^{f_s} \right]}{\mathcal{G}(x)} \right)^{1/\mathcal{K}(T)}, \tag{58}$$

where the functions $\mathcal{A}(x, t)$ and $\mathcal{K}(t)$ are defined in (38) and (39), respectively, and

$$\mathcal{G}(x) = M e^M + \frac{L e^M}{2} \left\| \Sigma^{-1/2}(x - m) \right\|^2 + \frac{L e^M}{8b} \left( 400d + d^2 \right).$$

The inequalities (57) and (58) yield that

$$\left| \frac{\mathrm{d}F(s)}{\mathrm{d}s} \right| \leqslant e^{\mathcal{A}(x,T)(\mathcal{K}(T)+1/\mathcal{K}(T))} \left( \frac{\mathcal{G}(x)}{e^M} \right) \left( \frac{\mathcal{T}_\infty \left[ |f_1 - f_0| e^{f_s} \right]}{\mathcal{G}(x)} \right)^{1/\mathcal{K}(T)} \left( \frac{e^M}{\mathcal{T}_\infty e^{f_s}} \right)^{\mathcal{K}(T)}$$

$$\leqslant e^{\mathcal{A}(x,T)(\mathcal{K}(T)+1/\mathcal{K}(T))} \left( \frac{\mathcal{G}(x)}{e^M} \right) \left( \frac{e^M \mathcal{T}_\infty |f_1 - f_0|}{\mathcal{G}(x)} \right)^{1/\mathcal{K}(T)} \left( \frac{e^M}{\mathcal{T}_\infty e^{f_s}} \right)^{\mathcal{K}(T)}$$

$$= e^{\mathcal{A}(x,T)(\mathcal{K}(T)+1/\mathcal{K}(T))} \left( \frac{\mathcal{G}(x)}{e^M} \right)^{1-1/\mathcal{K}(T)} (\mathcal{T}_\infty |f_1 - f_0|)^{1/\mathcal{K}(T)} \left( \frac{e^M}{\mathcal{T}_\infty e^{f_s}} \right)^{\mathcal{K}(T)}.$$

The expression in the right-hand side can be simplified, if one takes into account the normalization conditions $\mathcal{T}_\infty f_0 = \mathcal{T}_\infty f_1 = 0$. According to Jensen's inequality, we have

$$\mathcal{T}_\infty e^{f_s} \geqslant e^{\mathcal{T}_\infty f_s} = e^{s \mathcal{T}_\infty f_1 + (1-s) \mathcal{T}_\infty f_0} = 1.$$

Moreover, in view of Assumption 2, it holds that

$$\frac{\mathcal{G}(x)}{e^M} = M + \frac{L}{2} \left\| \Sigma^{-1/2}(x - m) \right\|^2 + \frac{L}{8b} \left( 400d + d^2 \right) \leqslant M + \frac{LR^2}{2} + \frac{L(50d + d^2/8)}{b}.$$

Hence, we finally obtain that

$$\left| \frac{\mathrm{d}F(s)}{\mathrm{d}s} \right| \leqslant \exp \left\{ \mathcal{A}(x, T) \left( \mathcal{K}(T) + \frac{1}{\mathcal{K}(T)} \right) + M \mathcal{K}(T) \right\}$$

$$\cdot \left( M + \frac{LR^2}{2} + \frac{L(50d + d^2/8)}{b} \right)^{1-1/\mathcal{K}(T)} (\mathcal{T}_\infty |f_1 - f_0|)^{1/\mathcal{K}(T)}$$

for any $s \in [0, 1]$. This yields the desired bound.

$$\square$$

### E.3 PROOF OF LEMMA E.3

Let $\eta = (\eta_1, \ldots, \eta_d)^\top \sim \mathcal{N}(0, I_d)$ be a Gaussian random vector, which is independent of $X_T^*$ and $X_0$. Then it holds that

$$\mathbb{E}\left[e^{2\lambda L\|\Sigma^{-1/2}(X_T^*-m)\|^2} \,\Big|\, X_0 = x_0\right] = \mathbb{E}_\eta \,\mathbb{E}\left[e^{2\sqrt{\lambda L}\,\eta^\top \Sigma^{-1/2}(X_T^*-m)} \,\Big|\, \eta, X_0 = x_0\right].$$

Applying Lemma E.4, we obtain that

$$\mathbb{E}_\eta \,\mathbb{E}\left[e^{2\sqrt{\lambda L}\,\eta^\top \Sigma^{-1/2}(X_T^*-m)} \,\Big|\, \eta, X_0 = x_0\right]$$

$$\leqslant \exp\left\{M\mathcal{K}(T) + \mathcal{A}(x_0, T)\mathcal{K}(T)\right\}$$

$$\cdot \mathbb{E}_\eta \exp\left\{2e^{-bT}\sqrt{\lambda L}\,\eta^\top \Sigma^{-1/2}(x_0 - m) + \frac{\lambda L(1 - e^{-2bT})\|\eta\|^2}{b}\right\}$$

$$= \exp\left\{M\mathcal{K}(T) + \mathcal{A}(x_0, T)\mathcal{K}(T)\right\}$$

$$\cdot \prod_{j=1}^{d}\left(\mathbb{E}_{\eta_j} \exp\left\{2e^{-bT}\sqrt{\lambda L}\,\eta_j\left(\Sigma^{-1/2}(x_0 - m)\right)_j + \frac{\lambda L(1 - e^{-2bT})\eta_j^2}{b}\right\}\right).$$

The expectations with respect to $\eta_j$'s can be computed explicitly. Indeed, it is straightforward to check that

$$\mathbb{E}_{\eta_j} \exp\left\{2e^{-bT}\sqrt{\lambda L}\,\eta_j\left(\Sigma^{-1/2}(x_0 - m)\right)_j + \frac{\lambda L(1 - e^{-2bT})\eta_j^2}{b}\right\}$$

$$= \left(1 - \frac{\lambda L(1 - e^{-2bT})}{b}\right)^{-1/2} \exp\left\{\frac{2e^{-2bT}\lambda L\left(\Sigma^{-1/2}(x_0 - m)\right)_j^2}{1 - \lambda L(1 - e^{-2bT})/b}\right\}.$$

This yields that

$$\mathbb{E}\left[e^{2\lambda L\|\Sigma^{-1/2}(X_T^*-m)\|^2} \,\Big|\, X_0 = x_0\right]$$

$$= \mathbb{E}_\eta \,\mathbb{E}\left[e^{2\sqrt{\lambda L}\,\eta^\top \Sigma^{-1/2}(X_T^*-m)} \,\Big|\, \eta, X_0 = x_0\right]$$

$$\leqslant \left(1 - \frac{\lambda L(1 - e^{-2bT})}{b}\right)^{-d/2} \exp\left\{M\mathcal{K}(T) + \mathcal{A}(x_0, T)\mathcal{K}(T)\right\}$$

$$\cdot \exp\left\{\frac{2e^{-2bT}\lambda L\left\|\Sigma^{-1/2}(x_0 - m)\right\|^2}{1 - \lambda L(1 - e^{-2bT})/b}\right\}$$

$$\leqslant \left(1 - \frac{\lambda L}{b}\right)^{-d/2} \exp\left\{M\mathcal{K}(T) + \mathcal{A}(x_0, T)\mathcal{K}(T)\right\}$$

$$\cdot \exp\left\{\frac{2e^{-2bT}\lambda L\left\|\Sigma^{-1/2}(x_0 - m)\right\|^2}{1 - \lambda L(1 - e^{-2bT})/b}\right\}.$$

Since $\lambda \leqslant \min\{1/2, b/(2L)\} \leqslant b/(2L)$, we conclude that

$$\mathbb{E}\left[e^{2\lambda L\|\Sigma^{-1/2}(X_T^*-m)\|^2} \,\Big|\, X_0 = x_0\right] \leqslant 2^{d/2} \exp\left\{M\mathcal{K}(T) + \mathcal{A}(x_0, T)\mathcal{K}(T)\right\}$$

$$\cdot \exp\left\{4e^{-2bT}\lambda L\left\|\Sigma^{-1/2}(x_0 - m)\right\|^2\right\}.$$

$\square$

### E.4 PROOF OF LEMMA E.4

Let us recall that the conditional density of $X_T^*$ given $X_0 = x_0$ is equal to

$$\pi^*(x_T \,|\, x_0) = \frac{e^{\varphi^*(x_T)}\pi^0(x_T \,|\, x_0)}{\mathcal{T}_T[e^{\varphi^*}](x_0)}.$$

Applying the change of measure theorem, we observe that

$$\mathbb{E}\left[e^{u^\top \Sigma^{-1/2}(X_T^* - m)} \,\big|\, X_0 = x_0\right] = \frac{1}{\mathcal{T}_T[e^{\varphi^*}](x_0)} \, \mathbb{E}\left[e^{u^\top \Sigma^{-1/2}(X_T^0 - m) + \varphi^*(X_T^0)} \,\big|\, X_0 = x_0\right]$$

$$\leqslant \frac{e^M}{\mathcal{T}_T[e^{\varphi^*}](x_0)} \, \mathbb{E}\left[e^{u^\top \Sigma^{-1/2}(X_T^0 - m)} \,\big|\, X_0 = x_0\right].$$

According to Lemma E.10, we have

$$\frac{e^M}{\mathcal{T}_T[e^{\varphi^*}](x_0)} \leqslant \left(\frac{e^M}{\mathcal{T}_\infty[e^{\varphi^*}]}\right)^{\mathcal{K}(T)} \cdot e^{\mathcal{A}(x_0, T)\mathcal{K}(T)}.$$

The expression in the right-hand side can be simplified, if one takes into account that

$$\mathcal{T}_\infty[e^{\varphi^*}] \geqslant \exp\left\{\mathcal{T}_\infty \varphi^*\right\} = 1.$$

Then it holds that

$$\frac{e^M}{\mathcal{T}_T[e^{\varphi^*}](x_0)} \leqslant e^{M\mathcal{K}(T) + \mathcal{A}(x_0, T)\mathcal{K}(T)}.$$

Furthermore, since $\{X_t^0 : 0 \leqslant t \leqslant T\}$, is the Ornstein-Uhlenbeck process, the conditional distribution of $X_T^0$ given $X_0 = x_0$ is Gaussian $\mathcal{N}\big(m_T(x_0), \Sigma_T\big)$. This yields that

$$\mathbb{E}\left[e^{u^\top \Sigma^{-1/2}(X_T^0 - m)} \,\big|\, X_0 = x_0\right] = \exp\left\{u^\top \Sigma^{-1/2}(m_T(x_0) - m) + u^\top \Sigma^{-1/2}\Sigma_T\Sigma^{-1/2}u/2\right\}$$

$$= \exp\left\{e^{-bT}\, u^\top \Sigma^{-1/2}(x_0 - m) + \frac{(1 - e^{-2bT})\|u\|^2}{4b}\right\}.$$

Hence, we obtain that

$$\log \mathbb{E}\left[e^{u^\top \Sigma^{-1/2}(X_T^* - m)} \,\big|\, X_0 = x_0\right] \leqslant M\mathcal{K}(T) + \mathcal{A}(x_0, T)\mathcal{K}(T)$$

$$+ e^{-bT}\, u^\top \Sigma^{-1/2}(x_0 - m) + \frac{(1 - e^{-2bT})\|u\|^2}{4b}.$$

$\square$

### E.5 PROOF OF LEMMA E.5

First, it is straightforward to check that

$$g'(\lambda) = \frac{\mathbb{E}\left[(\xi - \mathbb{E}\xi)e^{\lambda(\xi - \mathbb{E}\xi)}\right]}{\mathbb{E}e^{\lambda(\xi - \mathbb{E}\xi)}} = \mathsf{P}_\lambda(\xi - \mathbb{E}\xi),$$

so let us focus on the second and the third derivatives of $g(\lambda)$. Let us note that, for any Borel function $f : \mathbb{R} \to \mathbb{R}$ such that $\mathsf{P}_\lambda f(\xi)$ and $\mathsf{P}_\lambda\big(\xi f(\xi)\big)$ are well-defined, it holds that

$$\frac{\mathrm{d}\mathsf{P}_\lambda f(\xi)}{\mathrm{d}\lambda} = \frac{\mathbb{E}\left[f(\xi)(\xi - \mathbb{E}\xi)e^{\lambda(\xi - \mathbb{E}\xi)}\right]}{\mathbb{E}e^{\lambda(\xi - \mathbb{E}\xi)}} - \frac{\mathbb{E}\left[f(\xi)e^{\lambda(\xi - \mathbb{E}\xi)}\right]}{\mathbb{E}e^{\lambda(\xi - \mathbb{E}\xi)}} \cdot \frac{\mathbb{E}\left[(\xi - \mathbb{E}\xi)e^{\lambda(\xi - \mathbb{E}\xi)}\right]}{\mathbb{E}e^{\lambda(\xi - \mathbb{E}\xi)}}$$

$$= \mathsf{P}_\lambda\big[f(\xi)(\xi - \mathbb{E}\xi)\big] - \mathsf{P}_\lambda f(\xi)\mathsf{P}_\lambda(\xi - \mathbb{E}\xi). \tag{59}$$

Applying this formula to $f(\xi) = \xi - \mathbb{E}\xi$, we obtain that

$$g''(\lambda) = \mathsf{P}_\lambda(\xi - \mathbb{E}\xi)^2 - \big(\mathsf{P}_\lambda(\xi - \mathbb{E}\xi)\big)^2. \tag{60}$$

On the other hand, it holds that

$$\mathsf{P}_\lambda(\xi - \mathsf{P}_\lambda\xi)^2 = \mathsf{P}_\lambda\big(\xi - \mathbb{E}\xi - \mathsf{P}_\lambda(\xi - \mathbb{E}\xi)\big)^2$$

$$= \mathsf{P}_\lambda(\xi - \mathbb{E}\xi)^2 - 2\mathsf{P}_\lambda\big[(\xi - \mathbb{E}\xi)\big(\mathsf{P}_\lambda(\xi - \mathbb{E}\xi)\big)\big] + \big(\mathsf{P}_\lambda(\xi - \mathbb{E}\xi)\big)^2$$

$$= \mathsf{P}_\lambda(\xi - \mathbb{E}\xi)^2 - \big(\mathsf{P}_\lambda(\xi - \mathbb{E}\xi)\big)^2.$$

Hence, we showed that

$$g''(\lambda) = \mathsf{P}_\lambda(\xi - \mathsf{P}_\lambda\xi)^2.$$

It remains to consider $g'''(\lambda)$ to finish the proof. Taking into account (60) and applying (59), we obtain that

$$
\begin{aligned}
g'''(\lambda) &= \frac{\mathrm{d}}{\mathrm{d}\lambda}\left(\mathsf{P}_\lambda(\xi - \mathbb{E}\xi)^2 - \left(\mathsf{P}_\lambda(\xi - \mathbb{E}\xi)\right)^2\right) \\
&= \left(\mathsf{P}_\lambda(\xi - \mathbb{E}\xi)^3 - \mathsf{P}_\lambda(\xi - \mathbb{E}\xi)^2\mathsf{P}_\lambda(\xi - \mathbb{E}\xi)\right) \\
&\quad - 2\mathsf{P}_\lambda(\xi - \mathbb{E}\xi)\left(\mathsf{P}_\lambda(\xi - \mathbb{E}\xi)^2 - \left(\mathsf{P}_\lambda(\xi - \mathbb{E}\xi)\right)^2\right) \\
&= \mathsf{P}_\lambda(\xi - \mathbb{E}\xi)^3 - 3\mathsf{P}_\lambda(\xi - \mathbb{E}\xi)^2\mathsf{P}_\lambda(\xi - \mathbb{E}\xi) + 2\left(\mathsf{P}_\lambda(\xi - \mathbb{E}\xi)\right)^3.
\end{aligned}
\tag{61}
$$

Let us note that

$$
\begin{aligned}
\mathsf{P}_\lambda(\xi - \mathsf{P}_\lambda\xi)^3 &= \mathsf{P}_\lambda\left(\xi - \mathbb{E}\xi - \mathsf{P}_\lambda(\xi - \mathbb{E}\xi)\right)^3 \\
&= \mathsf{P}_\lambda\left(\xi - \mathbb{E}\xi\right)^3 - 3\mathsf{P}_\lambda\left(\xi - \mathbb{E}\xi\right)^2\mathsf{P}_\lambda(\xi - \mathbb{E}\xi) \\
&\quad + 3\mathsf{P}_\lambda(\xi - \mathbb{E}\xi)\left(\mathsf{P}_\lambda(\xi - \mathbb{E}\xi)\right)^2 - \left(\mathsf{P}_\lambda(\xi - \mathbb{E}\xi)\right)^3 \\
&= \mathsf{P}_\lambda(\xi - \mathbb{E}\xi)^3 - 3\mathsf{P}_\lambda(\xi - \mathbb{E}\xi)^2\mathsf{P}_\lambda(\xi - \mathbb{E}\xi) + 2\left(\mathsf{P}_\lambda(\xi - \mathbb{E}\xi)\right)^3.
\end{aligned}
$$

Thus, the right-hand side of (61) simplifies to

$$
g'''(\lambda) = \mathsf{P}_\lambda(\xi - \mathsf{P}_\lambda\xi)^3,
$$

as required. The proof is finished.

$\square$

### E.6 Proof of Lemma E.6

Due to Jensen's inequality, for any $\lambda \in \mathbb{R}$ it holds that

$$
\mathbb{E}e^{\lambda(\xi - \mathbb{E}\xi)} \geqslant \exp\left\{\lambda\mathbb{E}(\xi - \mathbb{E}\xi)\right\} = 1.
$$

This yields that

$$
\begin{aligned}
\mathsf{P}_\lambda e^{|\xi - \mathbb{E}\xi|/t} = \frac{\mathbb{E}e^{|\xi - \mathbb{E}\xi|/t + \lambda(\xi - \mathbb{E}\xi)}}{\mathbb{E}e^{\lambda(\xi - \mathbb{E}\xi)}} &\leqslant \mathbb{E}\exp\left\{\frac{|\xi - \mathbb{E}\xi|}{t} + \lambda(\xi - \mathbb{E}\xi)\right\} \\
&\leqslant \mathbb{E}\exp\left\{\left(\frac{1}{t} + \lambda\right)|\xi - \mathbb{E}\xi|\right\}.
\end{aligned}
$$

Taking

$$
t = \frac{\|\xi - \mathbb{E}\xi\|_{\psi_1}}{1 - \lambda\|\xi - \mathbb{E}\xi\|_{\psi_1}},
$$

we obtain that

$$
\mathsf{P}_\lambda e^{|\xi - \mathbb{E}\xi|/t} \leqslant \mathbb{E}\exp\left\{\left(\frac{1 - \lambda\|\xi - \mathbb{E}\xi\|_{\psi_1}}{\|\xi - \mathbb{E}\xi\|_{\psi_1}} + \lambda\right)|\xi - \mathbb{E}\xi|\right\} = \mathbb{E}\exp\left\{\frac{|\xi - \mathbb{E}\xi|}{\|\xi - \mathbb{E}\xi\|_{\psi_1}}\right\} \leqslant 2.
$$

The last inequality implies that

$$
\|\xi - \mathbb{E}\xi\|_{\psi_1(\mathsf{P}_\lambda)} \leqslant \frac{\|\xi - \mathbb{E}\xi\|_{\psi_1}}{1 - \lambda\|\xi - \mathbb{E}\xi\|_{\psi_1}} \quad \text{for all } 0 \leqslant \lambda < \frac{1}{\|\xi - \mathbb{E}\xi\|_{\psi_1}}.
$$

$\square$

### E.7 Proof of Lemma E.7

Let us fix an arbitrary $p > 0$ and introduce

$$
h(u) = u - p\log u.
$$

Since the function $h(u)$ is convex on $(0, +\infty)$, for any $u > 0$ we have

$$h(u) \geqslant h(p+1) + h'(p+1)(u - p - 1)$$
$$= p + 1 - p \log(p+1) + \frac{u - p - 1}{p + 1}$$
$$= p - p \log(p+1) + \frac{u}{p+1}.$$

This yields that

$$\Gamma(p+1) = \int\limits_0^{+\infty} e^{-h(u)}\, du \leqslant \left(\frac{p+1}{e}\right)^p \int\limits_0^{+\infty} e^{-u/(p+1)}\, du = (p+1)\left(\frac{p+1}{e}\right)^p.$$

Applying the inequality $1 + p \leqslant e^p$, which holds for all $p \in \mathbb{R}$, we obtain that

$$\Gamma(p+1) \leqslant (p+1)\left(\frac{p+1}{e}\right)^p \leqslant (p+1)^p.$$

$\square$

### E.8 PROOF OF LEMMA E.8

Since $|\xi|^p$ is a non-negative random variable, we can compute its expectation according to the formula

$$\mathbb{E}|\xi|^p = \int\limits_0^{+\infty} \mathbb{P}\left(|\xi|^p \geqslant t\right) dt = \int\limits_0^{+\infty} \mathbb{P}\left(|\xi| \geqslant t^{1/p}\right) dt.$$

Making a substitution $t^{1/p} = u$, $u \in (0, +\infty)$, we obtain that

$$\mathbb{E}|\xi|^p = p \int\limits_0^{+\infty} u^{p-1}\mathbb{P}\left(|\xi| \geqslant u\right) du.$$

Due to the Markov inequality and the definition of $\psi_1$-norm, the probability of the event $\{|\xi| \geqslant u\}$ does not exceed

$$\mathbb{P}\left(|\xi| \geqslant u\right) \leqslant e^{-u/\|\xi\|_{\psi_1}} \mathbb{E}e^{|\xi|/\|\xi\|_{\psi_1}} \leqslant 2e^{-u/\|\xi\|_{\psi_1}}.$$

This implies that

$$\mathbb{E}|\xi|^p \leqslant 2p \int\limits_0^{+\infty} u^{p-1}e^{-u/\|\xi\|_{\psi_1}} du = 2p\,\|\xi\|_{\psi_1}^p \int\limits_0^{+\infty} v^{p-1}e^{-v} dv = 2p\,\Gamma(p)\|\xi\|_{\psi_1}^p = 2\Gamma(p+1)\|\xi\|_{\psi_1}^p.$$

Applying Lemma E.7, we conclude that

$$\mathbb{E}|\xi|^p \leqslant 2(p+1)^p\|\xi\|_{\psi_1}^p.$$

### E.9 PROOF OF LEMMA E.9

Let us show that the inequality $u < a/\big(1 \vee (\varkappa \log a)\big)$ yields $u\big(1 \vee (\varkappa \log u)\big) < a$. Indeed, if

$$u < \frac{a}{1 \vee (\varkappa \log a)},$$

then it holds that

$$u\big(1 \vee (\varkappa \log u)\big) < \frac{a}{1 \vee (\varkappa \log a)} \vee \frac{a\varkappa \log a}{1 \vee (\varkappa \log a)} = \frac{a\big(1 \vee (\varkappa \log a)\big)}{1 \vee (\varkappa \log a)} = a.$$

Hence, the bound $u\big(1 \vee (\varkappa \log u)\big) \geqslant a$ implies that

$$u \geqslant \frac{a}{1 \vee (\varkappa \log a)}.$$

$\square$

# F    FURTHER DETAILS OF NUMERICAL EXPERIMENTS

In this section, we elaborate on details of numerical experiments presented in Section 4. The section is organized as follows. In Appendix F.1, we discuss general implementation details. Appendix F.2 presents information about the metrics used and describes the algorithm performance. Appendix F.3 provides additional details about the 25 Gaussian mixture experiment. Appendix F.5 provides additional information about the I2I translation experiment in the ALAE latent space. Finally, Appendix F.6 presents all final hyperparameter values used for the experiments.

## F.1    GENERAL IMPLEMENTATION DETAILS

In our experiments, we followed the official LightSB implementation from the repository of the authors. For comparison, the author's parameters were preserved where possible.

Our modification provides additional parameters of the Ornstein-Uhlenbeck process $b$ and $m$. For the experiments, these parameters were iterated over using grid search and Bayesian optimization in the form $C \cdot (1, 1, \ldots, 1)$, where $C$ is some constant. Such optimization is not perfectly comprehensive and does not cover all possible types of parameters. However, it is much simpler and cheaper computationally and allows for a statistically significant improvement in the performance of the basic LightSB.

Due to the strong restriction of the space in which the parameters are iterated to $[-1, 1]$ for both $b$ and $m$, it was possible to optimize them using Optuna (Akiba et al., 2019) and achieve good convergence of optimization. We used 50 trial optimization for optuna over the sliced $\mathbb{W}_1$ metric in the 25 Gaussian and I2I translation experiments and 100 trial optimization for optuna over the sliced $\mathbb{W}_1$ metric for Single Cell data.

The calculations were performed on the personal computer (AMD Ryzen 5 1600 Six-Core 3.20 GHz CPU, 24 GB RAM) and took a reasonable amount of time.

## F.2    METRICS USED IN THE EXPERIMENTS

Metrics below are industry standards because they:

(a)  provide quantitative, theoretically-grounded measures of distribution similarity;

(b)  are applicable to high-dimensional data;

(c)  can detect different failure modes of generative models (mode collapse, memorization, etc.);

(d)  have been widely adopted in research papers and practical applications.

Metrics used in the empirical analysis of LightSB-OU:

1. Sliced Wasserstein Distance:

$$\text{Sliced } \mathbb{W}_1(X, Y) = \frac{1}{N} \sum_{n=1}^{N} \mathbb{W}_1 \left( \{\langle x_i, \theta_n \rangle\}_{i=1}^{M}, \{\langle y_i, \theta_n \rangle\}_{i=1}^{M} \right),$$

where $\{\theta_n : 1 \leqslant n \leqslant N\}$ are i.i.d. random vectors drawn from the uniform distributed on the unit sphere $\mathbb{S}^{d-1}$.

2. Maximum Mean Discrepancy:

$$MMD^2(X, Y) = \mathbb{E}[k(X, X)] + \mathbb{E}[k(Y, Y)] - 2\mathbb{E}[k(X, Y)],$$

with Gaussian kernel $k(x, y) = \exp\{-\gamma \|x - y\|^2\}$.

3. Energy Distance:

$$\text{Energy Distance}(X, Y) = 2\mathbb{E}\|X - Y\| - \mathbb{E}\|X - X'\| - \mathbb{E}\|Y - Y'\|,$$

where $X'$ and $Y'$ are independent copies of $X$ and $Y$, respectively.

---

**Algorithm 1** LightSB-OU Training

---

1: **Input:** the number of mixture components $K$, the number of iterations $N_{\text{steps}}$, learning rate $\eta$, batch size $n_B$, time horizon $T$ and parameters $b > 0$, $m \in \mathbb{R}^d$, and $\varepsilon > 0$ of the reference Ornstein-Uhlenbeck process

$$\mathrm{d}X_t^0 = b(m - X_t^0)\,\mathrm{d}t + \sqrt{\varepsilon}\,\mathrm{d}W_t, \quad 0 \leqslant t \leqslant T.$$

2: Initialize parameters: $\theta = \{(r_k, S_k, \log \alpha_k) : 1 \leqslant k \leqslant K\}$ with vectors $r_1, \dots, r_K \in \mathbb{R}^d$, symmetric positive definite matrices $S_1, \dots, S_K \in \mathbb{R}^{d \times d}$, and with the positive weights $\alpha_1, \dots, \alpha_K$ such that $\alpha_1 + \dots + \alpha_K = 1$.

3: **for** $i$ from $1$ **to** $N_{\text{steps}}$ **do**

4:     Receive batches of fresh i.i.d. observations $Z_1^i, \dots, Z_{n_B}^i \sim \rho_0$ and $Y_1^i, \dots, Y_{n_B}^i \sim \rho_T$.

5:     Define the adjusted Schrödinger potential estimate

$$v_\theta(y) = \sum_{k=1}^{K} \alpha_k\, \mathsf{p}(y; r_k, \varepsilon \sigma_T^2 S_k), \quad y \in \mathbb{R}^d,$$

  where, for any $k \in \{1, \dots, K\}$, $\mathsf{p}(\,\cdot\,; r_k, \varepsilon \sigma_T^2 S_k)$ is the density of the Gaussian distribution $\mathcal{N}(r_k, \varepsilon \sigma_T^2 S_k)$.

6:     Define the normalizing constant

$$c_\theta(z) = \sum_{k=1}^{K} \alpha_k \cdot \exp\left\{ \frac{r_k^\top m_T(z)}{\varepsilon \sigma_T^2} + \frac{\|S_k^{1/2} m_T(z)\|^2}{2\varepsilon \sigma_T^2} \right\},$$

  where

$$m_T(z) = e^{-bT} z + (1 - e^{-bT})m \quad \text{and} \quad \sigma_T^2 = \frac{1 - e^{-2bT}}{2b}.$$

7:     Update the parameters via gradient descent:

$$\theta \leftarrow \theta - \eta \nabla_\theta \left[ -\frac{1}{n_B} \sum_{j=1}^{n_B} v_\theta(Y_j^i) + \frac{1}{n_B} \sum_{j=1}^{n_B} \log c_\theta(Z_j^i) \right]$$

8: **end for**

9: **Return** the adjusted potential $v_\theta(y) = \sum_{k=1}^{K} \alpha_k\, \mathsf{p}(y; r_k, \varepsilon \sigma_T^2 S_k)$.

---

4. Number of Covered Modes (for Gaussian Mixture Experiment):

$$\text{Covered Modes} = \sum_{k=1}^{K} \text{Covered}_k,$$

where

$$\text{Covered}_k = \mathbb{I}\left[ \min_{x \in X} (x - \mu_k)^\top \Sigma_k^{-1} (x - \mu_k) \leqslant \chi_{2,\alpha}^2 \right].$$

For the Sliced Wasserstein Distance, 100 random projections were used, and for the number of covered modes, a confidence interval of 90% was used.

We also provide a description of the training algorithm for LightSB-OU (Algorithm 1). The pseudocode provided allows a reader to visually evaluate how our approach differs from regular LightSB.

### F.3   DETAILS OF EVALUATION ON THE GAUSSIAN MIXTURE

To evaluate the algorithm's performance on Gaussian mixtures, the parameters for the basic LightSB and our modification were chosen based on reasonable considerations for such a problem and the parameters for the 2D Swiss Roll given in the original paper.

To find the metrics sliced $\mathbb{W}_1$, MMD, covered modes and energy distance, 5 iterations were used, including sampling and translation of 10000 random points from the standard normal distribution by trained models. They were used to find the mean values and standard deviations in the table and on the graphs. During the inference, we have used the fact that the conditional density of $X_T^{\varphi_\theta}$ given $X_0^{\varphi_\theta} = x$ has a form (see (20))

$$\pi^{\varphi_\theta}\left(y \mid x\right) = \frac{\pi^{\varphi_\theta}(x,y)}{\rho_0(x)} = e^{y^\top m_1(x)/(\varepsilon\sigma_1^2)}v_\theta(y)/c_\theta(x),$$

where $v_\theta(y)$ and $c_\theta(x)$ are given by (19) and (21), respectively. When $v_\theta(y)$ is a Gaussian mixture, it is straightforward to check that the conditional distribution is a Gaussian mixture as well. More specifically, if $v_\theta$ is defined by (19), then

$$\pi^{\varphi_\theta}\left(y \mid x\right) = \sum_{k=1}^{K}\widetilde{\alpha}_k(x)\mathsf{p}(y; r_k(x), \varepsilon\sigma_1^2 S_k),$$

where $\widetilde{\alpha}_k(x)$ is given by (22), $\mathsf{p}(y; r_k(x), \varepsilon\sigma_1^2 S_k)$ stands for the density of $\mathcal{N}\left(r_k(x), \varepsilon\sigma_1^2 S_k\right)$ and $r_k(x) = r_k + S_k m_1(x)$ for all $k \in \{1, \ldots, K\}$.

The Figure 3 shows the metrics for LightSB-OU for three experimental setups with a mixture of 25 Gaussians for different $K$ with the interval of one std shown in semi-transparent color. The Figure 4 shows joint graphs for LightSB and LightSB-OU, which allow us to evaluate the comparative behavior of the algorithms for different setups and different $K$, and finally the Figure 5 shows examples of the results of both algorithms for different setups with $K = 25$.

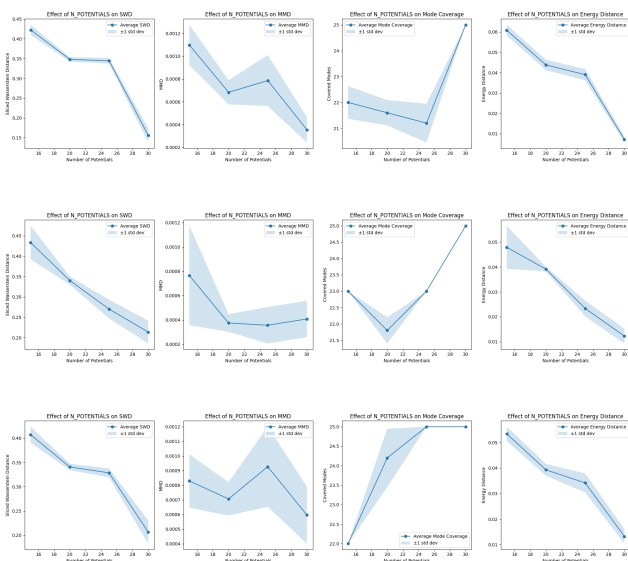

Figure 3: Translation results from standard normal distribution using LightSB. Top: uniform grid and standard covariance matrices. Middle: random location on the grid and standard covariance matrices. Bottom: uniform grid and anisotropic random covariance matrix.

## F.4 DETAILS OF EVALUATION ON THE SINGLE CELL DATA

We analyzed cell differentiation across five stages from day 0 to 27 (i.e. $t_0$ : day 0 to 3, $t_1$ : day 6 to 9, $t_2$ : day 12 to 15, $t_3$ : day 18 to 21, $t_4$ : day 24 to 27), processing the scRNA-seq data through quality filters and PCA to create distinct feature vectors. In this experiment the goal was to predict the cell distribution at an intermediate time point ($t_i$) by "transporting" data from the surrounding intervals ($t_{i-1}$ and $t_{i+1}$), where $i \in [1, 2, 3]$. To evaluate the model's accuracy, we calculated the Wasserstein-1 ($\mathbb{W}_1$) distance to see how closely our predictions matched the actual observed data.

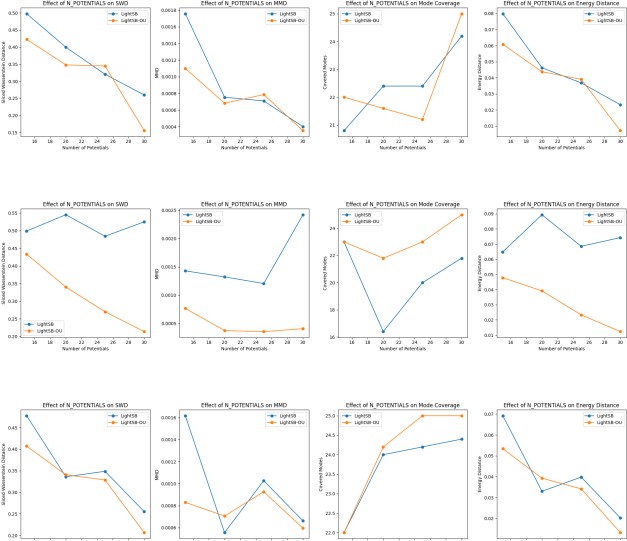

Figure 4: Translation results from standard normal distribution using LightSB and our modified approach. Top: uniform grid and standard covariance matrices. Middle: random location on the grid and standard covariance matrices. Bottom: uniform grid and anisotropic random covariance matrix.

## F.5 DETAILS OF EVALUATION ON THE UNPAIRED IMAGE-TO-IMAGE TRANSLATION

We used the official code and ALAE model from `https://github.com/podgorskiy/ALAE` and and neural network extracted attributes for the FFHQ dataset from `https://github.com/DCGM/ffhq-features-dataset`.

To evaluate the algorithm's performance in the I2I translation task, the parameters given for the LightSB model in the official implementation were used. From the trained model, 3 face translation results were sampled 100 times, and examples illustrating the best coverage of the latent space by our modification were found using manual selection.

It should be noted that the parameters for LightSB were given only for the Adult to Child translation task, while we performed the comparison on both it and the Male to Female translation task. Due to the proximity of these tasks and the moderately good behavior of both models, it was decided to keep the parameters from the Adult to Child task.

## F.6 FINAL HYPERPARAMETER VALUES

Below are the final hyperparameter values for all experiments. It should be noted that in the experiments that follow the original LightSB paper, we used the author's parameters, leaving only the Ornstein-Uhlenbeck process parameters for tuning.

**Empirical KL Divergence Experiment:**

- Parameters for experiment: batch size $= 128$, $\varepsilon = 9.0$, lr $= 0.002$, $K = 1$, diagonal $=$ false, $b = 2.0$, $m = 0.0$.

**ALAE I2I Experiment:**

- Parameters for experiment: batch size $= 128$, $\varepsilon = 0.1$, lr $= 0.001$, $K = 10$, diagonal $=$ true, $b = 0.02$, $m = 0.0$.

**Single Cell experiment:**

- Parameters for $i = 1$ : batch size $= 128$, $\varepsilon = 0.1$, lr $= 0.01$, $K = 100$, diagonal $=$ true, $b = -0.2$, $m = 4.0$,

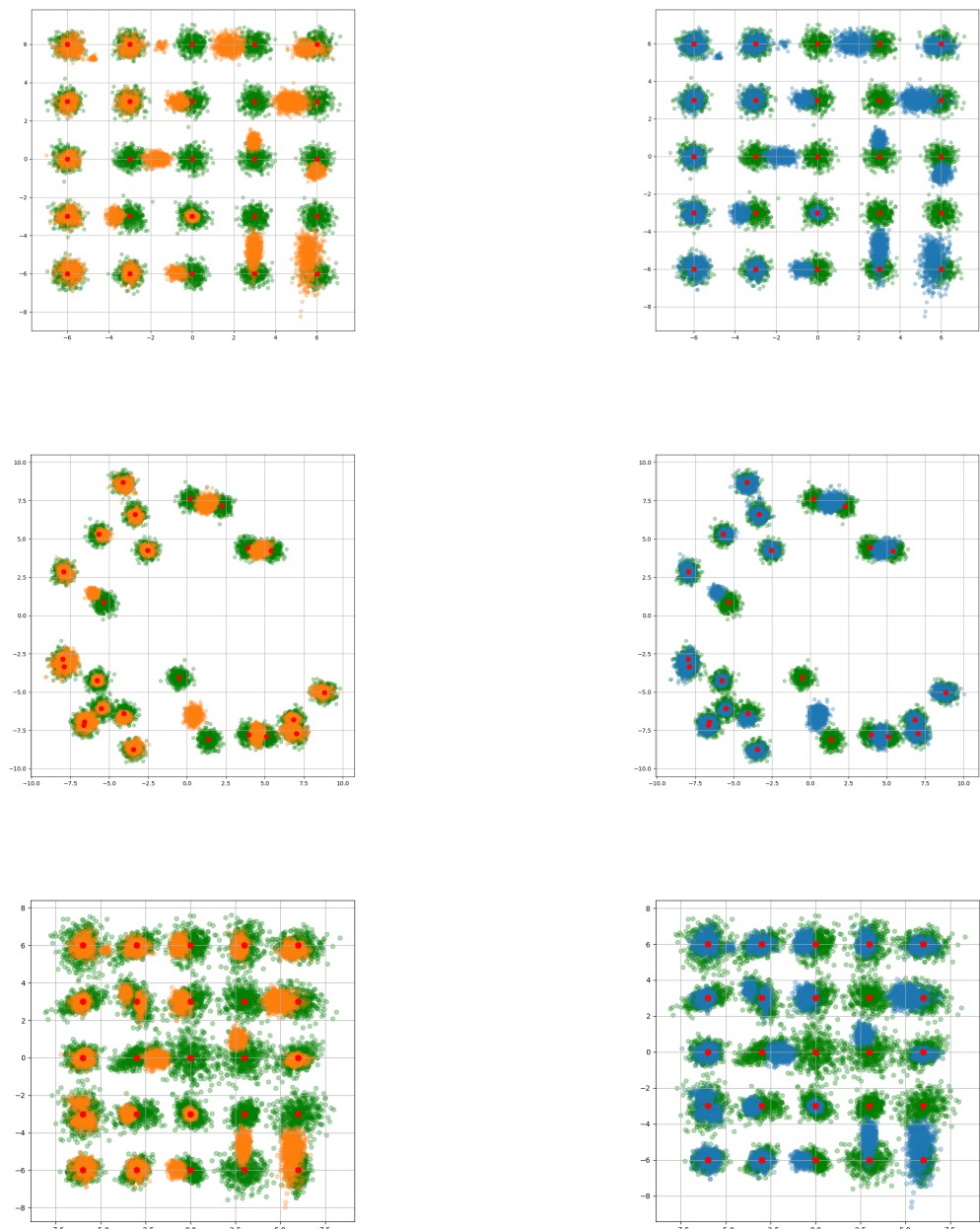

Figure 5: The translation result from a standard normal distribution using LightSB (left) and our modified approach (right). Top: uniform grid and standard covariance matrices. Middle: random location on the grid and standard covariance matrices. Bottom: uniform grid and anisotropic random covariance matrix. Green: samples from the target Gaussian mixture; orange: LightSB samples; blue: LightSB-OU samples.

- Parameters for $i = 2$ : batch size $= 128$, $\varepsilon = 0.1$, lr $= 0.01$, $K = 100$, diagonal $=$ true, $b = -0.2$, $m = 4.0$,

- Parameters for $i = 3$ : batch size $= 128$, $\varepsilon = 0.1$, lr $= 0.01$, $K = 100$, diagonal $=$ true, $b = 0.2$, $m = -1.0$.

In the experiment with a mixture of 25 Gaussians, on the contrary, we pre-selected good parameters for LightSB, and then tuned the parameters for the Ornstein-Uhlenbeck process.

It is worth noting that the process parameters $b$, $m$ remained quite stable in all three experiments with a mixture of Gaussians with respect to changing the number of components $K$ for approximating the potential, which is in line with expectations. The process should facilitate transport independently of subsequent control, and therefore is likely to be unique and stable with respect to the hyperparameters in a well-posed problem.

**Standard Gaussian mixture with 25 components:**

- Parameters for $K = 15$ : batch size $= 128$, $\varepsilon = 0.1$, lr $= 0.002$, $K = 15$, diagonal $=$ true, $b = -0.125$, $m = 0.901$,
- Parameters for $K = 20$ : batch size $= 128$, $\varepsilon = 0.1$, lr $= 0.002$, $K = 20$, diagonal $=$ true, $b = -0.125$, $m = 0.901$,
- Parameters for $K = 25$ : batch size $= 128$, $\varepsilon = 0.1$, lr $= 0.002$, $K = 25$, diagonal $=$ true, $b = 0.232$, $m = 0.197$,
- Parameters for $K = 30$ : batch size $= 128$, $\varepsilon = 0.1$, lr $= 0.002$, $K = 30$, diagonal $=$ true, $b = 0.101$, $m = 0.416$.

**Irregular Gaussian mixture with 25 components:**

- Parameters for $K = 15$ : batch size $= 128$, $\varepsilon = 0.1$, lr $= 0.002$, $K = 15$, diagonal $=$ true, $b = -0.125$, $m = 0.901$,
- Parameters for $K = 20$ : batch size $= 128$, $\varepsilon = 0.1$, lr $= 0.002$, $K = 20$, diagonal $=$ true, $b = 0.332$, $m = -0.575$,
- Parameters for $K = 25$ : batch size $= 128$, $\varepsilon = 0.1$, lr $= 0.002$, $K = 25$, diagonal $=$ true, $b = 0.332$, $m = -0.575$,
- Parameters for $K = 30$ : batch size $= 128$, $\varepsilon = 0.1$, lr $= 0.002$, $K = 30$, diagonal $=$ true, $b = 0.332$, $m = -0.575$.

**Anisotropic Gaussian mixture with 25 components:**

- Parameters for $K = 15$ : batch size $= 128$, $\varepsilon = 0.1$, lr $= 0.002$, $K = 15$, diagonal $=$ true, $b = -0.125$, $m = 0.901$,
- Parameters for $K = 20$ : batch size $= 128$, $\varepsilon = 0.1$, lr $= 0.002$, $K = 20$, diagonal $=$ true, $b = -0.125$, $m = 0.901$,
- Parameters for $K = 25$ : batch size $= 128$, $\varepsilon = 0.1$, lr $= 0.002$, $K = 25$, diagonal $=$ true, $b = 0.232$, $m = 0.197$,
- Parameters for $K = 30$ : batch size $= 128$, $\varepsilon = 0.1$, lr $= 0.002$, $K = 30$, diagonal $=$ true, $b = 0.101$, $m = 0.416$.

CONTENTS

