# OpenReview forum: "Tight Bounds for Schrodinger Potential Estimation in Unpaired Data Translation"
_ICLR.cc/2026/Conference — ICLR 2026 Poster_

### Official Review · Reviewer_Hswo · 2025-10-31

**Soundness:** 2
**Presentation:** 3
**Contribution:** 3
**Rating:** 4
**Confidence:** 3

**Summary:**

This paper studies tight generalization bounds for the empirical risk minimizer within a class of Schrödinger Bridge (SB) potentials. The authors focus on a fundamental yet widely studied setting where the risk functional is the KL divergence and the underlying dynamics follow an Ornstein–Uhlenbeck (OU) process. Under a set of explicit assumptions (Assumptions 1–5), they establish a fast convergence rate of order $O(\log^3 (n) / n$, which significantly improves upon prior results of order ($O(1\sqrt{n}$). As a practical instantiation, the paper adapts the LightSB algorithm to the OU reference process and presents experiments on Gaussian mixture models (GMMs) and single-cell datasets.

**Strengths:**

- The primary contribution lies in deriving a notably tighter generalization bound by adopting the OU reference process. The improved rate represents a substantial theoretical advance over previous works.

- The paper is well-structured and accessible, providing clear explanations of both the strengths and limitations of the proposed analysis. The authors carefully justify each assumption, discussing its practical implications and arguing convincingly that these assumptions are reasonable or attainable in practice.

**Weaknesses:**

While the theoretical contribution is valuable, my main concerns lie in the practical validation and experimental analysis.

- This is my main concern. The empirical section does not quantitatively assess the gap between the empirical and ground-truth SB potentials with respect to the number of samples $n$. Such analysis would strengthen the connection between theory and practice. It would be particularly insightful to evaluate this gap in the case of Gaussian marginals, where the exact SB potential is analytically tractable. Furthermore, constraining the neural network parameters to satisfy the boundedness and growth conditions (related to constants $L$ and $M$) should be discussed or enforced. A direct comparison with the Brownian-motion-based LightSB (with fixed number of samples $n$) should be discussed. Finally, the influence of the time horizon on the convergence rate should be investigated.

- In Table 2, the results focus solely on distributional alignment. It would be more comprehensive to also report transport cost metrics, as well as other measures such as the FID and transport cost in image-to-image translation tasks.

- The parametrization of the exponential SB potential through a GMM may not be practical for high-dimensional or complex data. In many applications, algorithms estimate the gradient of the log potential (i.e., the control function) rather than the potential itself.


Overall, I find the theoretical result strong and meaningful. With additional experiments addressing these points (especially the first bullet point), I would be inclined to raise my evaluation score.

**Questions:**

- Could the authors discuss the practical gap between the assumptions required by the theorem and the actual behavior of the LightSB implementation?

- Could the authors provide an intuitive explanation for why the OU process yields a tighter bound? Specifically, since the required time horizon  $T$ is roughly proportional to $1/b$, a larger $b$ would make the terminal reference distribution more Gaussian-like, and the joint distribution of the reference dynamics nearly independent (memoryless), as discussed in [1]. Is this correct? Could author provide the better intuition?

- Is it possible to extend the theoretical discussion to convergence guarantees for the gradient of the potential function? If so, are there existing works addressing this direction?

Reference

[1] Domingo-Enrich et al., Adjoint Matching.

---

> ### Author Response · Authors · 2025-11-21
>
> **1.** *This is my main concern. The empirical section does not quantitatively assess the gap between the empirical and ground-truth SB potentials with respect to the number of samples $n$. Such analysis would strengthen the connection between theory and practice. It would be particularly insightful to evaluate this gap in the case of Gaussian marginals, where the exact SB potential is analytically tractable. Could the authors discuss the practical gap between the assumptions required by the theorem and the actual behavior of the LightSB implementation?*
>
> In the revised version we added a plot showing the dependence of $\mathrm{KL}(\pi^\star, \widehat\pi)$ on the sample size $n$ in the Gaussian case (where the log-potential $\varphi^\star$ and the corresponding coupling $\pi^\star$ can be computed explicitly). The convergence curve is close to the theoretical dependence $\mathcal O(\log^3 n/n)$. Hence, the actual behavior of the LightSB-OU implementation agrees with our theoretical findings quite well.
>
>
> **2.** *Furthermore, constraining the neural network parameters to satisfy the boundedness and growth conditions (related to constants $L$ and $M$) should be discussed or enforced.*
>
> Neural networks with Lipschitz activations and bounded weights will automatically satisfy the growth condition $-\varphi(x) \leq L ||\Sigma^{-1/2}(x - m)||^2 + M$ with some constants $L$ and $M$ (because they are Lipschitz on $\mathbb R^d$). For this reason, we do not need to specifically enforce this constraint. To satisfy the requirement $\varphi(x) \leq M$, a learner can truncate the neural network output at level $M$. The constants $L$ and $M$ should be considered as parameters of the problem depending on the choice of the reference class $\Phi$.
>
>
> **3.** *A direct comparison with the Brownian-motion-based LightSB (with fixed number of samples $n$) should be discussed.*
>
> We have already compared our approach with LightSB in the sections with theoretical results and numerical experiments. From theoretical point of view, our high-probability upper bound is tighter than the one of [Korotin et al., 2024]. We also would like to note that [Korotin et al., 2024] do not track the dependence of any parameters, such as dimension, number of parameters, etc. For this reason, we can compare the dependence on $n$ only. Furthermore, as reported in the initial submission, LightSB-OU evidently outperforms LightSB in the numerical experiments with Gaussian mixtures and Single-cell data. This improvement is completely due to the choice of the base process, since both methods use similar classes of Schrödinger potentials.
>
>
> **4.** *Finally, the influence of the time horizon on the convergence rate should be investigated.*
>
> If $bT \geq bT_0 \gtrsim \log\log n$, then the rate of convergence is as reported in Theorem 1. Otherwise, we can prove only slow rates $\mathcal O(n^{1/2})$ following the same approach as in [Korotin et al., 2024] based on Rademacher complexities. The reason is that if $T$ does not exceed a threshold, then the loss class does not meet Bernstein's condition. We are not sure if any intermediate rates (between $\mathcal O(\log^3 n / n)$ and $\mathcal O(n^{-1/2})$) are possible if the condition $bT \geq bT_0 \gtrsim \log\log n$ is slightly violated.
>
>
> **5.** *In Table 2, the results focus solely on distributional alignment. It would be more comprehensive to also report transport cost metrics, as well as other measures such as the FID and transport cost in image-to-image translation tasks.*
>
> In Table 2, we use Wasserstein-1 distances, which is a particular case of transport cost metrics. As we reported in our initial submission, the results for all solvers but LightSB and LightSB-OU are from [Tong et al., 2023]. We did not performed the experiments with those solvers by ourselves.
>
> We report FIDs for the image-to-image translation task in the revised version. We also would like to note that, according to its definition, FID is the Wasserstein-2 distance between Gaussian distributions fitted to target and generated samples. Hence, FID can also be considered as a kind of transport cost metric.
>
> [Tong et al., 2023] Alexander Tong, Nikolay Malkin, Kilian Fatras, Lazar Atanackovic, Yanlei Zhang, Guillaume Huguet, Guy Wolf, and Yoshua Bengio. Simulation-free Schr¨odinger bridges via score and flow matching. Preprint. ArXiv:2307.03672, 2023.

---

> ### Author Response · Authors · 2025-11-21
>
> **6.** *The parametrization of the exponential SB potential through a GMM may not be practical for high-dimensional or complex data. In many applications, algorithms estimate the gradient of the log potential (i.e., the control function) rather than the potential itself. Is it possible to extend the theoretical discussion to convergence guarantees for the gradient of the potential function? If so, are there existing works addressing this direction?*
>
> We did not suggest to approximate Schrödinger potential by a Gaussian mixture in complex tasks. The goal of the numerical experiments was to illustrate that the proper choice of the base process can lead to notable improvements in data generation quality.
>
> We discuss the accuracy of the optimal control estimation in the revised version. The upper bound on the squared $L_2$-error follows immediately from Theorem 1 and Girsanov's theorem for SDE (see, e.g., [Domingo-Enrich et al., 2025], eq. (138))
>
> [Domingo-Enrich et al., 2025] Carles Domingo-Enrich, Michal Drozdzal, Brian Karrer, and Ricky T. Q. Chen. Adjoint matching: Fine-tuning flow and diffusion generative models with memoryless stochastic optimal control. ArXiv:2409.08861v5, 2025.
>
>
> **7.** *Could the authors provide an intuitive explanation for why the OU process yields a tighter bound? Specifically, since the required time horizon $T$ is roughly proportional to $1/b$, a larger $b$ would make the terminal reference distribution more Gaussian-like, and the joint distribution of the reference dynamics nearly independent (memoryless), as discussed in [1]. Is this correct? Could author provide the better intuition?*
>
> It is better to say that a larger $bT$ (rather than larger $b$) would make the terminal reference distribution more Gaussian-like, and the joint distribution of the reference dynamics nearly independent (memoryless). However, in our paper we discussed that, if $bT = \mathcal O(\log\log n)$, then the reference does not forget the initial distribution completely. This property is important when we are talking about data translation tasks.

---

### Official Review · Reviewer_TRip · 2025-10-31

**Soundness:** 3
**Presentation:** 2
**Contribution:** 2
**Rating:** 4
**Confidence:** 3

**Summary:**

In this paper, the authors introduce novel bounds for the estimation of the Kullback-Leibler divergence between a candidate distribution defined by its potential and the Schrodinger Bridge target. More precisely, they show that under strong assumptions on both the potential and using a Ornstein-Uhlenbeck reference dynamics they can provide a non-asymptotic upper bound on the empirical risk minimizer and the target Schrodinger Bridge. The bound is of order $O(1/n)$ where $n$ is the number of data points used in the approximation and $O(d)$ (up to logarithmic terms in both cases). Some of the technical lemmas seem to be inspired by [2]. In addition, of this main theoretical contribution, the authors consider the procedure of [1] but instead replace the Brownian reference motion by a Ornstein-Uhlenbeck process. In that case, they present experimental results on several benchmarks.

[1] Korotin et al. (2024) -- Light Schrodinger bridge

[2] Puchkin et al. (2025) -- Sample complexity of Schrodinger potential estimation

**Strengths:**

* The paper seems technical strong. In particular, there is an extensive discussion on the error bounds obtained in the literature and the ones obtained in this paper. In particular, the authors draw comparisons with [1].

* The theory on Schrodinger Bridge is still sparse and even though are extremely strong in the present work the authors derived a novel and interesting result.

[1] Korotin et al. (2024) -- Light Schrodinger bridge

**Weaknesses:**

* The presentation of the paper is quite hard to follow. In particular, the introduction is extremely long and seems to merge both the contribution and the related works. In particular, while I understand that it is a choice of the authors, I found it quite hard to follow the related work section. In particular, I did not understand the discussion with [1,2]. Related to this issue, I found the related work to be quite poor. There is no mention on the impact of Schrodinger Bridge work and its application in Machine Learning. For example, Diffusion Schrodinger Bridge [3] but also Stochastic Interpolants [4] and Adversarial approaches [5] are competitive works which are not mentioned (some of them are compared with in the Numerical Experiments section but it is never mentioned why those works are relevant).

* The assumptions are extremely strong. While I understand that the results obtained by the authors are new, the class of target distributions and reference potentials that is under consideration is extremely limited. This strongly limits the theoretical impact of the paper.

* The methodology contribution is incremental. In the applications, the authors consider Light SB which indeed fits their framework. However, the only modification to this work is in the reference process considered. The results obtained are quite underwhelming (see Table 2 for instance where the methods is similar to Light SB in performance and not as good as OT-CFM [7]). I do appreciate however that the authors highlight the limitations of their approach "We emphasize that these examples are not intended to suggest
universal superiority but rather to showcase specific strengths of our method in certain cases."

* In the theoretical results presented, even though this is discussed, I find the fact that the time must grow with the number of samples to be a bit concerning. Even though I agree that the doubly logarithmic dependence (which has a logarithmic effect on the regularisation due to the exponential convergence of the Ornstein-Uhlenbeck) can be mitigated, it is concerning that the time grows as the number of samples grows. I see this as one of  the main limitation of the paper.

[1] Pooladian et al. (2024) -- Plug-in estimation of Schrodinger bridges

[2] Tang et al. (2024) -- Simplified diffusion Schrodinger bridge

[3] De Bortoli et al. (2021) -- Diffusion Schrödinger Bridge with Applications to Score-Based Generative Modeling

[4] Albergo et al. (2023) -- Stochastic Interpolants: A Unifying Framework for Flows and Diffusions

[5] Gushchin et al. (2024) -- Adversarial Schrödinger Bridge Matching

[6] Korotin et al. (2024) -- Light Schrodinger bridge

[7] Tong et al. (2023) -- Improving and generalizing flow-based generative models with minibatch optimal transport

**Questions:**

* $\hat{\pi}$ in Page 3 hasn't been introduced yet.

* See my question above on the time dependency.

* Using a Ornstein-Uhlenbeck as a reference measure for Schrodinger Bridges is not new. For instance, it was used in [1]. In addition, it can be shown quite easily that using a Ornstein-Uhlenbeck as a reference path will lead to a quadratic cost function with a regularisation parameter that grows exponentially with the time of the process. It would be good if the authors could discuss this fact in the main paper.

[1] Shi et al. -- Diffusion Schrödinger Bridge Matching

---

> ### Author Response · Authors · 2025-11-21
>
> **1.** *The presentation of the paper is quite hard to follow. In particular, the introduction is extremely long and seems to merge both the contribution and the related works. In particular, while I understand that it is a choice of the authors, I found it quite hard to follow the related work section. In particular, I did not understand the discussion with [1,2]. Related to this issue, I found the related work to be quite poor. There is no mention on the impact of Schrodinger Bridge work and its application in Machine Learning. For example, Diffusion Schrodinger Bridge [3] but also Stochastic Interpolants [4] and Adversarial approaches [5] are competitive works which are not mentioned (some of them are compared with in the Numerical Experiments section but it is never mentioned why those works are relevant).*
>
> [1] Pooladian et al. (2024) -- Plug-in estimation of Schrodinger bridges
>
> [2] Tang et al. (2024) -- Simplified diffusion Schrodinger bridge
>
> [3] De Bortoli et al. (2021) -- Diffusion Schrödinger Bridge with Applications to Score-Based Generative Modeling
>
> [4] Albergo et al. (2023) -- Stochastic Interpolants: A Unifying Framework for Flows and Diffusions
>
> [5] Gushchin et al. (2024) -- Adversarial Schrödinger Bridge Matching
>
> In the initial version, we tried to focus on the most relevant papers. Unfortunately, only in few works the authors are interested in statistical guarantees for Schr\"odinger potential estimation. For instance, in the aforementioned papers [3]-[5], the authors focus on methodological aspects and leave non-asymptotic high-probability bounds beyond the scope of their papers. For this reason, the related work section looked scantily. In the revised version, we followed your recommendation and added a short overview of papers applying Schr\"odinger bridge framework to generative modelling. We also corrected the discussion with [1], and hope that it became clearer.
>
>
> **2.** *The assumptions are extremely strong. While I understand that the results obtained by the authors are new, the class of target distributions and reference potentials that is under consideration is extremely limited. This strongly limits the theoretical impact of the paper.*
>
> We would like to disagree that our assumptions on the classes of target distributions and reference potentials are extremely strong. In contrast to previous theoretical papers on Schrödinger bridge estimation [Korotin et al., 2024] and [Pooladian and Niles-Weed, 2024], we do not require the target density $\rho_T$ to have a compact support. Instead, we assume that it belongs to a large subclass of sub-Gaussian distributions.
>
> Furthermore, while in the numerical experiments the reference Schr\"odinger potentials are Gaussian mixtures, in our theoretical study we impose quite general assumptions. For instance, our assumptions on reference log-potentials are satisfied for classes of feed-forward neural networks with Lipschitz activations. Let us also note that the most popular activation functions, such as ReLU and sigmoid, are Lipschitz. For this reason, our theoretical analysis covers a wide spectrum of reference log-potential classes.
>
> [Korotin et al., 2024] A. Korotin, N. Gushchin, and E. Burnaev. Light Schrödinger bridge. In The Twelfth International Conference on Learning Representations, 2024.
>
> [Pooladian and Niles-Weed, 2024] A.-A. Pooladian and J. Niles-Weed. Plug-in estimation of Schrödinger bridges. Preprint, arXiv:2408.11686, 2024

---

> ### Author Response · Authors · 2025-11-21
>
> **3.** *The methodology contribution is incremental. In the applications, the authors consider Light SB which indeed fits their framework. However, the only modification to this work is in the reference process considered. The results obtained are quite underwhelming (see Table 2 for instance where the methods is similar to Light SB in performance and not as good as OT-CFM [7]). I do appreciate however that the authors highlight the limitations of their approach "We emphasize that these examples are not intended to suggest universal superiority but rather to showcase specific strengths of our method in certain cases."*
>
> [7] Tong et al. (2023) -- Improving and generalizing flow-based generative models with minibatch optimal transport.
>
> As reported in the introduction (see the paragraph ``Contribution''), the main contribution of the paper is theoretical, rather than methodological. Our methodological findings directly follow from theoretical analysis. We emphasize that our work is, to the best of our knowledge, the first to \emph{quantify the statistical error} under \emph{general assumptions on the target density}, specifically for subgaussian~$\rho_T$, and to \emph{characterize its dependence on the approximation error}. It is worth noting that the convergence of the Sinkhorn algorithm and related iterative schemes is not guaranteed under these assumptions; see, for instance, [Conforti et al., 2023].
>
> We disagree that the results of numerical experiments are underwhelming. First, as we mentioned in our paper, we did not pursue the goal to beat state-of-the-art solvers. The goal of the numerical experiments was only to illustrate that a proper choice of the reference process can lead to notable improvements in generative modelling. Second, it is incorrect to compare LightSB-OU with OT-CFM directly, because the last solver requires much more computational resourses (while training of LightSB and LightSB-OU can be performed on a laptop). Finally, OT-CFM and $[\mathrm{SF}]^2$M-Exact have much larger variance, than LightSB and LightSB-OU. At the same time the experiments clearly indicate that LightSB-OU outperforms LightSB. This improvement is completely due to the choice of the base process, because both methods use similar reference classes of potentials.
>
> [Conforti et al., 2023] G. Conforti, A. Durmus, and G. Greco. Quantitative contraction rates for Sinkhorn algorithm: beyond bounded costs and compact marginals. Preprint, arXiv:2304.04451, 2023.
>
>
> **4.** *In the theoretical results presented, even though this is discussed, I find the fact that the time must grow with the number of samples to be a bit concerning. Even though I agree that the doubly logarithmic dependence (which has a logarithmic effect on the regularisation due to the exponential convergence of the Ornstein-Uhlenbeck) can be mitigated, it is concerning that the time grows as the number of samples grows. I see this as one of the main limitation of the paper.*
>
> The condition $bT \gtrsim \log\log n$ should be considered as a condition on $b$ and $T$, rather than on the time horizon alone. Our results still allow us to take $T = 1$, we just need to choose $b \gtrsim \log\log n$ in this case. Hence, we do not require the time horizon to grow with the sample size.
>
> At the same time, $bT$ must increase proportionally to $\log\log n$. This is an extremely slow growth. Such a requirement looks as a reasonable price for substantial improvements in the rates of convergence. We are not aware of any results establishing similar rates of convergence under milder assumptions on $T$.
>
>
> **5.** *$\widehat\pi$ in Page 3 hasn't been introduced yet.*
>
> $\widehat\pi$ is defined in eq. (12) and in Table 1 on page 3.

---

> ### Author Response · Authors · 2025-11-21
>
> **6.** *Using a Ornstein-Uhlenbeck as a reference measure for Schrodinger Bridges is not new. For instance, it was used in [1]. In addition, it can be shown quite easily that using a Ornstein-Uhlenbeck as a reference path will lead to a quadratic cost function with a regularisation parameter that grows exponentially with the time of the process. It would be good if the authors could discuss this fact in the main paper.*
>
> [1] Shi et al. -- Diffusion Schrödinger Bridge Matching
>
> First, we would like to note that the paper [Shi et al., 2023] employs the Ornstein–Uhlenbeck (OU) process in a different context, not as a reference process in our formulation. Their approach is based on the time reversal of a forward unconditional OU diffusion process (cf. Eq. (B.2.2)), which aligns more closely with the framework of diffusion-based generative modeling. In contrast, our method does not rely on iterative procedures such as Iterative Proportional Fitting (IPF) or Iterative Markovian Fitting (IMF); instead, it is formulated as a single empirical risk minimization problem corresponding to the Kullback–Leibler divergence between couplings under the reference process. Second, replacement of Brownian motion by the OU process is just an initial step towards our main result, improved generalization bounds for Schr\"odinger potential estimation. In [Shi et al., 2023], leave this question out of the scope of their paper.  Hence, the novelty of using the OU process should not be evaluated separately from the main result.
>
> The connection between the Schr\"odinger bridge problem with entropic optimal transport was already discussed in [Leonard, 2014] (in fact, the author considers more general reference  processes). We are afraid that adding this discussion into introduction will not bring any novelty but will affect readability in a negative way (especially taking into account that we do not need this fact in our analysis). Exponential dependence of the penalty on $bT$ does not affect our results.
>
> [Leonard, 2014] Christian Leonard. A survey of the Schrödinger problem and some of its connections with optimal transport. Discrete and Continuous Dynamical Systems, 34(4):1533–1574, 2014.

---

### Official Review · Reviewer_ibUA · 2025-11-01

**Soundness:** 3
**Presentation:** 4
**Contribution:** 3
**Rating:** 6
**Confidence:** 4

**Summary:**

**Disclaimer**

*I do not have specific expertise in quantitative generalization bound estimation. Therefore, while I provide an assessment of the clarity, motivation, and overall contribution of the paper, I will defer to other reviewers with stronger expertise in the theoretical aspects when evaluating the technical soundness of the quantitative rate derivation.*

**Summary**

This paper studies the finite-sample generalization behavior of Schrödinger bridge (SB) estimation for unpaired data translation. While most prior works adopt Brownian motion as the reference process, this work introduces the Ornstein–Uhlenbeck (OU) process as the reference dynamics. The OU process exhibits exponential mixing, which enables a more stable and analytically tractable setting for sample-based SB estimation. The paper establishs the first non-asymptotic generalization bound for estimating Schrödinger potentials $\phi$ as the empirical risk minimizer and derives a convergence rate of $O(\log^{3} (n) / n)$ for the KL divergence between the optimal coupling $\pi^{*}$ and the empirical risk minimizer $\hat{\pi}$ (when the approximation error $\Delta$ is small). Empirically, the paper extends the Light Schrödinger Bridge framework to an OU-based version (LightSB-OU), demonstrating improved stability and translation quality in synthetic, biological (single-cell RNA), and unpaired image translation tasks.

**Strengths:**

- The paper provides a new tight finite-sample generalization bound for Schrödinger bridge (SB) estimation.
- Replacing Brownian motion with an OU process is both theoretically novel and practically beneficial.
- The proposed LightSB-OU algorithm effectively connects the theoretical framework to practical tasks. Notably, the OU process corresponds to the VP-SDE used in the diffusion model, while Brownian motion aligns with the VE-SDE. The experiments on both synthetic and real-world datasets demonstrate that the OU reference improves fidelity in unpaired data translation.
- The manuscript is well organized and clearly written. The motivation, the implications of assumptions, and the meaning of theoretical results are well described.

**Weaknesses:**

- Although the derived convergence rate of $O( \log^{3} (n) / n)$ is theoretically novel, there is no experimental study showing empirical convergence as a function of sample size. Including a convergence curve or ablation study on sample size would better demonstrate the practical significance of the bound.
- While the bound is tight in a theoretical sense, it remains unclear how much this improved rate translates into practical accuracy gains over the previous $O(n^{-1/2})$ regime established by LightSB.
- In the image-to-image translation experiments, only selected qualitative results are presented. Providing quantitative metrics, such as FID (for fidelity to target semantics) and LPIPS (for perceptual similarity between input and output), would significantly strengthen the empirical claims.

**Questions:**

- Beyond theoretical implications, are there practical improvements in unpaired translation performance (e.g., FID, MMD) that can be directly attributed to the tighter convergence rate?
- The title emphasizes "Unpaired Data Translation," but its primary contribution is theoretical. Perhaps a title that emphasizes the general results, such as "Tight Bounds for Schrödinger Potential Estimation between Empirical Distributions", would better reflect the scope of this paper? (Of course, this is left to the author's discretion.)

---

> ### Author Response · Authors · 2025-11-21
>
> **1.** *Although the derived convergence rate of $\mathcal O(\log^3 n / n)$ is theoretically novel, there is no experimental study showing empirical convergence as a function of sample size. Including a convergence curve or ablation study on sample size would better demonstrate the practical significance of the bound.*
>
> In the revised version we added a plot showing the dependence of $\mathrm{KL}(\pi^\star, \widehat\pi)$ on the sample size $n$ in the Gaussian case (where the log-potential $\varphi^\star$ and the corresponding coupling $\pi^\star$ can be computed explicitly, see Figure 1 in the revised version). The convergence curve is quite close to $\mathcal O(\log^3 n / n)$.
>
>
> **2.** *While the bound is tight in a theoretical sense, it remains unclear how much this improved rate translates into practical accuracy gains over the previous $\mathcal O(n^{-1/2})$ regime established by LightSB.*
>
> We provided numerical experiments to illustrate that replacement of the Brownian motion by the Ornstein-Uhlenbeck process leads not only to theoretical improvements but also to a simple yet efficient modification of LightSB. LightSB-OU evidently outperforms LightSB in the numerical experiments. This improvement is completely due to the choice of the base process, since both methods use similar classes of Schrödinger potentials.
>
>
> **3.** *In the image-to-image translation experiments, only selected qualitative results are presented. Providing quantitative metrics, such as FID (for fidelity to target semantics) and LPIPS (for perceptual similarity between input and output), would significantly strengthen the empirical claims. Beyond theoretical implications, are there practical improvements in unpaired translation performance (e.g., FID, MMD) that can be directly attributed to the tighter convergence rate?*
>
> We reported FIDs for the image-to-image translation task in the revised version. According to our results, LightSB-OU is slightly better, than the original LightSB algorithm. Similarly to the examples with Gaussian mixtures and Single-cell data, this improvement is completely due to the choice of the base process, since both methods use similar classes of Schrödinger potentials.
>
>
> **4.** *The title emphasizes "Unpaired Data Translation," but its primary contribution is theoretical. Perhaps a title that emphasizes the general results, such as "Tight Bounds for Schrödinger Potential Estimation between Empirical Distributions", would better reflect the scope of this paper? (Of course, this is left to the author's discretion.)*
>
> We use the words ``unpaired data translation'' in the title to distinguish from the papers focusing on unconditional generative modelling (i.e. on generating data from noise). Let us also note that we estimate the Schrödinger bridge between absolutely continuous distributions with densities $\rho_0$ and $\rho_T$, rather than between empirical ones.

---

### Official Review · Reviewer_BNq5 · 2025-11-01

**Soundness:** 3
**Presentation:** 3
**Contribution:** 3
**Rating:** 6
**Confidence:** 2

**Summary:**

This paper provides learning theory style high probability bounds for estimating Schrodinger bridge potentials when the reference dynamics is OU process. This is a theoretical paper where the main result is a nonasymptotic, high probability bound improving prior work.

**Strengths:**

- The paper is mathematically strong, well structured, well motivated on its own domain.
- There are mathematical novelties, for example, the high-probability bound is novel compared to prior work.
- The derived convergence rate is fast.

**Weaknesses:**

- Some assumptions of the paper are probably unrealistic for real-world scenarios
- The bound, while a nonasymptotic, high-probability bound, is not very clean.
- Practical implications of the result is not clear, there's a gap between theory and why how it explains the empirical improvements.

**Questions:**

- notation is dense. A short “notation table” early in the paper would help.
- Can Theorem 1 extend to sub-Gaussian cases? Please discuss Assumption 2 and its applicability in real-world settings.
- The authors claim the dependence of the bound to dimension is 'nearly linear'. The bound itself, however, is not clearly written. It would be great if the authors provide a corollary, perhaps cleaning up some unnecessary quantities and display the dimension dependence clearly.
- Please provide a discussion on when $\Delta$ can be made small (for the favourable convergence rate), i.e., for what classes of initial/target distributions, what families would make it small?
- The theoretical analysis is elegant, but it’s not clear what it tells practitioners: (i) Does the theorem suggest a way to choose the OU drift parameter b? (ii) Does it explain why LightSB-OU performs better numerically?

It is this last point I am more keen on seeing explained -- I appreciate the paper's main contribution is theoretical but it would be still good to understand how theoretical bound would give insights for practical improvements.

---

> ### Author Response · Authors · 2025-11-21
>
> **1.** *Some assumptions of the paper are probably unrealistic for real-world scenarios. Can Theorem 1 extend to sub-Gaussian cases? Please discuss Assumption 2 and its applicability in real-world settings.*
>
> The assumption that $\rho_0$ is compactly supported seems realistic, at least for the problem of image-to-image translation, where the data typically occupy a bounded region of the ambient space (for instance, pixel intensities lie within a finite range). Moreover, compact support helps to avoid technical complications related to integrability and tail behavior, which can otherwise obscure the main analytical aspects of the problem. Note that, in contrast, $\rho_T$ is assumed to be only subgaussian in our setting.
>
> It is worth emphasizing that most theoretical works on the Schrödinger bridge problem impose even more stringent assumptions on the marginals or on the cost structure. For example, [Conforti et al., 2024], consider settings where the measures possess additional regularity properties or satisfy exponential moment bounds. Such conditions ensure the existence of certain transport inequalities and contraction estimates. Although these assumptions may appear restrictive from a practical perspective, they are often essential for obtaining sharp quantitative convergence results for iterative schemes such as the Sinkhorn algorithm, or for guaranteeing well-posedness of the entropic interpolation in the absence of compactness.
>
> In this sense, the compact support hypothesis for $\rho_0$ together with the sub-Gaussian behavior of $\rho_T$ adopted here can be regarded as a reasonable compromise between practical relevance and mathematical tractability: it reflects the empirical structure of many data-driven problems while remaining compatible with the analytical framework developed in the theoretical literature on the Schrödinger bridge. Finally, we would like to mention that an extension to a sub-Gaussian density $\rho_0$ may be quite challenging (in particular, it requires an extension of Lemma B.1 to the sub-Gaussian case). For this reason, it goes beyond the scope of the present paper.
>
> [Conforti et al., 2023] G. Conforti, A. Durmus, and G. Greco. Quantitative contraction rates for Sinkhorn algorithm: beyond bounded costs and compact marginals. Preprint, arXiv:2304.04451, 2023.
>
>
> **2.** *The bound, while a nonasymptotic, high-probability bound, is not very clean. The authors claim the dependence of the bound to dimension is 'nearly linear'. The bound itself, however, is not clearly written. It would be great if the authors provide a corollary, perhaps cleaning up some unnecessary quantities and display the dimension dependence clearly.*
>
> We added a remark right after Theorem 1, where we provide a simplified bound tracking the dependence on $n$, $D$, $d$ and $\delta$. In particular, the bound scales as $\mathcal O(d \log^2 d)$.
>
>
> **3.** *Notation is dense. A short “notation table” early in the paper would help.*
>
> We added a table with frequently used notations in the revised version.
>
>
> **4.** *Please provide a discussion on when $\Delta$ can be made small (for the favourable convergence rate), i.e., for what classes of initial/target distributions, what families would make it small?*
>
> $\Delta$ denotes the best approximation error within our chosen function class. This error can be made arbitrarily small by enlarging the family of potentials $\Phi$. For instance, one may consider the class of deep neural networks satisfying our structural assumptions (Assumptions 4 and 5), as in [Schmidt-Hieber, 2020]. However, a rigorous analysis of the approximation error would require detailed regularity results for the optimal potential $\varphi^*$ in our noncompact setting (due to the subgaussian $\rho_T$). Establishing such regularity remains an open problem and lies beyond the scope of the present work.
>
> [Schmidt-Hieber, 2020] J. Schmidt-Hieber. Nonparametric regression using deep neural networks with ReLU activation function. The Annals of Statistics, 48(4):1875–1897, 2020.

---

> ### Author Response · Authors · 2025-11-21
>
> **5.** *Practical implications of the result is not clear, there's a gap between theory and why how it explains the empirical improvements. The theoretical analysis is elegant, but it’s not clear what it tells practitioners: (i) Does the theorem suggest a way to choose the OU drift parameter $b$? (ii) Does it explain why LightSB-OU performs better numerically? It is this last point I am more keen on seeing explained -- I appreciate the paper's main contribution is theoretical but it would be still good to understand how theoretical bound would give insights for practical improvements.*
>
> (i) Theorem 1 tells that if the parameters $b$ and $T$ are chosen in such a way that $bT$ is of order $\log\log n$, then a learner can hope for faster rates of convergence than $\mathcal O(n^{-1/2})$. Practitioners can take this insight into account.
>
> (ii) No, Theorem 1 studies just the performance of LightSB-OU. It does not compare directly LightSB and LightSB-OU. However, during our study of generalization bounds for LightSB we realized that we can significantly improve the result of [Korotin et al., 2024], if we take a geometrically ergodic base process, instead of the Brownian motion. Our theoretical findings lead to a simple but efficient modification of LightSB. We demonstrate that LightSB-OU outperforms LightSB in the numerical experiments with Gaussian mixtures and Single-cell data. This improvement is completely due to the choice of the base process.
>
> In fact, the main message of our paper is that employing more general reference processes can be practically beneficial while still admitting a sound theoretical justification. The Ornstein–Uhlenbeck (OU) process, in particular, occupies a prominent place in diffusion-based generative modeling, where it serves as a natural prior due to its stationary Gaussian structure and well-understood dynamics. However, its potential within the framework of the Schrödinger bridge problem has remained largely unexplored.
>
> Our work takes a first step in this direction by investigating the use of the OU process as a reference dynamics for image-to-image translation tasks. We demonstrate that this choice not only aligns well with the empirical characteristics of such problems (where data are typically centered and exhibit subgaussian behavior) but also allows for a rigorous theoretical treatment within the Schrödinger bridge formalism. In this sense, our contribution bridges a gap between the practical advances in diffusion modelling and the analytical framework of entropic optimal transport, showing that more structured reference processes can enhance both interpretability and performance.
>
> [Korotin et al., 2024] A. Korotin, N. Gushchin, and E. Burnaev. Light Schrödinger bridge. In The Twelfth International Conference on Learning Representations, 2024.

---

### Author Response · Authors · 2025-11-21

We are grateful to the reviewers for the constructive feedback. We addressed the questions point by point and made the corresponding changes in the submission.

---

### Author Response · Authors · 2025-12-03
**Short summary of the rebuttal**

Dear AC,

We provide a brief summary of our rebuttal. Below, we outline the key concerns raised by the referees and our responses.

**1. Strong assumptions.** We argue that our assumptions hold for quite general classes of distributions and log-potentials. Initial and target distributions with compact supports are common for theoretical papers on Schrödinger bridges. However, in our paper, we are able to deal with target distributions with unbounded supports. The admissible reference classes of log-potentials include feed-forward neural networks with Lipschitz activations.

**2. No experimental evidence of $\mathcal O(\log^3(n) / n)$ rates of convergence.** We plot dependence of the mean KL-divergence between couplings on the number of samples for synthetic Gaussian data. The theoretical rate $\mathcal O(\log^3(n) / n)$ agrees with the experimental data quite well.

**3. Incremental methodological contribution.** The primary contribution of the paper is theoretical. Methodological insights follow naturally from our theoretical findings. Our modifications of the LightSB procedure [Korotin et al., ICLR 2024] are simple, but they lead to consistent improvement in the performance.

**4. Assumption on the time horizon.** The rate of convergence $\mathcal O(\log^3(n) / n)$ holds under the assumption $bT \gtrsim \log\log n$. Here, $b$ is the coefficient in the Ornstein-Uhlenbeck process, $T$ is the time horizon, $n$ is the sample size, and the sign $\gtrsim$ stands for the inequality up to a positive absolute constant. However, $bT \gtrsim \log\log n$ should be considered as an assumption on $bT$, rather than on the time horizon $T$ alone. For instance, we can take $T = 1$ (as usually in the literature on Schrödinger bridges) and $b \gtrsim \log \log n$.

---

### Meta-Review · Area_Chair_EBrs · 2026-01-06

**Summary:**

This paper investigates the statistical properties of Schrödinger Bridge (SB) estimation for unpaired data translation. The core contribution is theoretical: the authors replace the standard Brownian motion reference process with an Ornstein-Uhlenbeck (OU) process. By leveraging the mixing properties of the OU process, they derive a non-asymptotic high-probability generalization bound with a fast convergence rate of $\tilde{O}(1/n)$, improving upon the standard $O(1/\sqrt{n})$ rates found in prior work. Methodologically, they propose LightSB-OU, a variation of the LightSB algorithm, and demonstrate its performance on synthetic (Gaussian mixtures), single-cell, and image-to-image translation tasks.

The primary concerns informing the decision process revolved around the balance between theoretical rigor and practical relevance. Initially, reviewers were divided on whether the theoretical assumptions (e.g., compact support for $\rho_0$, though relaxed to sub-Gaussian for $\rho_T$) were too restrictive for real-world applications and whether the method was sufficiently novel given its similarity to LightSB. A critical gap identified during the review process was the lack of empirical validation for the claimed $\tilde{O}(1/n)$ convergence rate, which made the theoretical contribution feel disconnected from the experimental results. Furthermore, the "incremental" nature of the methodological changes (swapping the reference process) raised questions about whether the practical gains (modest improvements in FID/Wasserstein metrics) justified acceptance.

Ultimately, I recommend acceptance because the authors successfully bridged the most critical gap, the validation of their theoretical bounds, during the rebuttal. The addition of empirical convergence plots confirming the $\tilde{O}(1/n)$ rate provides strong evidence that the theoretical contribution is sound and meaningful. While the method may be a variation of existing work, establishing and verifying tighter generalization bounds is a significant contribution to the Learning Theory track. The paper succeeds as a rigorous theoretical work that offers clear insights into SB estimation, outweighing the concerns about incremental practical gains.

**Reviewer Concerns:**

The initial reviews were mixed (6, 6, 4, 4), with a split between appreciating the theoretical novelty and concerns regarding empirical validation and assumptions. There were 3 main concerns that the authors addressed:

* A major blocker was the lack of experimental verification of the derived convergence rate (see reviewers ibUA and Hswo). In the rebuttal, the authors added a plot (Figure 1) showing the empirical convergence rate on Gaussian data matching the theoretical prediction. This directly addressed the "main concern" of Reviewer Hswo.
* Reviewers BNq5 and TRip worried that assuming compact support for $\rho_0$ was too strong. The authors clarified that they allow the target $\rho_T$ to be sub-Gaussian (unbounded), which is a relaxation compared to competing theoretical works (e.g., Conforti et al.) that often require compact support for both.
* Metrics related: FID scores were added as requested by Reviewer ibUA.

The only outstanding concern was the incrementality of the methodology (reviewer TRip). The resulting algorithm is a relatively minor modification of LightSB. While the authors argue the contribution is theoretical, the practical gains over the baseline are modest. This concern remains valid but is less critical for the Learning Theory track.

**Reviewer Scores:**

Based on the rebuttal, I estimate the scores would have shifted as follows:

* **Reviewer BNq5: Initial: 6. Estimated Final: 6**. The reviewer was positive about the math. The clarifications (Remark 2) solidify their positive stance, though the gap between theory and practice likely prevents a jump to 8.

* **Reviewer ibUA: Initial: 6. Estimated Final: 8**. This reviewer was enthusiastic but requested convergence plots and FIDs. The authors provided both. A score increase is the rational response to a successful rebuttal.

* **Reviewer TRip: Initial: 4. Estimated Final: 6**. This reviewer was the most critical. However, the authors successfully argued that their assumptions are weaker than the literature standard. While the "incremental" critique stands, the theoretical validity likely pushes this just over the bar to a 6.

* **Reviewer Hswo: Initial: 4. Estimated Final: 6**. This reviewer explicitly stated: "With additional experiments... I would be inclined to raise my evaluation score." The authors provided the exact experiment requested (Figure 1). Keeping this at a 4 would be unfair; a move to 6 is the expected outcome.

---

### Decision · Program_Chairs · 2026-01-26

Accept (Poster)